# Neurocomputational mechanisms involved in adaptation to fluctuating intentions of others

Rémi Philippe [1,2,9], Rémi Janet[1,2,9], Koosha Khalvati[3], Rajesh P. N. Rao [3,4], Daeyeol Lee [5,6,7,8] & Jean-Claude Dreher [1,2] ✉

Humans frequently interact with agents whose intentions can fluctuate between competition and cooperation over time. It is unclear how the brain adapts to fluctuating intentions of others when the nature of the interactions (to cooperate or compete) is not explicitly and truthfully signaled. Here, we use model-based fMRI and a task in which participants thought they were playing with another player. In fact, they played with an algorithm that alternated without signaling between cooperative and competitive strategies. We show that a neurocomputational mechanism with arbitration between competitive and cooperative experts outperforms other learning models in predicting choice behavior. At the brain level, the fMRI results show that the ventral striatum and ventromedial prefrontal cortex track the difference of reliability between these experts. When attributing competitive intentions, we find increased coupling between these regions and a network that distinguishes prediction errors related to competition and cooperation. These findings provide a neurocomputational account of how the brain arbitrates dynamically between cooperative and competitive intentions when making adaptive social decisions.

During social interactions, humans are often uncertain whether others intend to compete or cooperate. The intentions of other agents can fluctuate over time, making it challenging to develop successful behavioral strategies. A key question is how the brain decides whether the other is cooperating or competing during volatile situations in which the nature of the social interactions is not explicitly determined, as when others interact to achieve a common goal while maximizing their own benefits. This question is of importance since it lies at the heart of strategic social decision making[1–9]. In these types of situations, other agents can change behavior according to cooperative or competitive intentions.

Cooperation is generally defined as involving a group of individuals working together to attain a common goal[10,11]. In contrast, competition involves one person attempting to outperform another in a zero-sum situation[12]. A number of theoretical accounts and experimental results demonstrate that the ability to mentalize, i.e. to simulate the other's belief about one's next course of action, is crucial for strategically sophisticated agents[6,7,13,14]. The neurocomputational mechanisms engaged in attributing intentions to others have been studied in situations in which participants are explicitly informed about the nature of the interactions, either in a collaborative context alone[15–17] or in a competitive context alone[7,8,18–24]. For example, during a

[1]CNRS-Institut des Sciences Cognitives Marc Jeannerod, UMR5229, Neuroeconomics, reward, and decision making laboratory, Lyon, France. [2]Université Claude Bernard Lyon 1, Lyon, France. [3]Paul G. Allen School of Computer Science and Engineering, University of Washington, Seattle, WA, USA. [4]Center for Neurotechnology, University of Washington, Seattle, WA, USA. [5]Zanvyl Krieger Mind/Brain Institute, Johns Hopkins University, Baltimore, MD, USA. [6]Kavli Discovery Neuroscience Institute, Johns Hopkins University, Baltimore, MD, USA. [7]Department of Psychological and Brain Sciences, Johns Hopkins University, Baltimore, MD, USA. [8]Department of Neuroscience, Johns Hopkins University, Baltimore, MD, USA. [9]These authors contributed equally: Rémi Philippe, Rémi Janet. ✉e-mail: dreher@isc.cnrs.fr

cooperative game such as the coordination game, one of the best strategies is to try to choose one of two presented targets consistently. In contrast, in a competitive game such as the matching pennies game[19,24], the optimal strategy is to choose between two targets equally often and randomly across trials. If the identity of the game played is not known, the agent has to adjust his/her strategy based on repeated interactions with others and to infer cooperation/competition on the basis of observations. How the brain achieves such inference poses a unique computational problem because it not only requires the recursive representation of reciprocal beliefs about other's intentions, as in cooperative or competitive contexts alone, but it also requires one to decide whether the other is competing or cooperating to deploy an appropriate behavioral strategy.

Here, we sought to determine the neurocomputational mechanisms that underlie the inferences of whether a person is competing or cooperating during volatile situations in which the nature of the interactions is not explicitly signaled. A recent computational account proposed that arbitration between strategies is determined by their predictive reliability, such that control over behavior is adaptively weighted toward the strategy with the most reliable prediction[25]. This approach has been tested successfully in the domains of instrumental or Pavlovian action selection[26], model-based and model-free learning[27] and learning by imitation or emulation[28]. Extending this concept of a mixture of experts to social interactions, we investigated whether the brain relies on distinct experts to compute the best choice between two possible intentions attributed to others (cooperation or competition) and then weights them by their relative reliability. We tested and compared these mixtures of models, that attribute intentions to others dynamically, with different classes of learning models: non-Bayesian vs Bayesian and non-mentalizing vs mentalizing (see Table 1). This allowed us to identify the algorithms and brain mechanisms engaged with a key component of estimating other's intentions, i.e., whether the social partner was cooperating or competing.

The majority of theoretical frameworks used to model feedback-dependent changes in decision making strategies, such as choice reinforcement and related Markov Decision Process (MDP) models, assume that optimal decisions can be determined from the observable events and variables by the decision makers. Clearly, these assumptions do not capture the reality and complexity of human social interactions because observable behaviors of other individuals provide only very partial information about their likely future behaviors. Moreover, model-free RL algorithms assume that values (utility or desirability of states and actions), change incrementally across trials according to the choice outcomes. This assumption is invalid when option values change abruptly, such as when the intention of the other shifts between cooperation and competition. These limitations explain

why agents basing their behavior only on standard RL models can be exploited by opponents using more sophisticated algorithms[6,29].

A more accurate account of strategic learning is based on a family of RL models which adds a mathematical term to the classical Temporal Difference (TD) algorithm to consider the other as an agent having their own policy, which can be influenced by oneself[6,29,30]. For example, fictitious play learning proposes a basic form of mentalizing by having a representation of the other's strategy. Influence models also consider that RL can be supplemented by a mentalizing term that represents how our actions influence those of others, updated through a belief prediction error[2,6,7,19,29,31–33]. Such influence models formalize not only how players react to others' past choices, (first-order beliefs in Theory of Mind: ToM), but also how they anticipate the influence of their own choices on the others' behavior, (i.e., mentalizing-related second-order beliefs). Another modeling approach of theory of mind uses Bayesian algorithms to model inferences about the future actions of another by attempting to take their point of view and to simulate their decision[13,17,34]. This strategy can be performed recursively so that participants make inferences concerning the others' inferences and so on. Such a sophisticated approach could be grounded in the theoretical framework of Partially Observable Markov Decision Processes (POMDPs)[35]. POMDPs provide a probabilistic framework for solving tasks involving action selection and decision making under uncertainty[36,37]. Notably, this approach has recently been applied to strategic cooperation in groups[35,38,39]. These models, however, have mainly been limited to signaled cooperative or competitive tasks in which the intentions of players do not change over a given period[13,34,40,41].

Here, we tested the predictions of these different families of learning models against one another, investigating not only non-Bayesian vs Bayesian models and non-mentalizing vs mentalizing models, but also a mixture of models deploying an arbitration process whereby the influence of attributing intentions to others is dynamically modulated, depending on which type of intention (i.e. cooperative vs competitive) is most suitable to guide behavior at a given time. For the mixture of models, two expert systems implement an identical influence learning process, working together to make strategic decisions. These two experts differ only by their priors, one expert assessing competitive intentions and the other assessing cooperative intentions, while a controller weight between these experts according to their relative reliabilities. Each expert system uses a classic RL algorithm complemented with a mentalizing term to infer the other's actions. This hypothesis extends the mixture of models to a more general view in which the experts are differentiated not by their cognitive processes, such as model based vs. model-free learning or emulation vs. imitation[27,28], but by their priors.

To determine the neurocomputational mechanisms that underlie the inferences of whether a person is competing or cooperating during volatile situations in which the nature of the interactions is not explicitly signaled, we used a novel model-based fMRI design (Fig. 1). We used an iterative dyadic game in which participants were told that they would interact with another person via a computer. Unbeknownst to them, the other player was an artificial agent that switched between blocks of cooperative trials and blocks of competitive trials when playing a card matching game. Thus, the opponent algorithm's goals were the same as those of participants in the Cooperative blocks but were opposite in Competitive blocks. Participants remained uncertain with respect to the goals of their "partner" or "opponent", which alternated, without this being signaled. This task allowed us to investigate the algorithms used by the brain to recognize the "intentions" of others and to adopt appropriate strategies when the modes of interaction (cooperation vs competition) are not indicated.

We show that a mixture of influence models best accounted for the behavior in our task, referred to as the mixed-intentions influence model. This Mixed-intentions influence model accounts for observed

**Table 1 | Classification of models according to 3 categories**

| Model | Mentalizing | Bayesian | Mixed intentions |
|---|---|---|---|
| Influence model | + (coop and comp) | – | – |
| Mixed-intentions influence model | + | – | + |
| 1-ToM | + (coop and comp) | + | – |
| 1-ToM mixed intentions | + | + | + |
| Bayesian Sequence Learner (BSL) | – | + (depth 2 and 3) | – |
| RL | – | – | – |
| WSLS (Win/Stay - Lose/Switch) | – | – | – |

The first column indicates the ability of the model to mentalize, the second represents whether the model is a Bayesian model, and the third concerns models that could be used with a mixture of experts.

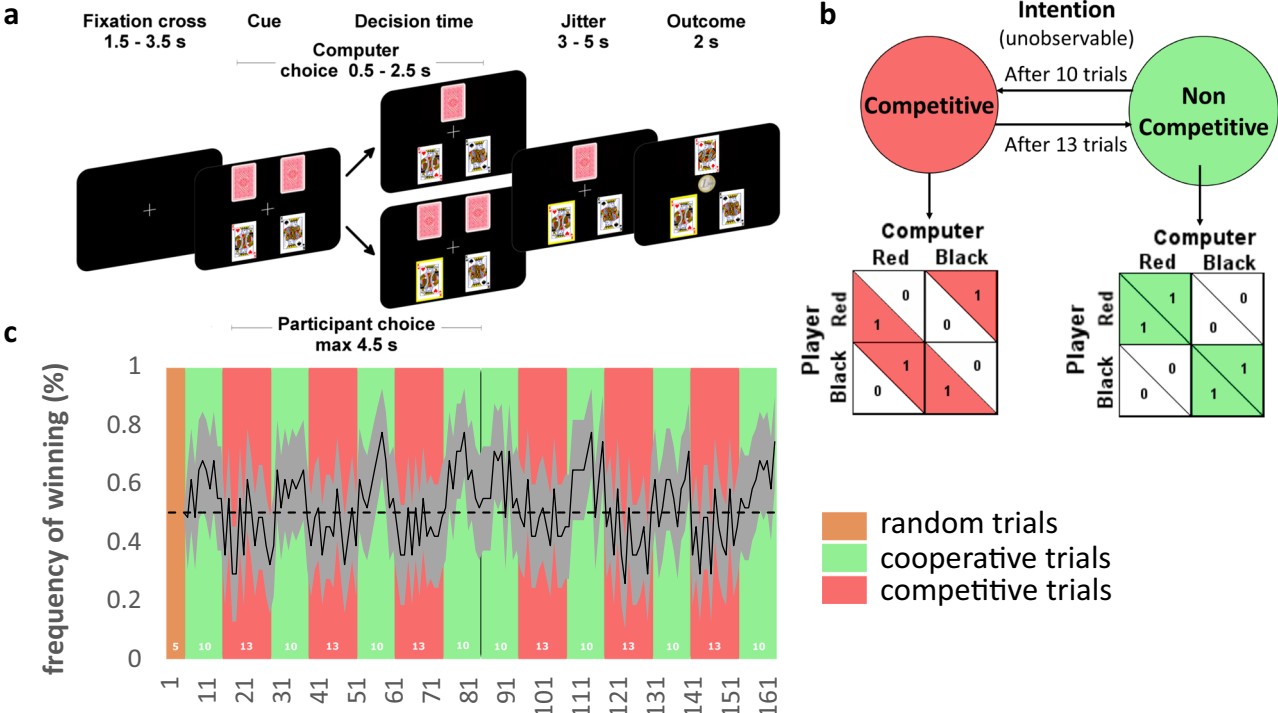

**Fig. 1 | fMRI experiment. a** After a fixation cross, four cards were presented on the screen. The two cards shown on top of the screen represent the cards presented to the opponent/partner (i.e., Artificial Agent), and not seen by the participant while the two kings (one black and one red) are the cards presented to the participant (shown in the bottom of the screen). The participants had to choose between these two cards. At the time of the decision, the upper screen represents the display if the AA makes its choice first, while the lower screen shows how one card is highlighted with yellow border if the participant makes his choice first. Then a screen presents the participant's and Artificial Agent's choices together. Finally, at the time of outcome the participant wins if both he/she chooses the same card as the AA (here red king). **b** Payoff matrix of the two types of block. Participant's payoff (bottom left of each small square) and the Artificial Agent payoff (top right of each small square). **c** Frequency of winning (black line) during Competitive (red background) and Cooperative (green background) blocks. The gray area represents the 95% confidence interval. The orange background represents 5 initial trials in which the AA was played randomly for initialization purposes.

behavior in which the other's goal is often only partially congruent with one's own, allowing us to explain a continuous range of behavior between pure cooperation and pure competition. At the brain system level, we show that computational signals (decision value, prediction error) from the two experts engage similar brain regions, due to the fact that these experts implement an identical influence learning process, each using different priors. Moreover, the ventromedial prefrontal cortex (vmPFC) and the ventral striatum track the reliability difference signal from the controller. When comparing trials classified as competitive versus cooperative by the controller, activity correlates positively with the reward prediction error (PE) signal more in the right temporo-parietal junction (rTPJ), dorsolateral prefrontal cortex (dlPFC), and the intra-parietal sulcus (IPS). In addition, when participants expect higher utility for choosing according to competitive rather than cooperative strategies, the vmPFC and the ventral striatum track the intentions of others and show changes in functional connectivity with the same brain system which discriminates reward PE between believed modes of interaction. Together, these results provide a model-based account of the neurocomputational mechanisms guiding human strategic decisions during games in which the intentions of others fluctuate between cooperation and competition.

## Results

### Behavioral signature of tracking intentions

In the Mixed-intentions task, participants were led to believe that they were interacting with another participant, in fact, they were interacting with an artificial agent (AA). Participants were asked to choose one of two cards to match the other participant's choice. Unbeknownst to the participant, the AA tried to match the two cards (coordination game) or to mismatch them (matching pennies game). To assess the degree

of participant's cooperation, we used their probability of staying on the same target as the previous trial. Indeed, there is one unique Nash equilibrium for the matching pennies which is to randomly switch target half the time, whereas less than 50% of switch signal cooperation. We assessed how participants used the history of previous interactions to make their choices to switch target. We used logistic regression to examine whether participants selected the same target as that from the previous trial ("Stay") or chose the other target ("Switch"), depending on whether the previous three trials (at t-1, t-2 and t-3) had been won or lost, whether the previous decisions had been to Stay or Switch, and whether the previous interactions from those trials indicated cooperation (Eq. 1). We also added sex, age and the rank of each trial (first, second or third trial etc. i.e., time indicator) as control variables. All trials except the first 5 trials were included in this analysis. Cooperation was defined as the binomial variable representing the interaction between the last action of the Artificial Agent (AA) and the participant's own previous outcome ("Cooperativity signature of AA"). This variable was set to 1 if either the participant had won on the previous trial and the AA stayed on the same target in the next trial, or if the participant lost on the previous trial and the AA switched to the other target in the next trial. Otherwise, the variable was set to 0. Indeed, if the AA is a cooperative partner, both players should choose to keep the same target after winning to be more predictable. Note that this definition needs to be distinguished from the degree of participant's cooperativity as defined by its probability to stay on the same target than on the previous trial. No multicollinearity was found between the regressors.

We found that the "Cooperativity signature of AA" predicted an increase in the "stay" probability of participants at t-1 and t-2 ($\chi^2(1) = 5.34$, Cooperativity signature of $AA_{t-1}$ : estimate = 0.05,    p = 0.021,

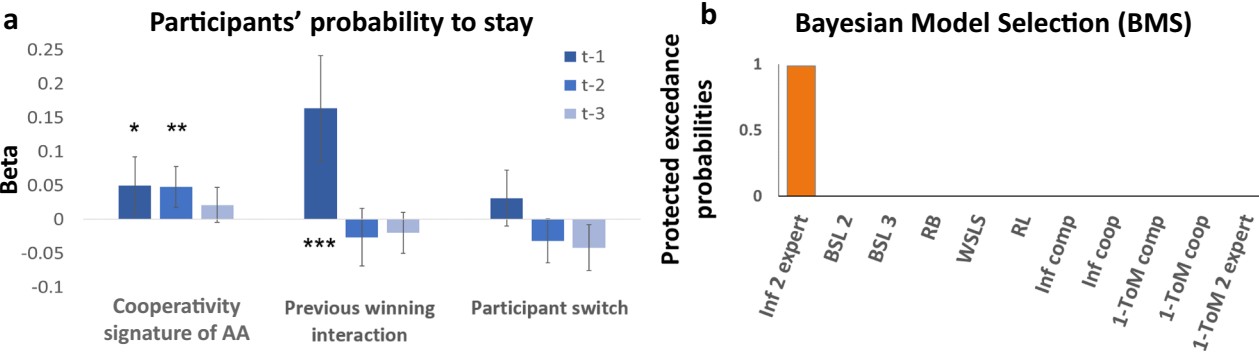

**Fig. 2 | Model-free behavioral analysis and models' comparison using Bayesian model selection. a** Model-free analysis. Random-effect logistic regression of the decision to stay after selecting a specific target with respect to the action of the artificial agent "Cooperativity signature of AA" (i.e., participant wins then AA – Artificial Agent – stays or participant loses then AA switches), the previous winning interaction (i.e., success or failure of past trials) and the choice to switch or stay, over the previous three trials. Bars represent the marginal effect in the percentage of each explicative variable on the probability to stay and the error bars represent the 95% confidence interval. *$p < 0.05$, **$p < 0.01$, ***$p < 0.001$ of a one-sided $\chi^2$, not corrected for multiple comparison. ($n = 31$ independent participants each making 158 time-dependent decisions). Concerning the Cooperativity Signature effect at t-

1, t-2, and t-3, the respective $p$ values were $p = 0.20$, $p = 0.002$ and $p = 0.101$. Similarly, for the previous winning interaction at t-1, t-2, and t-3, the respective p values were $p < 0.001$, $p = 0.225$, and $p = 0.200$. Finally, for the previous participant switch at t-1, t-2, and t-3, the respective p values were $p = 0.137$, $p = 0.057$, and $p = 0.016$. **b** Model comparisons based on Bayesian model selection. The protected exceedance probabilities indicate that the Mixed-intentions Influence model (Inf 2 expert) explains decisions in the mixed intention task better than others: Bayesian Sequence Learner (BSL), Heuristic models: Random Bias (RB), Win/Stay·Lose/Switch (WSLS), Reinforcement Learning (RL), influence model competitive, influence model cooperative, 1-ToM competitive, 1-ToM cooperative, 1-ToM 2 experts.

CI[0.01; 0.09]; $\chi^2(1) = 9.49$, Cooperativity signature of $AA_{t-2}$ : estimate $= 0.05$, $p = 0.002$, CI[0.02; 0.08]; Fig. 2a). This suggests the participants tracked whether the other agent was cooperating during the two previous trials (but not before). Participants used the outcome of the latest trial to make the next decision (staying or switching target) according to a win/stay, lose/switch strategy ($\chi^2(1) = 15.68$, winning$_{t-1}$ : estimate $= 0.16$, $p < 0.001$, CI[0.09; 0.24]; Fig. 2a). The lack of effect of the Cooperativity signature of AA at t-3 cannot be due to correlation with the regressors at t-1 or t-2, since a logistic regression model including only the regressors at t-3 (Previous winning interaction(t-3) (i.e. the boolean outcome of the previous interaction, 1 if it was a win, 0 if it was a loss), Switch(t-3), Cooperativity signature of AA(t-3)), showed that there was only an effect of Switch at (t-3) and not Previous winning interaction(t-3) and Cooperativity signature of AA(t-3). Additional behavioral data analyses showing evidence for separate cooperative and competitive experts are shown in Supplementary Note 1 and Supplementary Figs. 1-3.

## Computational models tracking intentions of the other agent

To elucidate the computations underlying strategic decision making, we compared the results of different computational models. These models were split into five classes (see Supp. Methods). The first class of models were based on heuristics and included Win-Stay/Lose-Switch and Random Bias models. The other four classes of algorithms can be classified into non-Bayesian versus Bayesian model families in one dimension and mentalizing versus non-mentalizing model families in the other dimension. Thus, the second class of models includes non-Bayesian, non-mentalizing models represented by reinforcement learning (RL) models. The third class represents non-Bayesian mentalizing models, namely the "influence models" which are RL models with an additional term representing how the actions of one player influence those of the other player. The fourth type corresponds to Bayesian non-mentalizing models, exemplified by the $k$-Bayesian Sequence Learner which tracks the probability that one target will be selected by the AA after a history of specific length $k$. The fifth class of models contains Bayesian mentalizing models, which are the $k$-ToM models using recursive Bayesian inferences of depth $k$ to predict the future choice of the AA. Each mentalizing model was tested using 3 versions: a competitive, a cooperative and a 'mixed intentions' version. The 'mixed intentions' version computes two separate decision values

according to competitive and cooperative experts, respectively, and arbitrates between them based on the difference in their respective reliability (see Fig. 3a and Supp. Methods). The two experts are running in parallel and there is no need for payoff matrices to be learnt. Indeed, they are hard coded in each expert and only the balance between the two experts has to be learned trial by trial. We defined reliability as the difference in unsigned value functions for two choices given by specific learning algorithms (See SI).

Next, we performed a group-level random-effect Bayesian model selection on the free energy computed by the model's estimation, taking into account potential outliers and the number of free parameters[42,43]. We found that the 'mixed intentions influence' model was most frequently the best fit across the population (pEP=0.98) (Fig. 2b), demonstrating that participants employed mentalizing-related computations in our mixed intentions task. This finding also indicates that arbitration between a cooperative and a competitive expert best explains observed participants' behavior, rather than either expert taken individually. To check the validity of the competitive/cooperative behavioral signature, we performed two additional analyses and found that each signature could be recovered by its corresponding expert model, but not by the other (Supplementary Fig. 6). These analyses consisted of logistic panel data regressions, (one for the cooperative model alone and one for the competitive model alone), clustered by simulation on the "Stay" strategy. To do this, we first generated a total of 310 datasets with the influence competitive model playing against the same sequences of AA's choices as the real players. To avoid contingencies between AA choices and model choices, we used the sequence of AA choices generated against real participants as a non-contingent opponent for the model which generates new data. We next generated data with the influence cooperative model using the same method. These logistic regression analyses included only a constant as predictor variable to directly compare the probability to stay independently of other variables, on behavior simulated by the competitive expert alone, or the cooperative expert alone. No multicollinearity was found between the regressors in the two analyses.

These two analyses showed that each expert is able to reproduce its corresponding behavioral signature. That is, the probability to stay on the same target of the competitive expert was 0.508 ($\chi^2(1) = 10269$, Probability to stay on the same target : estimate $= 0.508$, $p = 0.116$,

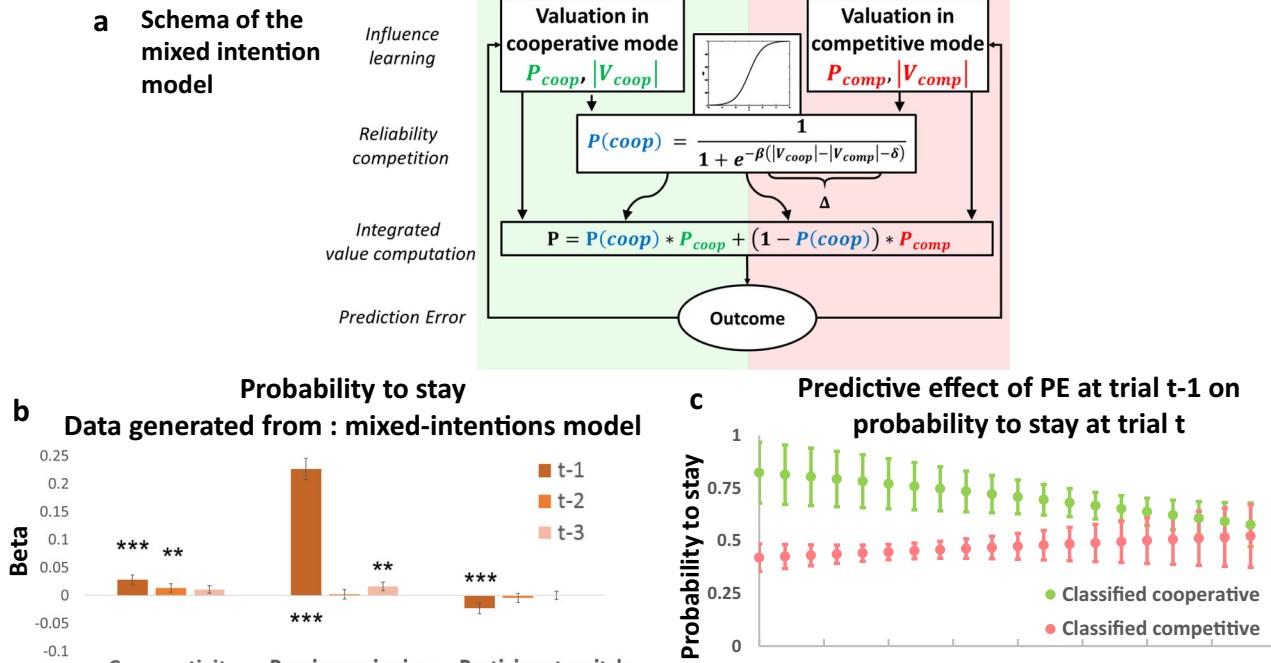

**Fig. 3 | Schema of the mixed-intentions Influence model, model-free generative analysis, and predictive effect of the prediction error on the probability to stay on the same target. a** Scheme of the Mixed-intentions Influence model. Two influence models (one cooperative – left, blue– and the other competitive – right, yellow) compute a value for choosing one specific target. A controller uses the difference between the absolute value of the value of each expert (called reliability) to compute a probability that the other is cooperating. Then, the model weights the value of each expert according to the probability of being in cooperative and in competitive modes to produce a final decision value. Then it compares its predictions to the actual reward and computes a new value for each expert. **b** Model-free generative analysis. We generated $n = 310$ sets of independent data using a free parameter from a normal distribution with mean and standard deviation calculated from the models fitted to the population, against the fixed sequences of choices that the artificial agent made against participants during the experiment. We regressed the behavioral decision to stay after selection of a specific target on the previous trial depending on the interaction of the previous outcome and the action of the artificial agent ("Cooperativity signature of AA"), the success or failure of up to three previous trials, and the action to switch or stay of the participant. Bars represent the marginal effect in the percentage of each explicative variable on the probability of stay. The error bars are the 95% confidence interval (random-effect logistic regression, $\chi^2$). For the cooperativity signature of the AA at t-1, t-2, and t-3, the p-values are respectively $p < 0.001$, $p = 0.003$ and p = 0.070. For the previous winning interaction at t-1, t-2, and t-3, the p-values are $p < 0.001$, $p = 0.649$, and $p < 0.001$, respectively, and finally, for the previous participant switch at t-1, the p-value is p < 0.001. **c** Marginal effect (in percentage) of the prediction error at trial t-1 on the probability to stay (at trial t) on the same target in trials classified as cooperative (at trial t, green dots) and trials classified as competitive (at trial t, red dots) (e.g. for a -1 prediction error, the probability to stay in a trial classified as cooperative increases by 82%). Error bars are the 95% confidence interval. *p < 0.05, **p < 0.01, ***p < 0.001 (random-effect logistic regression).

CI[0.499; 0.518]), whereas against the same sequences of AA choices the cooperative expert stayed on the same target significantly more frequently with a probability of 0.521 ($\chi^2(1) = 10935$, Probability to stay on the same target : *estimate* = 0.521, $p < 0.001$, CI[0.512; 0.530]). Thus, the cooperative expert cannot recover the competitive signature accurately and the competitive expert cannot recover the cooperative signature.

Additionally, only the Mixed-intentions Influence model (and not the cooperative or the competitive one) successfully reproduced the effect of the Cooperativity signature of AA on the probability to stay, as in the participants (Fig. 3b and Supplementary Fig. 6, and Supp. Note 1). We conducted a logistic regression to understand how the Mixed-intentions Influence model explained differences in behavioral strategy to stay or switch target. This analysis included the reward prediction error at t-1, the valence of the arbitration between cooperative and competitive intentions at time t (sign($\Delta$); 1 for cooperative and -1 for competitive), and the interaction between these two variables. No multicollinearity was found between the regressors in this analysis. This analysis revealed a main effect of the valence (cooperative or competitive) of the arbitration ($\chi^2(1) = 23.32$, *valence of the arbitration$_t$* : *estimate* = 0.239, p<0.001, CI[0.142; 0.0.336] Fig. 3c) indicating that participants tended to stay more on the same target when they attributed cooperative intentions to the other. Moreover, when we

tested how previous outcome might be used to make a decision according to the currently more reliable expert, we found an interaction effect, i.e. participants integrated the prediction error in their strategy differently depending on the attributed intention ($\chi^2(1) = 5.98$, *valence of the arbitration$_t$ * rPE$_{t-1}$ : estimate* = 0.185, p = 0.0145, CI[0.037; 0.333] Fig. 3c). That is, large negative prediction errors increased the probability that the participant would stay on the same target when the controller attributed cooperative intentions compared to when it attributed competitive intentions. In addition, we also performed another logistic regression analysis using the same variables and the actual mode of interaction (i.e., Competitive block trials versus Cooperative block trials). Again, no multicollinearity was found between the regressors in this analysis. We did not find the same interaction effect when we compared actual Competitive and Cooperative block trials ($\chi^2(1) = 16.32$, *Block type$_t$ * rPE$_{t-1}$ : estimate* = 0.007, p = 0.8394, CI [−0.056; 0.069], $\chi^2(1) = 1.44$, *Block type$_t$ : estimate* = 0.03, p = 0.229, CI[−0.07; 0.017] *and* $\chi^2(1) = 5.98$, *PE$_{t-1}$ : estimate* = 0.14, $p < 0.001$, Supplementary Fig. 7, see Supp. Note 1), showing that the classified intentions, rather than Competitive vs. Cooperative blocks, affected the use of prediction error. Finally, we conducted supplementary analyses to check the robustness of our Mixed-intentions influence model (See Supp. Note 1 and Parameter recovery matrix and confusion matrix in Supplementary Figs. 4 and 5).

We reasoned that when facing an individual who can change his/her intentions to compete or cooperate over time, the brain may implement distinct experts to compute the best choice based on these two possible intentions (i.e., cooperative or competitive) weighted by their relative reliabilities. We, therefore, built such an 'arbitrator' computation as a sigmoid function of the difference in reliability between the cooperative and competitive interactions (Δ), with an added bias (δ) that characterized each individual's tendency to attribute competitive (δ > 0) or cooperative (δ < 0) intentions to others. To assess the intentions of the other, participants only have access to the outcomes of previous interactions, the choice (to stay or switch) of the artificial agent in previous trials, and the interaction between these two types of information.

We hypothesized that repeated successes in a social context should favor the attribution of cooperative intentions because a series of victories suggests that both players are satisfied with the outcome. In such situations, the other player (i.e. AA) would become more predictable, which is an important feature to build cooperation[44]. Moreover, the interaction between the outcome and the AA's choice (i.e., the tendency of the AA to "stay" after a participant wins or "switch" after a participant loses) should drive the arbitrator to favor the cooperative mode, because playing the same winning target for both players corresponds to the pure-strategy Nash equilibrium of the Cooperative game. To test this hypothesis, we regressed the signed difference in reliability on (1) the participant's last outcome, (2) AA's choice to "stay" or to "switch" and (3) the interaction between the

participant's outcome and the AA's choice to stay or switch (Cooperativity signature of AA) for up to three retrospective trials. We found that the past two interactions between participant's outcome and AA's action (Cooperativity signature of AA), the last outcome, and switches by the AA at trial t-2 and t-3 explained the difference in reliability ($\chi^2(1) = 33.64$, *Cooperativity signature$_{t-1}$* : *estimate* = 0.59, p < 0.001, CI[0.393; 0.794]; $\chi^2(1) = 4.20$, *Cooperativity signature$_{t-2}$* : *estimate* = 0.06, p = 0.040, CI[0.003; 0.125]; $\chi^2(1) = 182.79$, *Victory$_{t-1}$* : *estimate* = 1.99, *p* < 0.001; $\chi^2(1) = 7.23$, *switch$_{t-3}$* : *estimate* = −0.097, *p* = 0.007), Fig. 4a). No multicollinearity was found between the regressors in this analysis.

Together, these analyses show that participants' behavior, when alternating between unsignaled cooperative and competitive blocks, is best explained by the Mixed-intentions Influence model. According to these findings, people use mentalization to update their beliefs about future chosen targets, and dynamically arbitrate between the predicted intentions of the other agent to compete or cooperate (Fig. 3a). Having characterized the computations of the dynamic adaptation to the changing intentions of others, we next tested where latent variables of these computations are encoded at the brain system level.

### Model-based fMRI analyses

First, we investigated whether the brain systems engaged by the two experts were similar regarding the decision value (DV) and reward prediction error (PE) of each expert. To do this, for the mixed-intention influence model, we built a GLM (GLM0) including, at the first level,

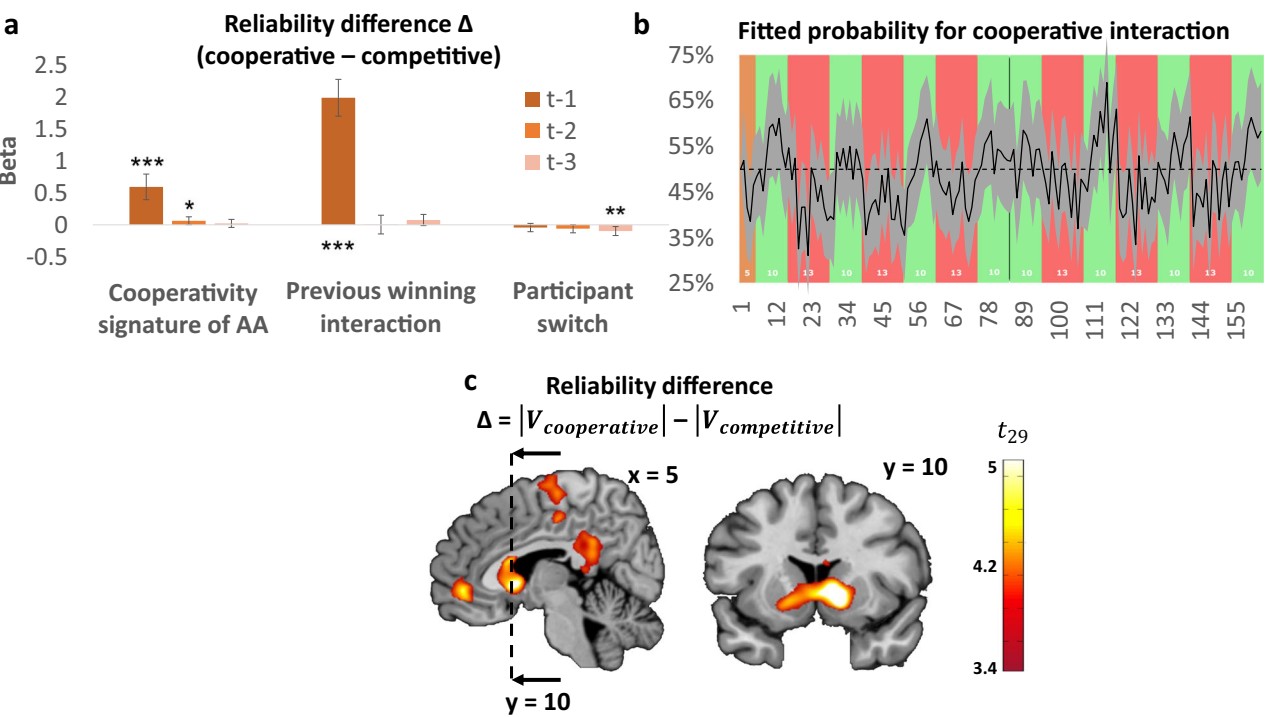

**Fig. 4 | Reliability difference from the controller as a function of behavioral signatures, predicted probability to cooperate computed by the model and BOLD signal correlating with the reliability difference of the Mixed-intentions Influence model. a** Difference in reliability is influenced by the Cooperativity signature of the Artificial Agent (AA), specifically the interaction of the previous participant's outcome followed by the action of the artificial agent (Participant wins then AA stays and participant loses then AA switches), the latest outcome and the computer's switch at trial t-2 and t-3. Bars represent the marginal effect in percentage of each explicative variable on the probability to stay. Error bars are the 95% confidence interval. *p < 0.05, **p < 0.01, ***p < 0.001 of a $\chi^2$, not corrected for multiple comparison. (n = 31 independent participants each making 158 time-dependent decisions). Concerning the Cooperativity Signature effect at t-1, t-2, and

t-3, the respective p values were p < 0.001, p = 0.040, and p = 0.466. Similarly for the previous winning interaction at t-1, t-2, and t-3, p < 0.001, p = 0.962, and p = 0.094. Finally, for the previous participant switch at t-1, t-2, and t-3, p = 0.198, p = 0.078, and p = 0.007. **b** Mean probability of the participants attempting to cooperate across all participants (black line) for the 163 trials computed by the Mixed-intentions influence model. The initial orange area is the 5 random initializing trials, green areas are the Cooperative blocks and red areas are the Competitive blocks. The gray area is the 95% confidence interval. **c** BOLD signal in ventral striatum, mPFC, and posterior cingulate cortex (PCC) is correlated with the difference in reliability, Δ, of estimated competitive and cooperative intentions (p < 0.05 whole-brain family-wise error).

decision values (DV) from the competitive expert, decision value from the cooperative expert, PE from the competitive expert and PE from the cooperative expert. Specifically, there were 4 onsets including the time of the cue presentation (cards on screen), participant's button press, AA's choice and feedback time. At the time of the cue onset, the parametric regressors were the DV for staying on the same target generated by the competitive expert, and the DV for staying on the same target generated by the cooperative expert. Similarly, at the time of feedback, the regressors were the PE generated by the competitive expert and the PE generated by the cooperative expert. We performed an ANOVA including DV and PE of each expert. Activity in the ventral striatum for the DV of staying on the same target was commonly observed for the two experts (P = 0.007 Family-Wise Error cluster corrected, initial cluster forming threshold of $p < 0.001$, Supplementary Fig. 8a). When directly comparing the DV of the two experts, we did not find any separate brain region (P > 0.95 FWE cluster corrected, initial cluster forming threshold of $p < 0.001$). For the reward prediction error (PE), the two experts showed common activity in the ventral putamen (P < 0.001), the anterior medial PFC (P < 0.001), posterior cingulate cortex ($p = 0.035$ and the lateral OFC (P < 0.001) (Supplementary Fig. 8b) (all Ps are FWE cluster corrected, initial cluster forming threshold of $p < 0.001$).

Next, we constructed a GLM (GLM1) to identify brain regions tracking the arbitration process (i.e., Δ: signed reliability difference, reliability for cooperation minus that of competition) between the two

experts (one for cooperation, the other for competition). We added the reliability difference Δ as parametric regressor at the decision stage, as well as the previous winning interaction (i.e. the boolean outcome of the previous interaction, 1 if it was a win, 0 if it was a loss), the previous switch and Cooperativity signature of AA of the previous trial, as non-orthogonalized parametric regressors to allow them to compete for the variance. At the outcome phase, we added the reward prediction error as a parametric regressor and controlled for the effect of the other's intention by adding Δ as a non-orthogonalized regressor. The bilateral ventral striatum (x,y,z = 14,12,−2, P < 0.001 and x,y,z = −13,7,−6, P < 0.001), and vmPFC (x,y,z = 6,46,−6, P = 0.001) tracked the difference in reliability between experts (Δ) at the decision time (all Ps are FWE cluster corrected, initial cluster forming threshold of p < 0.001, Fig. 4c and Supplementary Table 2), and this cannot be explained by the previous outcome alone or Cooperativity signature of AA. Bilateral dorsal striatum (DS; x,y,z = 17,6,−12, P < 0.001 and −14,3,−11, P < 0.001), bilateral orbitofrontal cortex (OFC; x,y,z = 44,36,−14, P = 0.003 and −44 52 8, P < 0.001), posterior cingulate cortex (PCC; x,y,z = 2,−35,38, P = 0.010), and bilateral angular gyrus (x,y,z = 45,−30,47, P < 0.001 and −54,−62,39. P = 0.005) encoded the reward prediction error at the outcome time (all Ps are FWE cluster corrected, initial cluster forming threshold of p < 0.001, Fig. 5a and Supplementary Table 3).

To identify the brain areas encoding the reward prediction error more robustly when the competitive expert was deemed more reliable

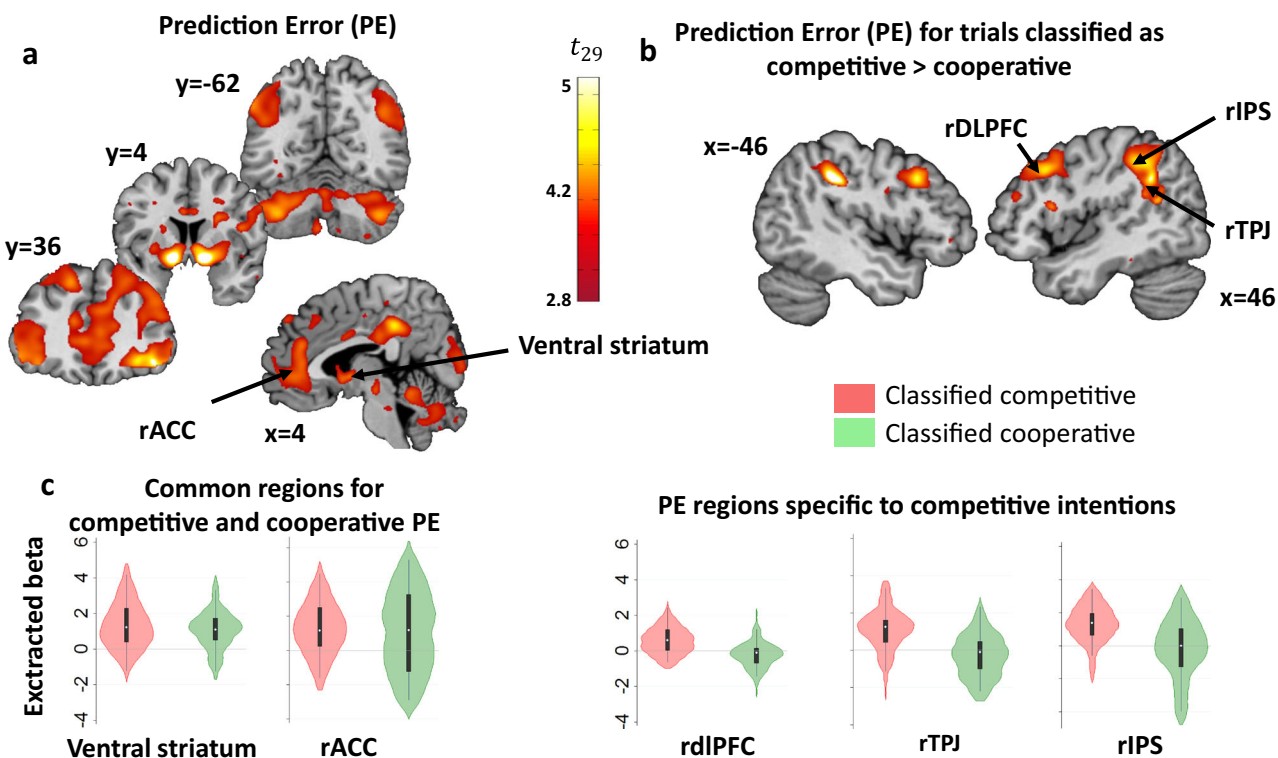

**Fig. 5 | Common regions for competitive and cooperative Prediction Error and brain system encoding Prediction Error more in competitive versus cooperative classified trials. a** Brain regions in which BOLD signal correlates with prediction errors for trials classified by the controller to be either competitive or cooperative. **b** Brain regions in which BOLD activity correlates more with PE on trials estimated to be competitive compared to trials estimated to be cooperative. This network comprised dlPFC (x,y,z = 30,9,42), IPS (x,y,z = 50,−50,32) and the rTPJ (x,y,z = 51,−50,33, p < 0.05 whole-brain family-wise error). **c** Beta value of PE correlation extracted at the outcome of trials estimated to be either competitive or cooperative. Left: regions in the ventral striatum (left x,y,z = −14,3,−11 + right x,y,z = 17,6,−12), and rACC (x,y,z = 6,42,−3) with increased correlation with PE in

trials estimated to be either competitive or cooperative. Right: specific brain regions correlating with PE when trials were classified as competitive as compared to cooperative: dlPFC (x,y,z = 30,9,42), IPS (x,y,z = 42,−47,42) and rTPJ (x,y,z = 51,−50,33) from 8 mm spheres centered on peak activation. N = 31 independent participants each making 158 dependent decisions. Black box plots represent the interquartile range (bold line), and the range of the data with center = median and whiskers = the most extreme data point that does not exceed 1.5 interquartile range. The colored shapes (red or green violin plots for trials classified as competitive or cooperative, respectively) represent the kernel probability density of the data.

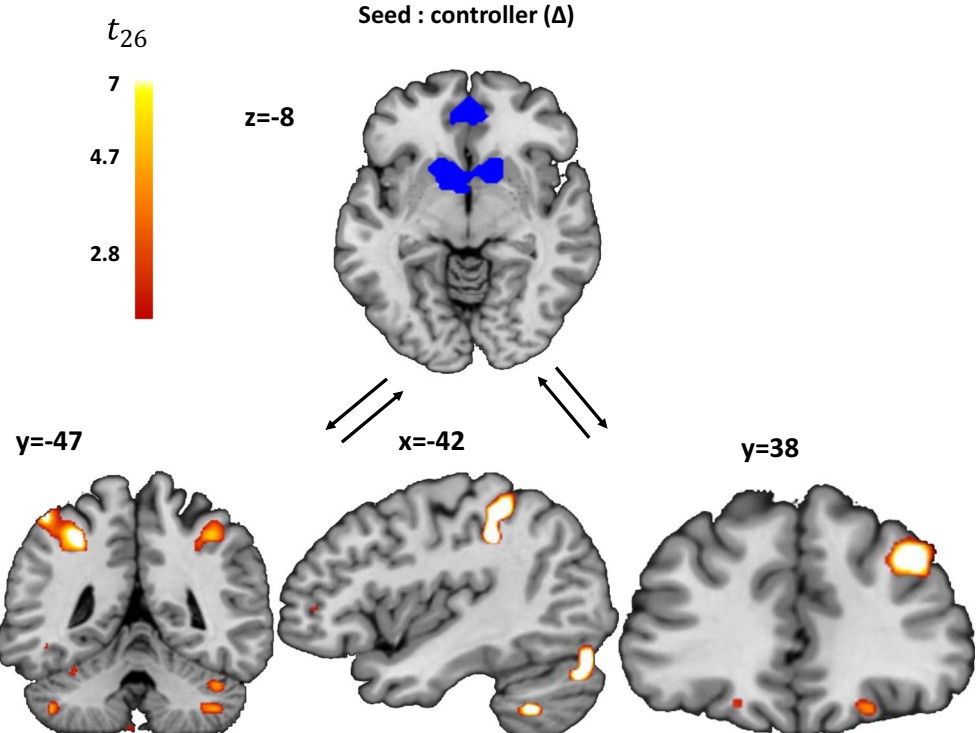

**Fig. 6 | Higher functional coupling during trials classified as competitive compared to those classified as cooperative, between 'reliability difference' brain regions (mPFC and ventral striatum) from the controller and regions encoding more the prediction error in these classified trials.** For the connectivity analysis, the BOLD signal was extracted from seed regions (mPFC and ventral striatum using GLM1) computing the reliability difference between cooperative and competitive intentions of others (in Blue). The psychophysiological interaction effect shows higher functional coupling (voxels in red) with the left TPJ (x,y,z = −42,−40,50), left IPS (x,y,z = −48;−44;58) and right dlPFC (x,y,z = 38, 34, 34, $p < 0.05$ FWE threshold at $p < 0.001$) in trials classified as competitive as compared to those classified as cooperative.

than the cooperative one, we tested another GLM (GLM2). Trial onsets were separated according to whether the value of the signed reliability difference Δ was positive or negative. If this value was positive, the trial was classified as cooperative, and competitive otherwise. The computed decision value for staying on the same target was used as a parametric regressor at the time of choice and the reward prediction error computed by the Mixed-intentions influence model was used as parametric regressor at the time of outcome. We found that the right dlPFC (x,y,z = 35,11,36, $P = 0.003$), the IPS region (x,y,z = 50, −50, 32, $P = 0.001$) and the right temporoparietal junction (rTPJ; x,y,z = 51,−50,33, $P = 0.001$ FWE cluster corrected) more stongly encoded reward prediction error in trials classified as competitive versus cooperative ($p < 0.05$, FWE, Fig. 5b, c and Supplementary Table 4). This effect could not be explained by smaller variance in the PE regressor in trials classified as competitive trials compared to those classified as cooperative, since we observed no difference in regressor variance on these two types of trials ($p = 0.57$, Levene's test). No areas showed stronger PE signals during the trials classified as cooperative compared to competitive. It should be noted that this differential PE coding reflects the classification of the current trial as cooperative vs. competitive (fMRI results, Fig. 5b). In contrast, the behavioral results in Fig. 3c show that the effect of rPE on behavior depends on how the next trial might be classified.

### Connectivity analysis
Finally, we performed a generalized psycho-physiological interaction (gPPI) seed-to-voxel connectivity analysis to understand the interactions between brain regions tracking the arbitration process (i.e. Δ: reliability difference) for the cooperative and competitive experts and those regions more engaged with the PE when the controller attributes competitive rather than cooperative intentions to the AA (see Methods). We used the ventral striatum and vmPFC, which encoded the controller, as a combined ROI, as the seed regions (ROI extracted from the GLM1 striatal and vmPFC activity) for trials classified as competitive compared to those classified as cooperative (i.e. trials for Δ < 0 or Δ > 0) at the decision time. We found stronger effective connectivity between regions encoding the difference in reliability and the right dlPFC (x,y,z = 38,34,34), the left IPS region (x,y,z = −48;−44;58) and the left TPJ (x,y,z = −42,−40,50, $p < 0.05$ FWE; Fig. 6 and Supplementary Table 5) at the decision time for trials classified as competitive compared to those classified as cooperative. This result indicates that the dlPFC, IPS and left TPJ receive an input signal at the time of choice according to the difference in reliability with respect to the anticipated competitive versus cooperative intention of others. This brain network largely overlaps with the brain network differentiating PE encoding between trials classified as competitive versus cooperative at the outcome (Fig. 5b). This difference in connectivity could not be due to a general higher coupling during classified competitive trials compared to classified cooperative trials. Indeed, we tested for a difference in global efficiency between classified competitive and classified cooperative trials using graph theory and found no differences between the two modes of interaction (Global efficiency: T(26) = 0.75; $p_{uncorrected}$ = 0.46). Moreover, we also tested for differences between modes of interaction (competitive and cooperative) in local efficiency, and found that no node exhibited different local efficiency between trials classified as competitive and those classified as cooperative.

### Discussion
To make a strategic decision when facing an individual with unknown and fluctuating intentions, it is necessary to make inferences as to

whether we are in a competitive or cooperative situation. In the context of minimal information, for example when only the past choices of the other are available, but not their outcomes, such inferences are much more difficult than when the other's intention is known (e.g., in a competitive game)[19]. Here, we provide evidence that the brain engages in dynamic tracking of another individual's intentions, despite having no explicit information regarding whether the situation is cooperative or competitive. We found that strategies of participants were mostly affected by the outcomes of previous interactions and by a "signature" of the other's cooperativity, i.e., the tendency of the other (here the Artificial Agent, AA) to stay on the same target after the participant wins. Comparison between computational models demonstrated that such behavior is best explained by a model in which choice is driven by a controller that tracks the reliability difference between cooperative and competitive intentions. The fMRI results show that the neural computations of this controller are implemented in the ventral striatum and the vmPFC. Thus, both behavior and brain imaging results can be accounted for by a model that includes a controller that allocates weights according to different experts' predictions. At the time of outcome, a brain network, including the rostral anterior cingulate cortex (rACC), ventral striatum and lateral OFC encoded prediction error similarly in trials classified as competitive or cooperative. However, prediction error signals in some brain areas depended on the classification of the current trial as cooperative or competitive as classified by the controller. That is, a distinct brain network, composed of the bilateral dlPFC, bilateral IPS regions and the rTPJ was more engaged for trials classified as competitive compared to those classified as cooperative. This latter brain network reflects a differential use of the outcome of the social interaction as a function of whether it is classified as competitive or cooperative (Fig. 2d).

Mentalizing processes are essential to correctly infer the strategy of others. This is true in the Cooperative context, in which participants performed above chance, reflecting their ability to effectively infer the other's (i.e., AA's) behavior. In the Competitive context, participants performed below the chance level, showing that the AA was able to predict their behavior and exploit their previous choices/outcomes. The Mixed-intentions Influence model was most consistent with the data and generated behavior similar to that of the participants. Each expert model is an expanded RL model, with a term accounting for one's previous choice influencing the choice of the other. Although only the influence term differed between the competitive and cooperative models, the Mixed-intentions Influence model tracked intentions based on this second-order mentalizing term by weighting the contribution of cooperative and competitive experts. One key aspect of this Mixed-intentions Influence model is that it captures higher order structures (fluctuations between cooperation and competition) during social interactions. In contrast, an important limitation of the classical RL model is that it does not exploit higher-order structures such as interdependencies between different stimuli, actions, and subsequent rewards. Previous studies demonstrated that models incorporating such structures can account for individual decision making in different situations[45–48]. Here, we showed that the representation of abstract states, such as whether the other is cooperating or competing, can be extended to social decisions and underlies the ability to build strategies. To confirm that the Mixed-intentions Influence model accounted more for neural activity in brain areas involved in social interactions, we formally compared the brain regions covarying more with the decision value for staying on the same target, as computed by the winning model, compared to the decision value for staying on the same target, computed by a simple RL model (Supplementary Fig. 9b). One crucial difference between a simple RL model and the Mixed-intentions Influence model is that in the former, only the value of the chosen option is updated and the valuation of the option that was not chosen does not change. In the latter, both the

values of the chosen and unchosen options are updated to incorporate the knowledge that the current state has a given reliability to be cooperative or competitive. The controller weights the valuation produced according to the competitive or cooperative hypothesis using a sigmoid function of the difference in reliability between the two experts.

Activity in the ventral striatum and vmPFC increased as the prediction of cooperation from the controller became more reliable than the competitive prediction. These brain regions dynamically track the difference in reliability between intentions classified as cooperative and competitive in a situation when the nature of the social interactions is unknown. Previous reports demonstrated a role of the ventral striatum when making cooperative choices alone, in response to a partner's cooperative choice in an explicit cooperation task[49] and also in the attribution of intentions in a competitive context[29]. Our findings show that strategic social behavior can be explained by a Controller (i.e. mixture of experts) according to which cooperative/competitive social behavior results from the interaction of multiple systems, each proposing possible strategies for action[25,27,28]. The ventral striatum was involved in computing both the reliability difference between the competitive and cooperative experts at the time of choice (Fig. 4c) and the prediction error at the outcome (Fig. 5a). This ventral striatum engagement could reflect that it anticipates more social victories at the time of choice, when one is more likely to be in a cooperative rather than a competitive context, and that it also encodes unexpected social victories. These findings indicate that the reliability difference and the PE signals are both used to adapt to fluctuating intentions of others during different interactional contexts (Fig. 3a). This is consistent with previous studies reporting that the ventral striatum is engaged with mutual cooperation[49] and with the fact that this region is reliably active in relation to others' rewards and contains cells that link own rewards to self or others' actions[50].

One strength of our computational approach was to assess and compare a large variety of competing models, including Bayesian Sequence learners, Reinforcement Learning, Heuristic models, recursive learning model 1-TOM, influence models for only cooperative strategies or competitive strategies and a mixture of experts either using influence models (Mixed-intentions influence model) or using 1-ToM (1-ToM mixed intentions). Many have never previously been directly tested against each other. Other models (active inference, fictitious learning and Hierarchical Gaussian Filter) were also tested but did not pass the model recovery analyses, and are only reported in the Supp. Methods for completeness. Our results agree with studies concluding that social learning may be driven by reinforcement processes that include a mentalizing term[3,6,8,29,51]. We demonstrate that when a task is not explicitly signaled as cooperative or competitive, this evokes the arbitration between strategies determined by predictive reliability. Behavior is hence controlled by giving a higher weight to the strategy with the most reliable prediction[28]. Note that in our setting the only information that can be integrated by participants is their own choices, rewards and the history of the choices made by the other (i.e., AA). The structure of the task was not directly observable because the nature of the social interaction (i.e cooperative or competitive) was never explicitly signaled to participants, and the rewards of the other were not observed. The resulting complexity and uncertainty might reduce the benefit of model-based strategies, as suggested recently[52,53], and might increase the use of simpler trial and error learning. This contrasts with previous neuroimaging studies that investigated learning of social interactions in either competitive or cooperative situations alone (matching pennies or rock-paper-scissors games against computerized opponents)[24,54]. Our findings also broadly agree with a cognitive hierarchy of strategic learning mechanisms, proposing that distinct levels of strategic thinking correspond to different levels of sophistication of learning mechanisms[55]. However, we propose a more general model based on a mixture of influence

learning experts that function in parallel and are then compared with respect to their relative reliability.

Competitive social interactions often emerge in situations where an agent's outcome depends on the choices of others, which requires the ability to infer the intentions of others[6]. In the context of our mixed intentions task, when participants make a choice driven by the competitive expert, which is more reliable than the cooperative expert, the PE signal is more robustly encoded in the dlPFC and rTPJ/IPS. Thus, priors on the context of interaction (competitive or cooperative) during the decision time, modulate the way the reward prediction error will be implemented at the time of outcome. Although this brain network has previously been reported when inferring the intentions of others[8,38,56], the strength of our computational account of theory of mind processes is to specify that this brain system computes a PE differently for trials that the controller classifies as competitive versus cooperative. This reflects a differentiation of PE signals in the implementation and use of the outcome of the social interaction, as a function of the classified interaction (Fig. 3c). Nevertheless, this finding does not indicate that each expert is encoded in separate brain regions. Indeed, computational signals (decision value and PE) from the two experts engaged common brain regions (see results and Supplementary Fig. 8). The fact that we did not observe that each expert is encoded in separate brain regions may be seen as a limitation, since the behavioral model assumes two 'experts' learning in parallel. However, the algorithmic and the brain system levels have to be distinguished[57]. Separate expert systems can be observed at the behavioral and computational levels for cooperation and competition, but it does not necessarily follow that separate brain systems must be assumed for these experts. Indeed, the two experts, each using different priors, implement the same cognitive process (i.e., influence learning). This is a different situation than previous mixture of models theories which weigh experts implementing different cognitive processes[28]. This is a key conceptual distinction which extends the mixture of models to a more general view in which the experts are differentiated not by their cognitive processes but by their priors.

It should be noted that PE was not more volatile in trials when the competitive expert was more reliable than the cooperative expert. This rules out the possibility that the observed difference in PE encoding reflects higher PE volatility in competitive contexts. When comparing intentions classified as cooperative compared to competitive, participants tended to be more predictable, staying more on the same target after experiencing an unexpected social defeat (i.e., after higher negative PE) (Fig. 3c). This behavior likely reflects a signal sent to the other to indicate one's willingness to stay on the same target, despite bearing the potential cost of staying on this target[6,8,58]. This is a key feature of successful coordination[44] in which agents who want to trigger reciprocity[49] are willing to incur a cost to promote cooperation from the other.

Finally, we found higher effective connectivity between seed regions encoding the reliability difference of the controller (vmPFC and striatum) and brain regions more engaged in PE for trials classified as competitive versus cooperative (dlPFC, TPJ) (Fig. 6). This indicates that brain regions engaged in the input of the arbitration process at the time of choice are more strongly coupled (respectively decoupled) with brain regions encoding PE for intentions classified as competitive (respectively for intentions classified as cooperative). This reflects a differential use of the outcome of the social interaction as a function of whether it is classified as competitive or cooperative. Thus, according to the intention attributed by the controller to the other, PE signals differed and the strength of effective coupling increased between regions encoding the reliability difference of the controller and the dlPFC-TPJ network. When one expert is more reliable than the other, the Mixed-intentions Influence model predicts that the PE is driven by the valuation of the more reliable expert. Since the only difference between experts is the sign of the second-order mentalizing term, this

suggests that the dlPFC-TPJ network at the outcome is more engaged when there is a need to mentalize intentions of other agents with opposing goals (i.e., intentions classified as competitive). Moreover, engagement of the dlPFC-TPJ network increases the probability of switching following a trial classified as competitive by the controller, allowing behavioral adaptation by virtue of the reliability difference signal.

In conclusion, our work provides evidence that the mixture of experts model explains behavior in socially volatile situations differing only by the reward function of other agents. These two experts only differ by their priors on how their reward function takes into account another agent's reward (i.e., the second order mentalizing term), and were sufficient to discriminate the others' intentions. These findings provide a mechanistic framework explaining the neurocomputations underlying learning in strategic social interactions. We extend to theory of mind processes (i.e., inferring cooperative vs competitive intentions and adapting to changes between these modes of interaction) a computational account similar to mixture of experts proposed to arbitrate between strategies in other domains, such as exploitation vs exploration[59,60], "model-based" vs "model-free" systems[25,27,52] and learning by imitation vs emulation[28]. Finally, our Mixed-intentions Influence model may be useful in the fields of computational developmental psychology and computational neuropsychiatry to identify how specific computational components of the theory of mind develop in healthy children and are modified in neurological disorders[61].

## Methods
### Participants
This study was approved by the National Ethics Committee (CPP Est II: 18/592, ANSM: 2018-A01135-50), and all participants gave their informed written consent. A total of 31 participants (aged 20–40, M = 27, SD = 5.1–17 women) were recruited via a daily local newspaper and the University of Lyon 1 mailing list. All participants were screened to exclude those with medical conditions including psychological or physical illnesses or a history of head injury to prevent having confounding variables. Sex was determined based on self-reporting. Information about gender has not been collected. Sex-based analyses were not performed because the n is relatively low to compare between men and women. No statistical method was used to pre-determine the sample size. No data were excluded from the analyses.

### Mixed intentions task
Participants performed a novel task comprising 163 trials in an MRI scanner. They were led to believe that they were interacting with another person via a computer interface, while in fact, they were playing against an artificial agent (AA) managed by a computer program. Such simulated social interactions allowed us to investigate the dynamics and neural mechanisms arbitrating between multiple learning algorithms. Participants were faced with a screen containing four cards, two face down (the other player's cards) and two face up (their own cards). Participants were informed that to win, they had to choose a card of the same color as the one the other person was going to choose. Experimenters were careful not to specify whether the other was an adversary or a partner. Participants were told that they and the other player had to make their choices in four seconds (Fig. 1a). If the Artificial Agent (AA) played before the participant, one of the two face-down cards was removed from the playing field. If the participant chose first, only the selected card remained on the playing field. Then, when both had chosen, the chosen cards were revealed and the participant received a reward if the card colors matched, otherwise they received nothing. Participants were led to believe that their final payoff would be increased by 10 c (euro) for each winning interaction. No information about the other's payoff was given to the participants, they only knew that after an interaction, the other 'participant' would

see the same screen but with their own outcome which could be different from the participant.

Importantly, unbeknownst to the participants, the artificial agent alternated between Competitive and Cooperative trial blocks. During this "mixed intentions" task, the AA's strategy was determined by alternating 13 trials of a matching pennies (MP) task (Competitive blocks), and 10 trials of a coordination game (Cooperative blocks). Importantly, if the regularity in the switches (every 10 or 13 trials) between modes was detected, we would expect a learning effect over time, reflecting higher success rate for late relative to early blocks, both for the Cooperative blocks and for the Competitive blocks. Logistic regression, however, indicated no statistically significant effect of the blocks' rank on the success rates in the Cooperative blocks (no significant rank effect, effect size on success rate = 0.14%, $p = 0.637$). Similarly, there was no significant learning effect over blocks for the Competitive blocks (no rank effect, effect size on success rate = 0.06%, $p = 0.779$). In addition, similar logistic regressions on the probability to switch indicated no statistically significant change in the switch strategy across Competitive blocks (size effect of Competitive block rank: 0.004, $p = 0.095$) or across Cooperative blocks (size effect of Cooperative block rank: 0.001, $p = 0.765$).

The artificial agent algorithm was designed to predict the color that would be chosen by the participant on the basis of a probabilistic analysis of the two previous choices and outcomes (see SI for the algorithm). Here we defined a competitive choice, made by the AA, as choosing the card of the color the participant was expected not to play and a cooperative choice as choosing the card with the same color. Thus, the artificial agent exploited the bias of the participants in a stochastic way, i.e. the more predictable the participant was, the more the algorithm made correct competitive or cooperative choices (see SI). Participants were not informed of the switches between the two blocks (Cooperative vs Competitive), however, their goal was always to choose the same color as that chosen by the other player (i.e. the AA).

The MP task is competitive, and the computer uses the record of the participant's choice and reward history to minimize the participant's payoff. Therefore, in this case the participant's optimal strategy during the MP task is to choose the two targets randomly across trials. During the coordination game, the AA tried to maximize the participant's payoff and in this case the participants should try to choose one of the two targets consistently so that the computer can choose the same target as them. Since the participant is not informed of either the goals of the AA or the switches between blocks, they must adjust their strategy based on recent experience and infer cooperation/competition on the basis of their observations.

This task was designed to identify key components of the adaptation to the other's intentions regarding whether others are cooperating or competing. We took advantage of the fact that an individual's estimates as to whether they are engaged in a cooperative or competitive interaction can be assessed even when the individual is interacting with a computer program rather than another person. Transitions between the Competitive and Cooperative blocks were unsignaled, therefore participants had to discover by trial and error the most successful strategy over consecutive blocks. This alternation between the two interaction modes functioned well because the participant's winning rate was significantly higher in Cooperative (mean 60% std 1%) than in Competitive (mean 44% std 1%) trials (two-tailed Wilcoxon sign rank test $p < 10^{-3}$ CI95 [0.10; 0.24]). Moreover, not only the winning rate (which could be driven by the other's change in strategy) shows that the design functions well, but also the change in switching strategies of participants. Indeed, the switching strategy of participants depended upon the mode (Cooperative or Competitive) of interaction on the previous trial (Mean difference =5.43%, $p = 0.036$), but participants also changed their switching strategy using the Cooperativity signature of AA depending on the mode (Cooperative or Competitive) of the Artificial agent of the previous trial (Mean

difference = 9.61%, $p = 0.0012$, marginal effect of panel data logistic regression clustered by participant of switch strategy on the mode of interaction, the Cooperativity signature of AA and their interactions, no multicollinearity was found between the regressors in this analysis). The cooperativity Signature of AA was defined as the binomial variable representing the interaction between the last action of the Artificial Agent (AA) and the participant's own previous outcome. That is, the Cooperativity signature of AA was set to 1 if either the participant had won on the previous trial and the AA stayed on the same target in the next trial, or if the participant lost on the previous trial and the AA switched to the other target in the next trial. Otherwise, the variable was set to 0. Indeed, if the AA is a cooperative partner, both players should choose to keep the same target after winning to be more predictable.

## Artificial agent

The AA calculated the probability p for the participant to select a particular target color based on the history of the two previous choices and their outcomes. Then, this prediction was exploited in a probabilistic fashion (see SI): in the Cooperative mode the AA chose the color card it predicted with probability p, while in the Competitive mode this color was chosen with probability 1-p.

## Behavioral analysis

For the logistic regressions, we reported significant marginal effect of a given variable under the name "*estimate*" (for example: *Cooperativity signature*$_{t-1}$ : *estimate*).

$$\text{Logistic regression} : ln\left(\frac{P}{1-P}\right) = x_0 + w_1X_1 + w_2X_2 + \dots \quad (1)$$

$X_i$ represents independent variable and $w_i$ represents the associated weight in the logistic regression. $P$ represent the probability of a given event. The marginal effect of the variable $X_1$ is defined as:

$$\hat{y}_1 = \text{mean}\left(\text{logit}^{-1}(w_1)\right) \quad (2)$$

The mean is computed across all observed data. Thus, the marginal effect called "*estimate*" can easily be interpreted as the discreet change of the dependent variable given a unitary change of an independent variable.

For the linear regressions, reported "*estimate*" represents $w_i$ i.e., the regression coefficient. Indeed, in a linear regression, marginal effect of a variable is equal to the estimated coefficient.

For both logistic and linear regression, we specified a data panel on Stata (Version 14.1) to account for the repeated measures within each participant's game. We then used a random-effects model with cluster–robust standard errors for panels nested within participants, allowing for intra-participant correlation. The optimization method was maximum likelihood estimation using Gauss–Hermite quadrature to approximate the likelihood. For the coefficient estimation, we used a robust estimator (sandwich method also called Eicker–Huber–White standard errors) to be robust for heteroscedasticity.

We assessed the distribution's normality and equality of variance where applicable. If these assumptions were met, we proceeded with a parametric test; otherwise, we opted for the non-parametric version of the test.

## Models

To test for a dynamic tracking of implicit intention, we compared 11 models with 6 involving theory of mind (*Inf, k-ToM*) and the remaining 5 to control for other possible strategies. The influence models (*Inf*) rely on Taylor expanded reinforcement learning[6] to take into account the influence of one's own strategy on the strategy of the other. *k-ToM* models also take into account the influence of one's own strategy on

the other but in a Bayesian fashion[13,34]. These two models were adapted in their cooperative and competitive versions. Moreover, we constructed an adaptation of these two models (*Inf*, *k-ToM*) in which an arbitrator weights the cooperative and competitive versions according to their reliability before making the decision. Finally, because k-ToM is a recursive model ("I think that you think that..."), we included k-ToM of depth one and two for each version.

To control for strategies that did not include theory of mind we added 5 other models including two Bayesian inference types (*HGF* and *BSL*). The Hierarchical Gaussian Filter (*HGF*)[62,63] basically tracks the external volatility of the artificial agent's choices in a Bayesian hierarchical way. The Bayesian Sequences Learner (*BSL*) strategy relies on Bayesian inference given past sequences of choices. In a model free analysis, we found that participants tended to use the past 2 choices to make their next choice, so we used sequences of depths 2 and 3. Finally, we added two non-Mentalizing non-Bayesian models, a reinforcement learning model (RL) and a model based on the heuristic Win-Stay / Lose-Switch that we observed in the model free analysis.

Models were individually fit using Variational Based method with the VBA toolbox. Every prior was set to their default values. With this method we were able to find free parameters that minimized the free energy of the model[64].

The Bayesian model selection (BMS) was performed using the VBA toolbox (Variational Bayesian Analysis) in a random effect analysis relying on the free energy as the lower bound of model evidence. We use protected Exceedance Probability measurements (pEP)[42] to select the model which was used most frequently in our population.

## fMRI data acquisition

MRI acquisitions were performed on a 3 Tesla scanner using EPI BOLD sequences and T1 sequences at high resolution. Scans were performed in a Siemens Magnetom Prisma scanner HealthCare at CERMEP Bron (single-shot EPI, TR / TE = 1600/30, flip angle 75°, multiband acquisition (accelerator factor of 2), in an ascending interleaved manner with slices interlaced 2.40 mm thickness, FOV = 210 mm. We also use the iPAT mode with an accelerator factor of 2 and the GRAPPA method reconstruction. The number of volumes acquired varied given the time the participant took to make their decisions. The first acquisition was made after stabilization of the signal (3 TR). Whole-brain high-resolution T1-weighted structural scans (0.8 ×0.8 ×0.8 mm) were acquired for each participant, co-registered with their mean EPI images and averaged across participants to permit anatomical localization of functional activations at the group level. Field map scans were acquired to obtain magnetization values that were used to correct for field inhomogeneity.

## fMRI data analysis

Image analysis was performed using SPM12 (Wellcome Department of Imaging Neuroscience, Institute of Neurology, London, UK, fil.ion.ucl. ac.uk/spm/software/spm12/). Time-series images were registered in a 3D space to minimize any effect that could result from participant head-motion. Once DICOMs were imported, functional scans were realigned to the first volume, corrected for slice timing and unwarped to correct for geometric distortions. Inhomogeneous distortions-related correction maps were created using the phase of non-EPI gradient echo images measured at two echo times (5.20 ms for the first echo and 7.66 ms for the second). Finally, in order to perform group and individual comparisons, they were co-registered with structural maps and spatially normalized into the standard Montreal Neurological Institute (MNI) atlas space using the DARTEL method. Then we ran ARTrepair to deweight scans that could include movement artefacts[65].

We ran general linear models (GLMs) analyses to identify which brain regions encoded: (a) one's belief that one is interacting in a cooperative or in a competitive situation (Δ); (b) the reward prediction error (PE) after interactions classified as cooperative or competitive; (c) the PE difference between the trials classified as cooperative vs competitive. In every GLM, an event was defined as a stick function. The participant's button press and the AA's selection of target were defined as onset of no interest in all GLMs. For all GLMs, missing trials were modeled with four events (cue, participant's button press, AA's choice and outcome) as separate onsets without additional parametric regressors. Head movement parameters were added as parametric regressors of no interest to account for motion-related noise. Because the behavioral analysis showed that the bias towards competitive interaction affects the strategy of participants, we added the competitive bias (δ) as a covariate at the second level analysis in all GLMs.

Specifically, in GLM1, there were 4 onsets, including the time of the cue presentation (cards on screen), participant's button press, AA's choice and the feedback time. Parametric regressors were the difference in reliability Δ, the Cooperativity signature of AA(t-1), the previous winning interaction (t-1) (i.e. the boolean outcome of the previous interaction, 1 if it was a win, 0 if it was a loss) and the participant switch(t-1) at the time of the cue onset. At the outcome phase, parametric regressors were the reward prediction error (PE) as well as Δ to control for the effect of the believed intention of the other on the PE brain encoding.

In a second GLM (GLM2), we separated trials given the sign of Δ - δ (positive or negative) to identify brain regions specifically engaged in cooperative or competitive mental states (δ is a free parameter capturing the participant's bias toward competitive intent). Δ refers to the difference in reliability of cooperative and competitive prediction and δ is the competitive bias. For this GLM, there were 6 onsets, including the cue for trials classified by the winning Mixed-intentions influence model as cooperative or competitive, participant's button press, AA's choice and the feedback time for trials classified by the Mixed-intentions influence model as cooperative or competitive. Trials were classified by the Mixed-intentions influence model as either cooperative or competitive and parametric modulators were: the difference in reliability Δ and the decision value for staying on the same target at the time of the cue and the PE and Δ at the time of feedback. Three participants who always attributed the same intention to the AA were not included in GLM2. The mean number of trials classified as competitive by participants was 81.4 ( ± 3.75 *SE*). This represents approximately half of the trials (total number of trials = 163).

To test the additional hypothesis that brain activation observed for belief in other's intentions (Fig. 4c) is also present in Competitive vs Cooperative blocks, we conducted two more GLMs. One of them, GLM3, is similar to GLM2, i.e., we separated trials into two categories (in Cooperative or Competitive), but the distinction was made using the real mode of the AA rather than the classification made by the controller. Other onsets and parametric regressors were left unchanged. Another GLM was applied to check that the results observed in GLM2 were not simply due to the effect of volatility of the rewarded target. This GLM (GLM4) is similar to GLM2, i.e., trials were classified according to the sign of Δ - δ. The only difference was that we added the actual probability that the AA would choose the same target as the previous trial as a parametric regressor at both the time of the cue and at the outcome. For each GLM, we turned off the serial orthogonalization function of regressors to allow them to compete for the variance.

We computed one paired t-tests with contrasts for main effect of Δ in GLM1 and effect of PE at the outcome time. Then we computed the contrast between competitive and cooperative PE regressors in GLM2, GLM3 and GLM4. Finally, we computed a paired t-test between this contrast, derived from GLM2 and GLM3, to formally test whether activation coming from the difference between classified trials was significantly higher than those coming from the difference between the actual modes of interaction as determined by the trial block.

Reported brain areas show a significant activity at the threshold of $p < 0.05$, whole brain family-wise error (FWE), corrected for multiple comparisons at the cluster level (threshold at $P < 0.001$ uncorrected).

## Psychophysiological interaction (PPI) analysis

We defined the attribution of cooperative or competitive intentions by the winning model at the time of decision-making as the psychological factor. Thus, we were able to investigate the difference in effective connectivity when deciding cooperative or competitive intent. For this PPI analysis, we focused on decision time and effective connectivity between regions encoding the others' intentions and all other voxels. Thus, for the physiological factor, we took the BOLD signal of the combined striatal and vmPFC regions elicited in GLM1 as encoding the intention of others. Otherwise, we used the same regressor parameters and onsets as GLM2.

Reported brain areas show a significant activity at the threshold of $p < 0.05$, whole brain family-wise error (FWE) corrected for multiple comparisons at the cluster level (threshold at $P < 0.001$ uncorrected).

To test the hypothesis of a difference in either global or local efficiency between the two classified modes of interaction (competitive or cooperative), we used the *conn toolbox*. We included in the analyses a network with all ROIs of the *conn toolbox*. We then selected the 15% most connected nodes (i.e., adjacency matrix threshold at cost =0.15). This threshold was selected to enable the connectivity graph to be as close as possible to a "small world" when referring to both global and local efficiencies.

## Reporting summary

Further information on research design is available in the Nature Portfolio Reporting Summary linked to this article.

## Data availability

All display items in the main manuscript and supplementary information can be reproduced using the data available at the links below: DOI for github with data and scripts : https://doi.org/10.5281/zenodo.10299140, https://zenodo.org/records/10299140. This repository includes the behavioral data, as well as generated data, the parameters and onsets used to build the first level fMRI GLM, and all the scripts used for modeling and analysis. The pre-processed fMRI data are available at: https://neurovault.org/collections/EOCXPHRJ/. Anonymized raw fMRI data are available upon request by contacting JC Dreher at: dreher@isc.cnrs.fr.

## Code availability

Codes supporting the results are available at GitHub (https://github.com/remiphilipp/Mixture_intention.git). It also includes the Zenodo link: https://zenodo.org/records/10299140.

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

## Acknowledgements

This research has benefited from the financial support of IDEXLYON from Université de Lyon (project INDEPTH) within the Programme Investissements d'Avenir (ANR-16-IDEX-0005) and of the LABEX CORTEX (ANR-11-LABX-0042) of Université de Lyon, within the program Investissements d'Avenir (ANR-11-IDEX-007) operated by the French National Research Agency. This work was also supported by grants from the Agence Nationale pour la Recherche to JCD (ANR Nos. 16-NEUC-0003-01 and ANR-21-CE37-0032), and by CRCNS NIMH grant no. 5R01MH112166-03, NSF grant no. EEC-1028725, and a Templeton World Charity Foundation grant to RPNR. We thank the CERMEP Staff for help during scanning and Pr Edmund Derrington for critically reading and correcting English in the draft of the manuscript.

## Author contributions

J-C.D., R.P., R.P.N.R and D.L. developed the general concept, experiment, and models. R.P. programmed the task and ran the experiment under the supervision of J.-C.D. R.P. and R.J. developed the models and implemented the algorithms under the supervision of J.-C.D., and analyzed the data in collaboration with J.-C.D. R.P., R.J. and J.-C.D. wrote the manuscript in collaboration with K.K., D.L. and R.P.N.R.

## Competing interests

The authors declare that they have no competing interests.
