## [Peer Review File · Nature Communications]

Neurocomputational mechanisms involved in adaptation to fluctuating intentions of othersREVIEWER COMMENTS

Reviewer #1 (Remarks to the Author):

In this study, Philippe et al propose an arbitration mechanism involved in learning about the cooperative versus competitive intentions of another agent. In a novel fMRI task, 32 participants played with an artificial agent (which they believed was a real human player), choosing between a black and a red card. Participants were rewarded if both players chose the same card. However, unbeknownst to them, the other player's intentions change throughout the task, such that at times they were cooperative (motivated to choose the same card as the participants) but at times they were competitive (choosing the opposite card). Using computational modelling, the authors provide evidence that behavior is best explained by a reinforcement learning model modified to include a mentalizing component (the 'influence' model) as well as dynamically varying intentions (cooperative and competitive) weighted by their relative reliability. They also identify a network of brain regions tracking the reliability difference between the two strategies, the reward prediction error at the time of outcome, as well as differential encoding of prediction error depending on the nature of the other player's intentions. Functional connectivity within this network was also found to be elevated during competitive trials.

Overall, this is a very elegant study and well-written paper, which complements very nicely the current literature on arbitration between learning strategies and provides novel insights on the complex social cognitive processes of cooperation and competition. My comments are mostly minor and addressable, except one potentially major concern on the distinguishability/independence of the two strategies in this task, and what this means for the proposed arbitration framework.

Specifically, if I understand the task and the model correctly, the predictions of the cooperative expert will always be completely opposed to the predictions of the competitive expert. In other words, if the model predicts that a cooperative agent would choose the black card with probability p , it also predicts that a competitive agent would choose the red card with probability p (or the black card with probability $1-p$). If that's the case, then the predictions of the two experts cannot be dissociated as they would always be perfectly anti-correlated. As a consequence, I don't believe it would be theoretically sound to model this under a 'mixture of experts' framework, since really it's more about having one expert that either cooperates OR competes, rather than two experts that make independent predictions, can be dissociated behaviorally, computationally and neurally, and combined in a mixture model.

Generally, I came to this reasoning because the answers to the following questions are missing. It would be great if the authors could address them, to the extent possible:

- 1) What are separate and independent behavioral signatures of cooperation and competition in the task?

- 2) Can each signature be recovered by its corresponding computational model, but not the other?
- 3) Currently the model assumes that both cooperation and competition come for the same influence model, simply with a sign difference. Could it be that cooperation and competition rely on different models altogether?
- 4) How correlated are choice probabilities for cooperation and for competition? How about the two reliability signals?
- 5) What are the neural correlates of cooperation learning and of competition learning?

If those questions cannot be addressed because of the anti-correlation issue between the two strategies, then I think the whole framework of arbitration between two experts needs to be re-assessed, as the study fails to distinguish two separate experts in that case.

Other comments:

6) I understand that the intentions of the partner alternated without being signaled, but what were participants told in the instructions about the partner? It may be helpful to provide the exact instructions in the supplemental.

7) Given the structure of the manuscript (methods at the end), it would be helpful to have a short description of the task at the beginning of the result section, to make it easier for the reader to follow.

8) Logistic regression analysis:

- a- Was this run as a mixed effect regression? If so which software and optimization methods?
- b- Number of trials as a control variable: why was the number of trials different across participants?
- c- How correlated are the regressors from t-1, t-2, and t-3? Is it possible that the lack of effect at t-3 is because of high correlation with t-1 or t-2?
- d- Could a similar analysis be run to show that behavior is a mixture of cooperativity and competitiveness learning?

9) Definition of reliability (l.183-184): since the reliability signal is a key part of the arbitration mechanism, it would be great if the authors could motivate a little better their definition of reliability. For the winning model, it may also be interesting to test a variety of reliability calculation to ensure that the chosen 'difference in unsigned value' is the most predictive way of assessing reliability.

10) Fig. 3b and Extended Figure 1 – the authors claim that only the Mixed-Intention model (but not the cooperative or competitive model only – see l.191-193) can predict the behavioral signature of cooperativity shown in Fig. 1. To me, it seems challenging to distinguish between cooperative and mixed-intention model predictions (i.e. Extended Fig. 1b vs Figure 3b) in terms of which one quantitatively reproduce the behavioral patterns from Fig 2a best. The cooperative model seems to be doing quite well in predicting the behavioral patterns from Fig 2a (in particular it does predict cooperativity, presumably as expected given its nature), while the mixed-intention model seems to be predicting some effects that are not actually there in the data (e.g. the effect of participant switch). This makes me wonder if the glm presented in Fig 2a is the best behavioral signature to examine in terms of posterior predictive checks, or if the authors may instead determine a more specific behavioral signature of either strategy separately, as well as of the arbitration, then show that only the mixed-intention model will recover the arbitration effect.

11) Presumably, the k-ToM mixed intention model should also be able to recover behavioral signatures of the two strategies and of their arbitrations. Was that the case? And what features of the influence model makes it a better predictor of behavior than the k-ToM model?

12) l.515: the authors report a winning rate difference between the two conditions, which is expected given that participants are told that their goal was always to choose the same color as the other player, but the other player's behavior only did that during the cooperative blocks. What I am more unclear about however, is why this winning rate difference is an indication that the alternation between the two interaction modes functioned well? Could this actually suggest that participants are failing to detect the competitive mode, hence their worse performance?

13) Was any debriefing questionnaire administered to assess whether participants had a sense that the AA is switching between two strategies?

14) Computational modelling analyses:

a- The model set is quite big. How close to each other are some of these models in terms of their predictions? It would be good to provide a confusion matrix.

b- How was parameter recovery for the winning model, as well as correlations between parameters?

c- How were individual models fit to the data?

d- Why was depth 1 and 2 used for the k-ToM model, and depth 2 and 3 used for BSL?

e- While the Bayesian Model Selection makes sense, it would be helpful to provide metrics such as the pseudo-R2 and/or predictive of each model to give the reader an idea of the 'absolute' fit of the models.

15) Influence model – Equations 9 and 10: was the sign difference between the cooperative and competitive mode attributed by design (i.e. the sign changed on the first trial of a new mode?). This would assume that the participant knows there is a change in the player's strategy and when this change happens, which is not the case. A model where the sign changes after the participant has learned that the other player changed strategy would seem more realistic. Similarly in the k-ToM model, is the utility table learned or fixed according to the true payoffs?

16) fMRI analyses:

a- GLM2: if a participant's bias parameter is really high or really low, this will result in a large majority of trials being classified as a single category. I can see in the methods that three participants were excluded because of that; however, in the remaining included participants what were the number of trials in each classified category?

b- Logistic regression predicting $P(\text{stay})$ from timeseries and controller valence (l.270-282). Because the dlPFC and rTPJ/IPS regions used in this analysis were identified as showing stronger RPE signals in competitive versus cooperative, and RPE is calculated by fitting the model to choice data, it seems circular that the activity in those specific regions would in turn predict choice. I would remove this analysis, or find an unbiased way to run it (for example, run it across all voxels, similar to a searchlight procedure, to find in an unbiased way which regions of the brain significantly interact with the controller valence to predict $P(\text{stay})$).

c- Functional connectivity analyses: because learning in the competitive condition is more difficult (as seen by lower winning rate), and generally the brain seems more engaged in trials classified as competitive, could these results be explained by generally higher functional coupling in the brain in competitive trials, independent of the specific regions that were examined?

17) Table S1: it's unclear how the AA's probability of choosing black is calculated from each of the 16 possible histories presented in the table.

18) There are quite a lot of typos/tense mistakes/missing third person 's', etc, throughout the manuscript. It may be helpful to have the manuscript proofread more carefully.

19) It would be great if the authors were willing to share their code and data, for open science and reproducibility practices.

Reviewer #2 (Remarks to the Author):

Philippe and colleagues present a paper that explores a novel social decision-making task that switches rapidly between interactional contexts. In the competitive context two participants play “Hide and Seek”, whereas in the cooperative context they play a complementary task, in which identical actions are rewarded. They undertake a comprehensive model comparison of between different “modularized” social decision-making models and conclude that the Mixed-intention Influence model provides the best fit to the behavioral choice data. They then explore neural correlates of RPEs in both interactional contexts in a model-based fMRI analysis and find ventral striatum (which also correlates with what the authors call reliability difference), rACC, lateral OFC and probably IPS as common PE effects in both contexts, whereas in right dlPFC, TPJ and IPS exhibited significantly stronger PE effects in the competitive contexts. Finally, they showed through a PPI analysis that the regions encoding the reliability difference (vStr, and vmPFC) are functionally connected to those regions that showed a stronger PE effect in the competitive context.

This is a very comprehensive paper with an interesting task and lots of details in both the modeling of behavior and in terms of model-based fMRI. In general, the paper is well-written (although at times a bit long). It addresses a topic that should be also interesting to researchers outside the immediate field of social decision-making and therefore suited for the broad readership of Nature Communications. I would support a publication, if the authors could respond to the following questions and suggestions.

Computational modeling

I was a bit surprised by how the Mixed Intention Influence model computed the reliability of both controllers. In essence, the difference of the updated value signal in both controllers is filtered through a bias-corrected softmax function to compute a linear weight for both controllers. This is quite different from how one would usually estimate the reliability of two signals through the precision of their estimates (as it is done in the HGF model or theoretically in the model-based/model-free RL literature (e.g. Daw et al., 2005)). I don't see how this softmax has anything to do with the reliability of the controllers, it is only calculating which controller provides the higher estimate at the moment. The reliability estimate is a critical element in the model and also for the results, so this part needs to be better explained and justified (and maybe named differently).

Furthermore, the posterior predictives calculated from the Mixed Intention model (Fig 3b) are quite different from the model-free analysis of the participant data in Figure 2a. Overall, the model appears to be more sensitive to a longer trial history than the participants as demonstrated by significant effects for trials t-3 for all 3 model-free indices. I think this difference and what it means in terms of the decision-making process deserves some discussion.

Neural data

Overall I find the model-based fMRI analyses convincing, but I suggest that the authors spend some time explaining that the ventral striatum is involved in the “reliability difference” at the time of choice (which right now seems more like a value difference), and in prediction error representations after the outcome. What does this dual involvement mean for the characterization of the network that supports social decision-making in different interactional contexts?

Minor points and clarifications

Figures

in Figure 5 there are some inconsistencies in the display threshold for different contrasts. in panel A the authors use a continuous color scale (probably) from $p < 0.001$ uncorr to $p < 0.05$ FWE, whereas in panel B we see a discrete color scale with only uncorrected p-values. This should be homogenized. Furthermore, it would be really helpful, if the violin plots in that figures could be put on the same y-axis and a visual indicator at the zero line. Based on this, it seems to me that rACC only shows a selective effect for competitive, but not for cooperative. Is this just a visual impression and can it be statistically verified that rACC also shows a PE effect in competitive blocks?

In Figure 6 it remains unclear whether all blue regions (vStr and vmPFC) are used as a single combined seed region or whether the PPI was calculated separately for each?

Brain-behavior correlation

The paragraph on page 9 (lines 270-283) are very convoluted and almost incomprehensible. When I read this paragraph I was unable to understand how this analysis was conducted. For instance, what is the “average of the weighted time series”? Why do you take 7 scans (longer than the trials actually lasts) of the weighted signal (what is that anyways?) and use that as a predictor, even though each trial has separable events (cue, decision, outcome) in which distinct decision-related computations are carried out. Averaging across them appears to be too coarse in this case. I think this whole paragraph has to be completely re-written.

Data Sharing

I looked at the Github page from the project and was surprised to find only the VBA code for the winning model, but not for all the others. Were the other models also estimated using the VBA toolbox, or were different estimation methods used. In any case, please provide the code for other models in the comparison as well.

Reviewer #3 (Remarks to the Author):

In this paper, the authors used a dyadic card matching game, where the participant will play against an artificial agent (referred to as AA; though they believe they are playing against a real human). The task involves making a decision between a red and black card by each of the players, and the participant's goal is to always match the cards. The goal of the computer agent switches between cooperation (matching cards) and competition (non-matching cards), in fixed block sizes of 10 and 13 trials, respectively. The participants are not informed about the switches in the strategy of the opponent, and indeed the experimenters did not even mention anything about the possible goals of the other opponent in the instruction phase of the experiment. The computer agent uses the recent history of the participants' game play to make a decision, given the agent's goal at the time.

Using model-free analyses (logistic regression over probability of staying), the authors showed that the participants' behavior is mainly influenced by the history of their outcomes (win/lose), and a so-called "cooperativity signature" which is approximately whether the agent's last action signalled a cooperative intent. The authors used a comprehensive suite of computational models, including heuristic models, as well as different variations of four classes of model-free and model-based algorithms that can be organized along two dimensions of Bayesian/non-Bayesian and mentalizing/non-mentalizing models. For each mentalizing model they considered three versions: competition-only, cooperation-only and mixed-intentions. The mixed-intention models are characterized by two expert modules, which compute the best choice given each competitive and cooperative intentions for the opponent and the final strategy is determined by weighting these strategies with their corresponding reliabilities. Using Bayesian model selection, they identified the best fitting model to be a non-Bayesian mentalizing model (referred to as influence model) which considers both cooperation and competition intentions for the opponent in determining the best strategy to play against such an opponent (mixed-intention). They demonstrated that the choice behavior of the mixed-intention influence model, as well as a hidden variable in the model which tracks the difference between the reliability of the predictions of the competition expert module and the cooperation expert module, are influenced by the same variables that affect the choice behavior of participants, identified in the logistic regression analyses.

Finally, using the mixed-intention influence model to build model-based GLMs, they have identified brain regions that track the difference in reliability between experts at the decision time, as well as the reward prediction error at the time of the outcome. They have also shown that right dlPFC, and the rTPJ are more correlated with reward prediction error in trials classified as competitive versus cooperative. Last but not least, they used two regions associated with the representation of reliability between experts (i.e., ventral striatum and vmPFC) as the seeds in a connectivity analysis. They showed that in trials classified as competitive compared to cooperative, the connectivity of these two seed regions is stronger with brain regions which were more involved in encoding reward prediction error in the same contrast of classified competitive versus cooperative trials.

Big picture comments

The authors propose to study how people “detect cooperative and competitive intentions of others” (according to the title of the paper). Yet the experimental design is not conducive to testing how people detect cooperative and competitive intentions.

First, participants are never asked to report the intentions of the other player. So the most important variable — participants’ detection of others intentions — must be recovered from a model of a different behaviour — card selection. Thus, from this experimental design, we can’t separate the cognitive processes involved in the detection of intentions, from the cognitive processes involved in the formulation of a policy to best respond to those intentions.

Second, the intentions that the participants are asked to infer are atypical to the point of being completely inexplicable. The ‘other player’ (a simple artificial agent) switches every 10 or 13 trials, between ‘cooperative’ and ‘competitive’ modes. Although the authors argue plausibly that in real life, the same individuals sometimes cooperate and sometimes compete, this deterministic phase switching is not plausible. Also, the agent “learns to cooperate” by probability matching to the player’s history of behaviour, without distinguishing between rounds when the agent was cooperating vs competing. Because of this learning rule, the agent’s cooperative behaviour is much less predictable than a human partner who intends to cooperate, with much lower success rates. Because participants were not asked, we don’t know what participants thought was going on, to explain these confusing patterns of behaviour.

Setting design issues aside, what can we make of the results that the authors do report?

For the behavioural data and model selection. The model selection analyses of the behavioural data are thorough and technically well done. Yet the models that are tested are not drawn from a set of plausible alternative psychological or neural hypotheses, particularly for human social cognition. For example, a whole set of models tested here assume that people do not adjust their play at all in response to the switching strategies of the agents — but these are not plausible alternative models of human cognition. It is not a plausible hypothesis about human cognition that people are completely inflexible or insensitive to changing play by their partners. Likewise, while the simpler Influence model is sufficient to explain the current data, the authors themselves suggest that more complex models like POMDP or k-TOM were not selected not because they are poor models of human cognition, but “because in our setting the only information that can be integrated by participants is their own choices, rewards and the history of the choices made by the other.”

So what scientific conclusion is supported by the model selection?

Possibly, the behavioural and model selection component of the paper might have been included mainly to set up the regressors for the fMRI analyses, in which case the authors should say so.

From the discussion of the paper, I infer that one hypothesis is that "social behavior can be explained by a Controller Theory according to which cooperative/competitive social behavior results from the interaction of multiple systems, each proposing possible strategies for action." However, the model fit in this paper does not test this hypothesis. This stated hypothesis predicts that on every trial, two systems in the participants' brain each represent an independent expected reward (and later a prediction error), and a separate arbitration system then weights the two to choose an action. To test this hypothesis, the authors should have built a GLM with the predicted reward, at the decision phase, and the expected reward (at the outcome phase) from each system, over all trials regardless of how they are classified; and a third expected reward (and RPE) from the weighted sum of the two based on reliability. Then they could test whether there are distinct RPEs computed on every trial, for each system and the arbitrated value.

Instead, the authors claim that the VMPFC and ventral striatum represent the 'control system' arbitrating between the two systems. But the actual evidence in the paper shows that these regions are correlated, across trials, with a regressor composed of (a) highest weighted, whether the participant won on the previous trial, and (b) less weighted, whether the agent's action on the previous trial signalled a cooperative intention. I honestly don't see how this pattern of activation supports the argument that the VMPFC and ventral striatum are involved in "arbitration". In the fMRI GLM, this regressor competes for variance with a regressor for expected reward — it would be helpful to see what are the unique components of each of those regressors when controlling for one another.

FWIW, I do think it would be interesting to test whether the VMPFC and ventral striatum (or other brain regions) are independently predicted by the whether the participant won on the previous trial, and by whether the agent signalled an intent to cooperate on the previous trial. The authors specific algorithmic model suggests that these two parameters are composed into a single weighted sum, but there are many alternative representations that could implement the same computation, including representing each of these parameters separately. One possible test of the authors' specific model, for example, would be whether the relative weight of those two features, in one subset of one participant's data, better predicts the activity in the brain in the other half of the same participant's data, compared to when shuffling across participants.

Honestly, one thing that's extremely confusing throughout the paper is that sometimes they compute functions of the probability of choosing black, sometimes of the probability of 'staying' on the same choice as last time, and sometimes the probability that the agent is cooperating, but these are mixed up in a confusing way. The authors need to be much clearer about when the probabilities are in terms of choosing black. For example, Figure 3c baffles me. Prediction error is $\text{Received_reward} - \text{Predicted_reward}$, so should be high when things turn out better than expected; the Figure shows that

— just in cooperative trials — when the outcome was WORSE than expected participants are highly likely to STAY on their current card choice. That makes no sense, and it is a symptom of my trouble with this paper that I cannot even guess which kind of problem this is.

However it helped me understand another puzzle in the paper, which is that RTPJ and DLPFC are recruited for high prediction errors in competitive trials. Assuming this is the same RPE shown in Fig 3, and ‘high positive prediction errors’ mean that participants are more likely to switch cards on the next trial — I am going to assume high RPE is unintuitively defined as when the outcome was WORSE than expected, not better. That is, I am assuming that the more participants were surprised by a loss, on a trial classified as competitive, the more activity in these brain regions. I’m not sure what the authors think this means (they conclude it shows “a differentiation in the implementation and use of the outcome of the social interaction as a function of the classified interaction”, which is a restatement of the results.) Thinking about this task, I imagine these may be the first one or two trials after the model switches from classifying a trial as cooperative (generating higher expected rewards) to competitive (because of lower achieved rewards). Thus one possible interpretation of the results is that RTPJ and DLPFC activity are involved when participants infer that the agent is now playing more competitively — or see the agent as successfully catching or tricking them. I am not sure whether the authors would consider this an alternative interpretation, or a restatement of their own interpretation?

In sum, this review is getting unpardonably long. I have tried very hard to evaluate the scientific claims about the human mind and brain that these authors are making with the results.

Minor points:

- In figure 1b, they should make it clear that these are the payoff matrices of the AA, not the participant.
- Isn't the analysis of predicting $P(\text{stay})$ as a function of time series of activity in dIPFC and rTPJ a circular analysis? Because dIPFC and TPJ correlate with PE, PE affects $P(\text{stay})$ and so the activity of dIPFC and TPJ should affect $P(\text{stay})$ by definition. Are they trying to make causal statements using this analysis?
- What is the claim for extended figure 3b?
- In general, the text can be improved language-wise; it requires lots of edits. Also there are errors in the equations of supplementary material.

Rebuttal letter

Reviewer #1 (Remarks to the Author)

In this study, Philippe et al propose an arbitration mechanism involved in learning about the cooperative versus competitive intentions of another agent. In a novel fMRI task, 32 participants played with an artificial agent (which they believed was a real human player), choosing between a black and a red card. Participants were rewarded if both players chose the same card. However, unbeknownst to them, the other player's intentions change throughout the task, such that at times they were cooperative (motivated to choose the same card as the participants) but at times they were competitive (choosing the opposite card). Using computational modelling, the authors provide evidence that behavior is best explained by a reinforcement learning model modified to include a mentalizing component (the 'influence' model) as well as dynamically varying intentions (cooperative and competitive) weighted by their relative reliability. They also identify a network of brain regions tracking the reliability difference between the two strategies, the reward prediction error at the time of outcome, as well as differential encoding of prediction error depending on the nature of the other player's intentions. Functional connectivity within this network was also found to be elevated during competitive trials.

Overall, this is a very elegant study and well-written paper, which complements very nicely the current literature on arbitration between learning strategies and provides novel insights on the complex social cognitive processes of cooperation and competition. My comments are mostly minor and addressable, except one potentially major concern on the distinguishability/independence of the two strategies in this task, and what this mean for the proposed arbitration framework.

0) Specifically, if I understand the task and the model correctly, the predictions of the cooperative expert will always be completely opposed to the predictions of the competitive expert. In other words, if the model predicts that a cooperative agent would choose the black card with probability p , it also predicts that a competitive agent would choose the red card with probability p (or the black card with probability $1-p$). If that's the case, then the predictions of the two experts cannot be dissociated as they would always be perfectly anti-correlated. As a consequence, I don't believe it would be theoretically sound to model this under a 'mixture of experts' framework, since really it's more about having one expert that either cooperates OR competes, rather than two experts that make independent predictions, can be dissociated behaviorally, computationally and neurally, and combined in a mixture model.

We thank the reviewer for describing our study as very elegant and well-written. Below, we address the main concern of the reviewer about the distinguishability/independence of the two strategies. Specifically, the reviewer was concerned that the predictions of the cooperative expert could always be completely opposite (anti-correlated) to the predictions of the competitive expert. However, this is not the case. Whereas the objective probability for the artificial agent to cooperate is directly related to its probability to compete, the model fit to the participant's choices was designed in a way that allows these probabilities to be either correlated, anti-correlated or not correlated at all. The two experts start with the same prediction concerning the choice of the other player ($p_{coop,0}^* = p_{comp,0}^* = 0.5$), but then undergo an independent evolution across trials:

$$\begin{aligned} p_{coop,t+1}^* &= p_{coop,t}^* + \eta_1(P_t - p_{coop,t}^*) + \eta_2 2\beta p_{coop,t}^* (1 - p_{coop,t}^*) (Q_t - q_{coop,t}^{**}) && \text{cooperative expert} \\ p_{comp,t+1}^* &= p_{comp,t}^* + \eta_1(P_t - p_{comp,t}^*) - \eta_2 2\beta p_{comp,t}^* (1 - p_{comp,t}^*) (Q_t - q_{comp,t}^{**}) && \text{competitive expert} \end{aligned}$$

Thus, we can distinguish 3 main cases. First, if the first order prediction error $\eta_1(P_t - p_{coop,t}^*)$ is globally high with respect to the second order prediction error $\eta_2 2\beta p_{coop,t}^* (1 - p_{coop,t}^*) (Q_t - q_{coop,t}^{**})$, strategies of both experts will be correlated throughout the game (because the first order PE is identical for both experts). Second, if the two PE (i.e first and second order) are of the same order of magnitude, the cooperative and competitive strategies will not be correlated at all. Third, if the second order PE is globally high with respect to the first order PE, the two strategies will be anti-correlated. These 3 cases demonstrate

the validity of the 'mixture of experts' framework for modeling our task: since the 2 experts are not structurally anti-correlated, our theoretical approach cannot be reduced to only one expert that either cooperates or competes. In addition, to fully address the reviewer's concern, see below (our answer to point 4). We show the actual correlations between the predictions of the 2 experts for each of the 32 participants. They show either a correlation, no correlation or anti-correlation between the predictions of the 2 experts. This clearly demonstrates that the 2 experts are not always anti-correlated. Finally, additional analyses (see below) indicated no correlation between the reliability of the cooperative expert and the reliability of the competitive expert. This shows that the two strategies (cooperative and competitive) are not reliable at the same time.

Thus, overall, the above demonstrates that the two separate experts are not anti-correlated and that they make independent predictions. These two experts can be dissociated computationally and behaviorally. At the brain system level, both experts engage the same brain system. Moreover, as reported before, the controller (mixture of experts model) identified at the brain system level, is based on the reliability of the 2 experts, which are not correlated.

Generally, I came to this reasoning because the answers to the following questions are missing. It would be great if the authors could address them, to the extent possible:

1) What are separate and independent behavioral signatures of cooperation and competition in the task?

We apologize for the lack of definition of the signatures of cooperation and competition for the behavior of the participants in the original version. We used the probability of staying on the same target as the cooperative signature of the participants, and the probability of switching to the other target as the competitive signature of participants. We have now explicitly stated these definitions on **page 6, line 154**.

Formally, these definitions can be justified by the fact that the Nash equilibrium can be used to identify two independent behavioral strategies. When the subject plays the matching pennies game, the Nash equilibrium is to switch the target with the probability of 0.5 in order to be unpredictable. By contrast, when they play the coordination game, there is a pure-strategy Nash equilibrium which is to always choose the same target. Although the coordination game also has a mixed strategy equilibrium, this is not evolutionarily stable and produces less desirable outcomes for both players (i.e., Pareto dominated by the pure Nash equilibrium). Therefore, the probability of staying or switching is much more likely to deviate from 0.5 when the subject plays the coordination game than the competitive matching-pennies task.

Please note that these definitions of behavioral signatures concern the participants' behavior. These signatures need to be distinguished from the 'cooperativity signature' of AA introduced on previous lines 151-153, which concerned the cooperativity of the Artificial Agent from the point of view of the participant (i.e., how the participant estimated the cooperativity of the "other" trial by trial).

2) Can each signature be recovered by its corresponding computational model, but not the other?

To check if the behavioral signature of competition and cooperation described above could be recovered by either the competitive or cooperative expert alone, we conducted 2 logistic panel data regressions (one for the cooperative model alone and one for the competitive model alone), clustered by simulation on the "Stay" strategy. To do this, we first generated a total of 100 datasets with the influence competitive model playing against the same sequences of AA's choices as the real players. To avoid contingencies between the AA's choices and the model's choices, we used the sequence of AA's choices generated against real participants as a non-contingent opponent for the model which generates new data (see point 10). We next generated data with the influence cooperative model using the same method. These logistic regression analyses included only a constant as predictor variable to study the probability to stay, independently of other variables of behavior that were simulated by the competitive expert alone or the cooperative expert alone. These two analyses showed that each expert is able to reproduce its corresponding behavioral signature. That is, the probability to stay of the competitive expert was 0.4966 (CI[0.4807; 0.5105]) whereas against the same sequences of AA's choices, the cooperative expert stayed on the same target significantly more frequently with a probability of 0.5773 (CI[0.5610; 0.5936]). Since $P(\text{stay}) = 0.5773$ which is > 0.5 (logistic regression: $p < 0.0001$), the cooperative expert cannot recover the

competitive signature accurately and the competitive expert cannot recover the cooperative signature. We thank the reviewer for asking this point of clarification. We have now added these analyses and new results on page 7, line 214 of the paper:

To check the validity of the competitive/cooperative behavioral signature, we performed two additional analyses showing that each signature could only be recovered by its corresponding expert model, and not by the other. These analyses consisted of logistic panel data regressions (one for the cooperative model alone and one for the competitive model alone) clustered by simulation on the “Stay” strategy. To do this, we first generated a total of 100 datasets with the influence competitive model playing against the same sequences of AA’s choices as the real players. To avoid contingencies between AA’s choices and model’s choices, we used the sequence of AA’s choices generated against real participants as a non-contingent opponent for the model, which generates new data. We next generated data with the influence cooperative model using the same method. These logistic regression analyses included only a constant as predictor variable to directly compare the probability to stay independently of other variables (the logistic link function let us take into account the fact that as probabilities, the data we are studying are comprised between 0 and 1), on behavior simulated by the competitive expert alone, or the cooperative expert alone. These two analyses showed that each expert is able to reproduce its corresponding behavioral signature. That is, the probability to stay on the same target of the competitive expert was 0.4966 (CI[0.4807; 0.5105]) whereas against the same sequences of AA’s choices the cooperative expert stayed on the same target significantly more frequently with a probability of 0.5773 (CI[0.5610; 0.5936], logistic regression: $p < 0.0001$). Thus, the cooperative expert cannot recover the competitive signature accurately and the competitive expert cannot recover the cooperative signature.

3) Currently the model assumes that both cooperation and competition come for the same influence model, simply with a sign difference. Could it be that cooperation and competition rely on different models altogether?

It is theoretically possible that competition and cooperation could rely on different models. However, we assumed for the purpose of parsimony that they come from the same influence model. Indeed, from the point of view of the participants, there is no indication that there are two modes of interaction. Therefore, it is more parsimonious to assume that a single process (i.e., same computational model for both experts) is engaged throughout the task.

We believe that if the two modes were explicitly presented, the two cognitive processes engaged in the Cooperative and Competitive modes would not be expected to be *a priori* the same (Devaine, Hollard, and Daunizeau 2014b; Hill et al. 2017). Specifically, we would expect that in an explicit cooperation mode, participants would never switch target, to be as predictable as possible (e.g., win-stay vs loose-switch, ..) , and, that in explicit competition, participants would use deeper mentalizing strategies (e.g., k-TOM, influence model ...) (Devaine, Hollard, and Daunizeau 2014a). Again, the situation is very different in our experiment because the Cooperative and the Competitive modes were not explicitly signaled to participants.

Also, as noted by the reviewer, the two influence models differ only by a sign difference. Therefore, we think that it is important to clarify the meaning of ‘sign difference’ in the models. The reviewer might have assumed that if a cooperative expert chooses the red card with probability p , the competitive expert chooses the red card with probability $1-p$, but this was not the case. In fact, each expert model makes choice prediction according to three terms: its own previous prediction, its own reward prediction error (first order prediction error) and its own second order prediction error (reflecting the learning of the other). Thus, the only difference between the two experts is the sign of the third term (the second order PE). However, because the next prediction of each expert depends on its previous prediction, each expert will follow its own ‘walk’. To clarify this, we have added the following sentences to the description of the mixed-intention influence model in the Supplementary Information (line 365 page 12, SI):

When considering mixed-intentions models (*k*-TOM and influence models), we made the assumption that the cooperative expert and the competitive expert come from the same model (i.e., influence model or *k*-TOM) because, from the point of view of the participants, there is no indication that there are two modes of interaction. Therefore, it is more parsimonious to assume that a single process (i.e., same computational model for both experts) is engaged along the task.

4) How correlated are choice probabilities for cooperation and for competition? How about the two reliability signals?

Our data showed the correlation between choice probabilities of the two experts varied substantially across participants (see figure I). In fact, this correlation can either be positive, negative or non-significant as illustrated by the figure I below.

Figure I. Subject by subject correlations between choice probability computed by the cooperative expert (x-axis) and the competitive expert (y-axis). Each dot represents one trial. The line represents the linear regression.

Choice probabilities of the two experts were significantly correlated at the group level: $R^2 = 0.21$ (CI[0.10;0.36]) ($p < 0.0001$). However, this correlation is not forced by the equations of the models (i.e., this correlation is conjectural and is NOT structural) since it depends upon free parameters, the choices of the participants and the choices of the AA.

Regarding the question of correlation between the reliability signals, we observed no correlation: $R^2 = 0.005$ ($p = 0.148$) between the reliability of the cooperative expert and the reliability of the competitive expert (see figure II below). **This shows that the two strategies are not necessarily reliable at the same time. This is an important point because it shows the importance of the second order prediction error term to differentiate the two strategies (competitive or cooperative).**

Figure II. Graph representing the reliability of the competitive expert according to the reliability of the cooperative expert. Each dot represents one trial. No correlation between the two reliability signals was found $R^2=0.005$ ($p=0.148$).

Please note that we neglected to mention in the original version of the manuscript that the probability of the next action of the opponent computed by each expert is transformed into a decision value according to the payoff matrix (figure 1 from the original article). To transform probability into decision value, we used the following equation : $DV = P_{expert} * (Payoff(1,1) - Payoff(1,2)) + (1 - P_{expert}) * (Payoff(2,1) - Payoff(2,2))$. Then, the decision value is the sum of the two decision values weighted by their respective probability to be in cooperative and competitive mode: $DV_{total} = DV_{coop} * P(\text{be in cooperatif}) + DV_{comp} * (1 - P(\text{be in cooperatif}))$. We refer to each of the two terms as the cooperative component and the competitive component respectively. We have now introduced these equations and definitions in the SI.

We made the following changes in the Supplementary information:

Lines **272, page 9** :

“Then, the fictitious agent uses the inferred probability that the other chooses a, the payoff matrix and the other temperature to compute a decision value.

$$DV = \frac{p^*}{Temperature} * (Payoff(\text{self} = a, \text{other} = a) - Payoff(\text{self} = a, \text{other} = \bar{a})) + \frac{(1-p^*)}{Temperature} * (Payoff(\text{self} = \bar{a}, \text{other} = a) - Payoff(\text{self} = \bar{a}, \text{other} = \bar{a})) \quad \text{Eq 8}''$$

Lines **297, page 10** :

“As for the fictitious agent, the influence learner uses the inferred probability that the other chooses a, the payoff matrix and the other temperature to compute a decision value (Eq 8).”

Lines **396, page 13** :

“Then, with $P_{comp}^{a,t}$ and $P_{coop}^{a,t}$ we computed the decision value given the Competitive and Cooperative payoff matrix, DV_{comp}^a and DV_{coop}^a respectively and weighted them by the probability of the corresponding mode of interaction to compute the total decision value:

$$DV^t = P_{coop}^t * DV_{coop}^a + (1 - P_{coop}^t) * DV_{comp}^a \quad \text{Eq 27}$$

We call $P_{coop}^t * DV_{coop}^a$ the cooperative component of the model and $(1 - P_{coop}^t) * DV_{comp}^a$ the competitive component of the model. The sigmoid function s generated the probability of selecting choice a at trial t .

These 2 components (cooperative and competitive components) were only weakly correlated with only 0.4% of common variance ($R^2 = 0.0044$, $p = 0.001$). Below (Fig III), we show this weak correlation. Note that when the decision value computed by the cooperative expert (respectively competitive expert) is around 0, the variance of the decision value of the competitive expert (respectively cooperative expert) is large. This indicates that when one expert is reliable, the other one is often unreliable. Globally, when one expert has a precise prediction regarding the best future choice for the participant, the other expert often proposes a less reliable choice. Thus, overall, the two experts are complementary.

Figure III. Graph representing the competitive component of the global decision value in function of the cooperative component of the mixed-intention influence model. The global decision value is computed as follows: $DV = P(\text{be in competitive}) * DV_{competitive} + P(\text{be in cooperative}) * DV_{cooperative}$. The first term of the sum is called the competitive component; the second term of the sum is called the cooperative component. Each dot represents one trial. The line represents the linear regression. We found a slight correlation with only 0.4% of common variance ($R^2=0.0045$, $p= 0.001$).

Finally, one might expect the two experts to have common variance because they both include first order prediction errors which allow the “win-stay/lose-switch” strategy to be generated. Indeed, we found that either competitive and cooperative experts alone could not use the positive outcome to increase the probability to stay on the same target in the next trial (see answer to question 10).

To answer the reviewer’s concern about the distinguishability of the two experts at the behavioral and computational levels, we have now included the following paragraph in the Supp. Information on page 5, and added three extended figures in the SI.

Evidence for separate cooperative and competitive experts

Below, we demonstrate the distinguishability and independence of the two experts at the computational and behavioral levels.

First, it should be noted that the predictions of the cooperative expert are not always completely inversely-correlated to the predictions of the competitive expert. Indeed, the model was designed in a way that allows the two experts be either correlated, inversely-correlated or not correlated at all. The two experts start with the same prediction on the choice of the participant ($p_{coop,0}^ = p_{comp,0}^* = 0.5$), but then have an*

independent evolution across trials (see Equation 11). Thus, we can distinguish 3 main cases. First, if the first order prediction error $\eta_1(P_t - p_{coop,t}^*)$ is globally high with respect to the second order prediction error $\eta_2 2\beta p_{coop,t}^* (1 - p_{coop,t}^*) (Q_t - q_{coop,t}^{**})$, strategies of both experts will be correlated throughout the game (because the first order PE is identical for both experts). Second, if the two PE (i.e, first and second order) are of the same order of magnitude, the cooperative and competitive strategies will not be correlated at all. Third, if the second order PE is globally high with respect to the first order PE, the two strategies will be inversely-correlated. These 3 cases demonstrate the validity of the 'mixture of experts' framework for modeling our task since the 2 experts are not structurally inversely-correlated. Thus, our model cannot be reduced to a single expert that either cooperates or competes.

Second, Extended Figure 1 displays the correlations between the predictions of the 2 experts for each of the 31 participants. They show either a correlation, no correlation or inverse-correlation between the predictions of the 2 experts. Thus, the two experts can be distinguished and they are not necessarily inversely-correlated. Choice probabilities of the two experts were significantly correlated at the group level: $R^2 = 0.21$ (CI[0.10;0.36]) ($p < 0.0001$). However, this correlation is not forced by the equations of the models (i.e., this correlation is conjectural and is NOT structural) since it depends upon free parameters, and the interaction between the choices of the participants and the choices of the AA.

Third, we observed no correlation between the reliability of the cooperative expert and the reliability of the competitive expert ($R^2 = 0.005$, $p = 0.148$) (Extended figure 2). This indicates that the two strategies (cooperative and competitive) are not reliable at the same time. This also demonstrates the importance of the second order prediction error term to differentiate the two strategies.

Fourth, the cooperative and competitive components of the decision value are correlated with only 0.4% of common variance ($R^2 = 0.0044$, $p = 0.001$) (Extended figure 3). Note that when the decision value computed by the cooperative expert (respectively competitive expert) is around 0, the variance of the decision value of the competitive expert (respectively cooperative expert) is large. This indicates that when one expert is reliable, the other one is often unreliable. Globally, when one expert has a precise prediction regarding the best future choice for the participant, the other expert often proposes a less reliable choice. Thus, overall, the two experts are complementary.

5) What are the neural correlates of cooperation learning and of competition learning?

To investigate the brain system engaged by the two experts, we performed an ANOVA including DV and PE of each expert. The signals related to the two experts showed common brain activity in the ventral striatum for the decision value of staying on the same target ($P = 0.007$ FWE cluster corrected, threshold at $p < 0.001$ at the cluster level, Fig IV). When directly comparing between the decision value of the two experts, we did not find any separate brain region ($P > 0.95$ FWE cluster corrected, threshold at $p < 0.001$ at the cluster level). For the reward prediction error (PE), the two experts showed common activity in the ventral putamen, the anterior medial PFC, posterior cingulate cortex and the lateral OFC (Fig V) (all $P_s < 0.02$ FWE cluster corrected, threshold at $p < 0.001$ at the cluster level). These results, displayed in Fig V, were obtained by building, for the mixed-intention influence model, a new GLM including, at the first level, decision values (DV) from the competitive expert, decision values from the cooperative expert, PE from the competitive expert and PE from the cooperative expert. Specifically, there were 4 onsets including the time of the cue presentation (cards on screen), participant's button press, AA's choice and feedback time. At the time of the cue onset, the parametric regressors were the decision value for staying on the same target generated by the competitive expert, and the DV for staying on the same target generated by the cooperative expert. Similarly, at the time of feedback, the regressors were the reward prediction error (PE) generated by the competitive expert and the PE generated by the cooperative expert.

Figure IV. Common ventral striatal region encoding decision value of the cooperative and competitive experts.

Figure V. Common brain regions encoding prediction error of the cooperative and competitive experts included the ventral putamen, the anterior medial PFC, posterior cingulate cortex and the lateral OFC.

We have added the following paragraph in the Supp. Information on pages 6 and added these 2 new extended figures to the Supp. Information:

To investigate the brain system engaged by the 2 experts, we performed an ANOVA including DV and PE of each expert. The experts showed common brain activity in the ventral striatum for the decision value of staying on the same target ($P = 0.007$ FWE cluster corrected, threshold at $p < 0.001$ at the cluster level, Extended Figure 8.a). When directly comparing between the decision value of the 2 experts, we did not find any separate brain region ($P > 0.95$ FWE cluster corrected, threshold at $p < 0.001$ at the cluster level). For the reward prediction error (PE), the two experts showed common activity in the ventral putamen, the anterior medial PFC, posterior cingulate cortex and the lateral OFC (Extended Figure 8.b) (all P s < 0.02 FWE cluster corrected, threshold at $p < 0.001$ at the cluster level). These results were obtained by building a new GLM for the mixed-intention influence model including, at the first level, decision values (DV) from the competitive expert, decision values from the cooperative expert, PE from the competitive expert and PE from the cooperative expert. Specifically, there were 4 onsets including the time of the cue presentation (cards on screen), participant's button press, AA's choice and feedback time. At the time of the cue onset, the parametric regressors were the decision value for staying on the same target generated by the competitive expert, and the DV for staying on the same target generated by the cooperative expert. Similarly, at the time of feedback, the regressors were the reward prediction error (PE) generated by the competitive expert and the PE generated by the cooperative expert.

If those questions cannot be addressed because of the anti-correlation issue between the two strategies, then I think the whole framework of arbitration between two experts needs to be re-assessed, as the study fails to distinguish two separate experts in that case.

We think that the answers made above clearly indicate that there is no anti-correlation between the two strategies, that the two experts make clearly distinct predictions regarding the probability of the

next choice and its reliability, and that the behavioral signature of cooperativity can be recovered by the cooperative expert alone but not by the competitive expert (and conversely the behavioral signature of competitiveness can be recovered by the competitive expert alone but not by the cooperative expert). Thus, the two experts can be dissociated at the behavioral and computational levels and can be combined in a mixture model. At the brain system level, we observed a common ventral striatal activity for the DV of the competitive expert and the cooperative expert. We did not find separate brain regions for the 2 experts when directly comparing their neural correlates. This is not surprising given our design which is very different from a design in which Cooperative modes and Competitive modes would be explicitly signaled. In particular, we did not assume that there should be different brain representations for cooperation learning and competition learning because the experts only differ by their priors (based on the payoff matrix). We did assume, however, that we can identify the brain representation of the reliability difference signal (representing the other's most likely intention). It should be emphasized that we did not assume, as in O'Doherty et al. 2021, that the two experts engage separate brain regions in the context of our task (in which the only cognitive process was inference learning). To conclude, we showed that it is necessary to have two separate experts to compute the reliability difference which is encoded in the brain. The brain system encoding this controller is based on the reliability of two separate experts which make distinct predictions.

Other comments:

6) I understand that the intentions of the partner alternated without being signaled, but what were participants told in the instructions about the partner? It may be helpful to provide the exact instructions in the supplemental.

Instructions were in French originally. We have added both the English and French versions to the supplementary files. Deliberately, we never used words synonymous with "against" or "partner" since such vocabulary could influence the participant's prior with respect to the goal of the other. We added the French and English instruction in SI **page 1**.

Instructions: You are going to be faced with 4 cards: two face down, those of the person you will be interacting with, and two face up, your cards. Each turn you will have to choose one of your 2 cards. When the other player has chosen her card, the card will be put in the middle, face down, without you being able to know which one has been chosen. When both of you have made your choices, you will see the card that the other player has chosen and a one-euro coin if you win, or a crossed-out one-euro coin if you lose. You win if both of you chose the same color card, otherwise you lose. You do not know what are the rules of the game for the person you are interacting with and you do not know which reward she will receive. There will be around 150 trials to perform in the scanner. There will be a one minute break half way through. The person with whom you interact will not change between the 2 blocks. Every time you win, you will receive an extra 10 cents as a reward.

7) Given the structure of the manuscript (methods at the end), it would be helpful to have a short description of the task at the beginning of the result section, to make it easier for the reader to follow.

We added the following text in the manuscript lines **150 page 6** :

In the mixed-intention task, participants were led to believe that they were interacting with another participant, while in fact they were interacting with an artificial agent (AA). Participants were asked to choose one of two cards to match the other participant's choice. Unbeknownst to the participant, the AA tried to match the two cards (coordination game) or to mismatch them (competitive game: matching pennies).

8) Logistic regression analysis:

a- Was this run as a mixed effect regression? If so which software and optimization methods?

We specified a data panel on *Stata* (Version 14.1) to account for the repeated measure within each subject's game. We then used a random-effects model with cluster-robust standard errors for panels nested within subjects, allowing for intra-subject correlation. The optimization method was maximum likelihood estimation using Gauss-Hermite quadrature to approximate the likelihood. For the coefficient

estimation, we used a robust estimator (sandwich method also called Eicker–Huber–White standard errors) to be robust for heteroscedasticity.

We have now made the following changes to the paper to specify this point on **page 20, line 621**:

For both logistic and linear regression, we specified a data panel on Stata (Version 14.1) to account for the repeated measure within each subject’s game. We then used a random-effects model with cluster–robust standard errors for panels nested within subjects, which allowed for intra-subject correlation. The optimization method was maximum likelihood estimation using Gauss–Hermite quadrature to approximate the likelihood. For the coefficient estimation, we used a robust estimator (sandwich method also called Eicker–Huber–White standard errors) to be robust for heteroscedasticity.

b- Number of trials as a control variable: why was the number of trials different across participants?

We apologize for the confusion. This was not the total number of trials which was added as a fixed effect, but the rank (which we called number) of each trial, i.e., whether it was the first, second or third trial etc. used as a random effect of time to account for learning or other time effects.

We have made the following changes on **line 162**: *We also added sex, age and the rank of each trial (first, second or third trial etc. i.e., time indicator) as control variables.*

c- How correlated are the regressors from t-1, t-2, and t-3?

The correlations between regressors are reported in the table below (all trials included).

Regressors	Victory (t-1)	Victory (t-2)	Victory (t-3)	Switch (t-1)	Switch (t-2)	Switch (t-3)	Wslsaa (t-1)	Wslsaa (t-2)	Wslsaa (t-3)
Victory (t-1)	1								
Victory (t-2)	0.0496	1							
	0.0203								
Victory (t-3)	0.0419	0.0473	1						
	0.1328	0.0375							
Switch (t-1)	-0.0283	-0.1717	0.0171	1					
	1	0	1						
Switch(t-2)	0.0004	-0.0258	-0.1707	0.0234	1				
	1	1	0	1					
Switch(t-3)	-0.043	0.0009	-0.026	0.0818	0.0227	1			
	0.1031	1	1	0	1				
Wslsaa(t-1)	0.087	0.0245	0.0604	-0.035	-0.0495	-0.0168	1		
	0	1	0.001	0.5355	0.0204	1			
Wslsaa(t-2)	0.0509	0.0589	0.0858	-0.0656	-0.0778	-0.0324	0.0926	1	
	0.0155	0.0017	0	0.0002	0	0.9064	0		
Wslsaa(t-3)	0.014	0.0511	0.058	-0.0681	-0.0655	-0.0757	0.0669	0.0644	1
	1	0.0154	0.0023	0.0001	0.0002	0	0.0001	0.0003	

Table 1: Correlation between regressors in the panel data logistic regression, all trials included. First line is the estimate of the correlation, second line is the p-value of this estimate corrected with Bonferroni correction.

As can be seen in the tables below including only the actual Cooperative trials (not the classified ones) or the Competitive trials, these correlations were mainly driven by Cooperative blocks. For the Competitive blocks, there was much less covariance between regressors. Such a correlation would be expected between win and switch regressors because of the structure of the task and the policy of the artificial agent, which bases its choices on the previous two choices and victories of the participant.

Regressors	Victory (t-1)	Victory (t-2)	Victory (t-3)	Switch (t-1)	Switch (t-2)	Switch (t-3)	WsLsaa (t-1)	WsLsaa (t-2)	WsLsaa (t-3)
Victory (t-1)	1								
Victory (t-2)	0.1105	1							
	0								
Victory (t-3)	0.0745	0.0988	1						
	0.0097	0							
Switch(t-1)	-0.1014	-0.1754	0.0053	1					
	0	0	1						
Switch(t-2)	-0.0199	-0.091	-0.1727	0.0676	1				
	1	0.0003	0	0.0317					
Switch(t-3)	-0.0679	-0.018	-0.0917	0.1051	0.0465	1			
	0.0322	1	0.0003	0	0.8316				
WsLsaa(t-1)	0.1023	0.1046	0.0255	-0.1655	-0.0693	-0.1289	1		
	0	0	1	0	0.0235	0			
WsLsaa(t-2)	0.0645	0.063	0.0724	-0.0904	-0.1222	-0.1155	0.0924	1	
	0.062	0.0801	0.0156	0.0004	0	0	0.0003		
WsLsaa(t-3)	0.0214	0.055	0.0645	-0.0881	-0.0841	-0.103	0.0923	0.0783	1
	1	0.2856	0.0669	0.0008	0.0018	0	0.0003	0.0056	

Table 2: Correlation between regressors in the panel data logistic regression, Cooperative blocks only included. First line is the estimate of the correlation, second line is the p-value of this estimate corrected with Bonferroni.

Regressors	Victory (t-1)	Victory (t-2)	Victory (t-3)	Switch (t-1)	Switch (t-2)	Switch (t-3)	WsLsaa (t-1)	WsLsaa (t-2)	WsLsaa (t-3)
Victory (t-1)	1								
Victory (t-2)	-0.0404	1							
	1								
Victory (t-3)	-0.0128	-0.0214	1						
	1	1							
Switch(t-1)	0.0535	-0.1629	0.0345	1					
	0.3067	0	1						
Switch(t-2)	0.025	0.0418	-0.1673	-0.0214	1				
	1	1	0	1					
Switch(t-3)	-0.0201	0.019	0.0378	0.0591	-0.0007	1			
	1	1	1	0.1305	1				
WsLsaa(t-1)	0.0489	-0.0743	0.0799	0.0997	-0.0276	0.0929	1		
	0.5819	0.0092	0.003	0	1	0.0002			
WsLsaa(t-2)	0.0189	0.04	0.0874	-0.0371	-0.033	0.0477	0.0813		
	1	1	0.0006	1	1	0.6795	0.0023	1	
WsLsaa(t-3)	-0.0089	0.0351	0.0424	-0.0456	-0.0461	-0.05	0.0331	0.0435	1
	1	1	1	0.9035	0.837	0.4997	1	1	

Table 3: Correlation between regressors in the panel data logistic regression, Competitive blocks only included. First line is the estimate of the correlation, second line is the p-value of this estimate corrected with Bonferroni.

Is it possible that the lack of effect at t-3 is because of high correlation with t-1 or t-2?

The lack of effect of the cooperativity signature of AA at t-3 cannot be due to a high correlation between regressors with t-1 or t-2 because a logistic regression of probability to switch target with only the regressors at t-3 (Victory(t-3), Switch(t-3), cooperativity signature of AA(t-3)), shows that there was only an effect of Switch at (t-3) (size effect : 0.047, p=0.010) and not Victory(t-3) (size effect : -0.025, p=0.088) and cooperativity signature of AA(t-3) (size effect : 0.005, p=0.754). We still do not find any effect of cooperativity signature of AA on previous winnings at t-3 (after removing regressors at t-1 and t-2), thus the lack of effect is not due to correlation between regressors.

We have added this analysis on **line 178, page 6**:

The lack of effect of the AA's cooperativity signature of AA at t-3 cannot be due to correlation between regressors with t-1 or t-2 because a logistic regression of probability to switch target with only the regressors at t-3 (Victory(t-3), Switch(t-3), Cooperativity signature of AA(t-3)), shows that there was only an effect of Switch at (t-3) and not Victory(t-3) or Cooperativity signature of AA(t-3).

d- Could a similar analysis be run to show that behavior is a mixture of cooperativity and competitiveness learning?

In Cooperative learning, the best strategy (pure Nash equilibrium) is to always stay on the same target. If we run a similar logistic regression analysis assuming such strategy, this would lead to a probability to stay close to 1 for each regressor (cooperativity signature of AA, previous winning interaction and participant's switch). Respectively, in Competitive learning, the best strategy is to randomly switch target. If we run a logistic regression analysis assuming such strategy, this would now lead to a probability to stay close to zero for each regressor (cooperativity signature of AA, previous winning interaction and participant's switch). The results of our logistic regression on the probability to stay of our participants' behavior (shown in Fig 2a) show that the actual participants' strategies are intermediate between these two extreme strategies (since, for example, the betas of the cooperativity signature of AA differ significantly from 0 at t-1 and t-2). Thus, our results do indeed indicate that the observed behavior is a mixture of cooperative and competitive learning.

9) Definition of reliability (l.183-184): since the reliability signal is a key part of the arbitration mechanism, it would be great if the authors could motivate a little better their definition of reliability. For the winning model, it may also be interesting to test a variety of reliability calculation to ensure that the chosen 'difference in unsigned value' is the most predictive way of assessing reliability.

We thank the reviewer for making this important point. Although we did not include them in the original manuscript, we have constructed and compared several reliability signals for the winning model (mixed-intention influence model). We previously presented only the one which was best able to explain the behavioral data. We have now included all the reliability signals tested in the SI (**pages 13 line 383**). *To motivate our definition of the reliability signal, we tested 4 definitions of reliability signals for the winning model:*

- $|DV_{competitive}| - |DV_{cooperative}|$ *Difference of unsigned DV*
- $|PE_{competitive}| - |PE_{cooperative}|$ *Difference of unsigned PE (O'Doherty et al. 2021)*
- $PE_{competitive} - PE_{cooperative}$ *Difference of PE (Wan Lee, Shimojo, and O'Doherty 2014)*
- $Entropy_{competitive} - Entropy_{cooperative}$ *Difference of entropy: $-p * \log(p) - (1 - p) * \log(1 - p)$ (Charpentier, ligaya, and O'Doherty 2020)*

The motivation for adding the difference of unsigned DV as a measure of reliability difference is that this measure is based on the current expectation of each expert at the time of choice t. This is not the case for the reliability signals based on the difference of unsigned PE or PE, which are based on the difference between the previous expectations (DV_{t-1}) and the previous outcomes (R_{t-1}). When we performed a formal Bayesian model selection between these 4 reliability difference measures, the model with the 'Difference of unsigned DV' as a controller was the best model (protected exceedance probability pEP = 0.83).

Moreover, any mixed-intention influence model with one of these 4 reliability difference signals performs better than all the other 15 models presented in Fig 2.b. This demonstrates the robustness of the mixed-intention influence model, regardless of the chosen reliability difference measure.

We have added this information to the Supplementary Information on line **383, page 13**

To motivate our definition of the reliability signal, we tested 4 definitions of reliability signals for the winning model:

- $|DV_{competitive}| - |DV_{cooperative}|$ *Difference of unsigned DV*
- $|PE_{competitive}| - |PE_{cooperative}|$ *Difference of unsigned PE (O’Doherty et al. 2021)*
- $PE_{competitive} - PE_{cooperative}$ *Difference of PE (Wan Lee, Shimojo, and O’Doherty 2014)*
- $Entropy_{competitive} - Entropy_{cooperative}$ *Difference of entropy: $-p * \log(p) - (1 - p) * \log(1 - p)$ (Charpentier, ligaya, and O’Doherty 2020)*

When we performed a formal Bayesian model selection between these 4 reliability difference measures, the model with the ‘Difference of unsigned DV’ as controller was the best model (protected exceedance probability $pEP = 0.83$). Moreover, the mixed-intention influence model with any one of these 4 reliability difference signals performs better than all the other 16 models presented in Fig 2.b. This demonstrates the robustness of the mixed-intention influence model, regardless of the chosen reliability difference measure.

10) Fig. 3b and Extended Figure 1 – the authors claim that only the Mixed-Intention model (but not the cooperative or competitive model only – see l.191-193) can predict the behavioral signature of cooperativity shown in Fig. 1. To me, it seems challenging to distinguish between cooperative and mixed-intention model predictions (i.e. Extended Fig. 1b vs Figure 3b) in terms of which one quantitatively reproduce the behavioral patterns from Fig 2a best. The cooperative model seems to be doing quite well in predicting the behavioral patterns from Fig 2a (in particular it does predict cooperativity, presumably as expected given its nature), while the mixed-intention model seems to be predicting some effects that are not actually there in the data (e.g. the effect of participant switch). This makes me wonder if the glm presented in Fig 2a is the best behavioral signature to examine in terms of posterior predictive checks, or if the authors may instead determine a more specific behavioral signature of either strategy separately, as well as of the arbitration, then show that only the mixed-intention model will recover the arbitration effect.

To answer the reviewer’s comment, we have now generated new data with the cooperative, competitive and mixed-intention influence models, each playing against the choice sequences that the artificial agent actually generated when facing the real players. The reasoning behind this new analysis is that it now came to our attention that the effects of ‘Previous winning interaction’ (at t-2 and t-3) and the ‘Participant switch’ (at t-1 and t-2), observed on the probability to stay (see Fig 2a compared to previous Fig 3b), result from an interaction between the mixed-intention influence model and the AA. Indeed, in the simulation of the previous Fig 3b, the mixed-intention influence model played with an Artificial Agent. Thus, choices of the AA depended upon the history of choices and outcomes from the mixed-intention influence model. This explains why the previous mixed-intention influence model appears to behave in different ways than participants **if** there is a contingency between the probabilistic choice of the AA and the probabilistic choice of the mixed-intention model.

To disrupt this contingency, we have now generated data with the cooperative, competitive and mixed-intention influence model, playing against the choice sequences that the artificial agent generated. In this approach the mixed-intention influence model adapts to the choices of the AA, but the choices of the AA **do not** depend upon the choices of the mixed-intention influence model. We randomly drew free parameters from a normal distribution centered at the mean of the observed parameters of participants and with the standard deviation observed in the data. We generated 320 sets (32 sequences of the AA choices corresponding to the real sequences obtained when playing real participants * 10 simulations) of data per model type (cooperative, competitive and mixed-intention influence models). The results of these new logistic regressions are shown in Fig VI (new Fig 3b and new extended Figs 1a and 1b). All three models

are now able to reproduce the win/stay-lose/switch strategy, but only the mixed intention influence model is able to reproduce the subtle effect of the cooperativity signature of AA on the strategy to stay. Moreover, the mixed-intention influence model tends to generate less switching after a switch in the previous trial than the other experts alone.

We think that this new analysis consisting of using fixed sequences of the AA choices (to make models play against the same Artificial Agent's data as participants) is more appropriate than our previous approach, because it does not introduce divergences due to the interactions between the AA and the mixed-intention influence model's probabilistic choices. This new approach also does not require us to develop new behavioral signatures for each strategy and their arbitration. We have therefore replaced our previous analyses and Fig 3b and Extended figure 1 in the paper.

We have made the following changes in the Supplementary Information on page 3:

*To determine whether the mixed intention influence model could recover the observed effect of the Cooperativity signature of AA (interaction between the participant's outcome and the following choice by the AA to switch or not), on the probability to stay on the same target, we simulated 310 sets using the influence model from each of the 3 versions (i.e. the competitive expert alone, the cooperative expert alone and the mixed intentions version that arbitrates between the cooperative and competitive experts) playing against the choice sequences that the artificial agent generated against real participants. In this approach the mixed-intention influence model adapts to the choices of the AA, but the choices of the AA **do not** depend upon the choices of the mixed-intention influence model. We randomly drew free parameters from a normal distribution centered at the mean of the observed parameters of participants, and with the standard deviation observed in the data. The results of these logistic regressions are shown in Extended Figure 6. All three models are able to reproduce the win/stay-lose/switch strategy, but only the mixed-intention influence model is able to reproduce the subtle effect of the cooperativity signature of AA on the strategy to stay. Moreover, the mixed-intention influence model tends to generate less switches after a switch in previous trial than the other experts alone.*

We have also updated previous Figure 3b and extended figure 1a and b with the following new figures:

(new Fig 3b)

Figure VI. Model-based generative analysis. We generated 310 sets of data using free parameters from a normal distribution with mean and standard deviation calculated from the models fitted to the population, against the fixed sequences of choices that the artificial agent made against participants during the experiment. We regressed the behavioral decision to stay after selection of a specific target on the previous trial, depending on the interaction of the previous outcome and the action of the artificial agent (“Cooperativity signature of AA”), the success or failure of up to three previous trials, and the action to switch or stay of the participant. Error bars are the 95% confidence interval (random-effect logistic regression). The graph (top) represents the data generated by the mixed-intention influence model. The graphs (bottom) represent the data generated by the cooperative influence model alone (bottom left), and the data generated by the competitive influence model (bottom right). Error bars are the 95% confidence interval. * $p < 0.05$, ** $p < 0.01$, *** $p < 0.001$ (panel data random-effect logistic regression).

11) Presumably, the k-ToM mixed intention model should also be able to recover behavioral signatures of the two strategies and of their arbitrations. Was that the case? And what features of the influence model makes it a better predictor of behavior than the k-ToM model?

To answer this point, we used the same approach as in the previous question 10 (i.e., fixing the sequence of AA choices). To do this, we generated 310 sets of data from the 1-ToM mixed-intention influence model facing the same behavior of the artificial agent that participants faced during the experiment. We then ran a new logistic regression on the new generated data. The results showed that this model does not reproduce the cooperativity signature of AA effect and the weight of previous switch is much more important than that of previous feedback (Figure VII). These results do not fit the behavior observed in participants.

We therefore hypothesize that the mixed-intention influence model performs better than the k-ToM models because it learns from its own prediction error, corrected by a second order term to represent learning of the other. The mixed-intention influence model switches strategy according to the choices of the other. In contrast, the 1-ToM agent predicts the choice of the other, assuming that it is a 0-ToM agent that predominantly imitates the 1-ToM’s previous choices. Thus, the 1-ToM agent bases its current strategy (next choice) on its own previous choices.

Figure VII. Model-free generative analysis. We generated 310 sets of data using mixed intention 1-ToM with free parameters drawn from a normal distribution with mean and standard deviation calculated from the models fitted to the population, against the same artificial agent that participants played. We regressed the behavioral decision to stay after selection of a specific target on the previous trial, depending on the interaction of the previous outcome and the action of the artificial agent (“Cooperativity signature of AA”), the success or failure of up to three previous trials, and the action to switch or stay of the participant. Error bars are the 95% confidence interval (random-effect logistic regression). * $p < 0.05$, ** $p < 0.01$, *** $p < 0.001$ (panel data random-effect logistic regression).

12) I.515: the authors report a winning rate difference between the two conditions, which is expected given that participants are told that their goal was always to choose the same color as the other player, but the other player’s behavior only did that during the cooperative blocks. What I am more unclear about however, is why this winning rate difference is an indication that the alternation between the two interaction modes functioned well? Could this actually suggest that participants are failing to detect the competitive mode, hence their worse performance?

The winning rate difference between the two conditions is not, by itself, sufficient to ascertain that participants detect the two modes of interaction. It is also necessary to observe an adaptation of behavioral strategy after the transition between modes to claim that participants are indeed adapting to the Cooperative and Competitive modes. To confirm our previous interpretation, we have now performed an additional analysis about the switching strategy.

The switching strategy of participants depends on the mode (Cooperative or Competitive) of interaction of the previous trial (Mean difference = 5.43%, $p=0.036$). However, participants also change their switching strategy using the cooperativity signature of the AA, depending on the mode (Cooperative or Competitive) of the Artificial agent on the previous trial (Mean difference = 9.61%, $p=0.0012$, marginal effect of panel data logistic regression clustered by subject of switch strategy on the mode of interaction, the cooperativity signature of the AA and their interactions).

In addition, although these new analyses demonstrate that participants do indeed detect the competitive mode of interaction, they could not perform better than chance in this mode because of the way our artificial agent (AA) was constructed. This AA strategy was based on our pilot data, showing that participants were using a win-stay/lose-switch strategy based on the two previous trials. Thus, the AA was built to predict the participants’ behavior better than chance, and play to defeat them.

We have now included this new analysis on **pages 19** :

‘Moreover, that the design functions well is not only shown by the winning rates, but also by the change in switching strategies of participants. Indeed, the switching strategy of participants depends on the mode (Cooperative or Competitive) of interactions on the previous trial (coefficient of variation=5.43%, $p=0.036$).

but participants also change their switching strategy using the cooperativity signature of AA, depending on the mode (Cooperative or Competitive) of the Artificial Agent in the previous trial (coefficient of variation=9.61%, $p=0.0012$, marginal effect of panel data logistic regression clustered by subject of switch strategy on the mode of interaction, the cooperativity signature of AA and their interactions)'.

13) Was any debriefing questionnaire administered to assess whether participants had a sense that the AA is switching between two strategies?

We did debrief participants and asked the following 3 questions: Did you notice any changes in the way the other player played? If so, which ones? How did you notice it?

At least 16 participants had a sense that the AA was switching between two different strategies (some participants did not answer anything specific to these questions). Below, we translate these debriefing reports:

1. The other player was losing money as he tried to do the "long sequence" technique, but it backfired (nb. The participant seems to think he is playing competitively.). However, sometimes he tried to avoid me.
2. He changes his strategy. In the second game, it was tighter, he understood my strategy.
3. Sometimes the other player would change a move when they were both winning, it looked like he was bored. It looked like the other player was changing.
4. The other had two strategies, either he stayed on the same card while losing, or he changed every other time or tried to change the sequence.
5. Sometimes he would make series of up to 6 times the same card, sometimes he would alternate every two or three choices.
6. What worked at one moment did not work afterwards [...]. We found a logic and then suddenly it didn't work anymore.
7. Sometimes the other would always choose the same card for a long time, then he would vary, then choose the other card all the time.
8. The other person changes his strategy, his intention. At first it looked like a teammate, then an opponent.
9. The other was not consistent. [Sometimes he was consistent, and sometimes he was not consistent.
10. Several changes. When I had the right result, he would change. He would make me feel confident and then change the strategy.
11. He adapted, he made changes in strategy like me.
12. He tried to adapt different strategies by repeating choices or alternating.
13. I noticed changes of rhythm. Sometimes the opponent would follow me and then stop following me. Maybe he was changing the instructions.
14. He would change sequences. Sometimes he would choose 7 times the black and then alternate once of each color [...].
15. At first the other had the same rules, then he changed his strategy (Or did he have reasons to do so?), then sometimes he would go back to the same rules so that we would synchronize again.
16. Did we have the same goal? At first the other seemed to be cooperative. Then I realised that he didn't, I said to myself that we didn't have the same objective. Sometimes we would get into the same rhythm and then it would stop. [...]. The other would do quite long sequences choosing the same color, then he would change after 2 or 3 repetitions.

14) Computational modelling analyses:

a- The model set is quite big. How close to each other are some of these models in terms of their predictions? It would be good to provide a confusion matrix.

We agree with the reviewer that identifiability of models is important. To check that our models make distinct prediction on behavior, we generated 5 datasets for each of our 31 participants independently (total of 155 datasets). To do so, we used randomly generated free parameters with mean

value equal to the fitted free parameters of each participant. Generative models made decision facing a non-contingent Artificial Agent (AA, i.e., we used the real sequence of choices that participants observed during the experiment, see point 10). Here we show the exceedance probability (the probability that one model is more frequent than the other in the observed population) across our set of models for every generative models. Figure VIII (the vertical red rectangle) shows that no model produced behavior that could be confounded with the winning model (mixed-intention influence model). Moreover, the behavior generated by the mixed-intention influence model could not be recovered by another model in our model set (horizontal red rectangle).

Figure VIII. Confusion matrix. We generated 5 datasets for each of our 31 participants (total of 155 datasets). To do so we used randomly generated free parameters with mean value, the fitted free parameters of each participant independently. The rows are the generative models. The columns are the predicted model. The colors represent the probability that one model is more frequent (in our population of generated dataset) than another given a Bayesian model selection (Exceedance probability)

We added the following paragraph in supplementary files **page 5**:

“To check that our models make distinct prediction on behavior, we generated 5 datasets for each of our 31 participants independently (total of 155 datasets). To do so we used randomly generated free parameters with mean value, the fitted free parameters of each participant. Generative models made decisions facing a non-contingent Artificial Agent (AA, i.e., we used the real sequence of choices that participants observed during the experiment, see point 10). We found that no model produces behavior that could be confounded with the winning model (mixed-intention influence model). Moreover, the behavior

generated by the mixed-intention influence model could not be recovered by any other model in our model set (horizontal red rectangle), see **Extended figure 6.**“

b- How was parameter recovery for the winning model, as well as correlations between parameters?

We agree with the reviewer that the identifiability of free parameters is important. The parameter recovery analysis from the mixed-effect model showed that all the estimated parameters significantly explain the variance of the corresponding generated parameters (Fig IX).

As can be seen in Table IV, there are interactions between free parameters of our generative model (mixed-intention influence model). However, we have not tried to interpret the meaning of the value of these parameters. In addition, we wanted to keep the mixed-intention model exactly the same as the one used in the literature to relate and compare our results to it (Devaine, Hollard, and Daunizeau 2014a; Hampton, Bossaerts, and O’Doherty 2008). Obviously, it would have been possible to twist these models to avoid such correlations between parameters, but this would have been at the expense of the difficulty of the comparison with existing models.

Figure IX. Parameter recovery matrix (full parameter set). Each line shows the squared partial correlation coefficient between a given generated parameter and every recovered parameter (across 1000 simulations). Note that perfect recovery would exhibit a diagonal structure, where variations in each estimated parameter are only due to variations in the corresponding simulated parameter. Diagonal elements of the recovery matrix measure “correct estimation variability”, i.e., variations in the estimated parameters that are due to variations in the corresponding simulated parameter. In contrast, non-diagonal elements of the recovery matrix measure “incorrect estimation variability”, i.e., variations in the estimated parameters that are due to variations in other parameters. Strong non-diagonal elements in recovery matrices thus signal pairwise non-identifiability issues. Parameter 1 is the weight of first order PE (PE1), parameter 2 is the weight of second order PE (PE2), parameter 3 is the opponent’s temperature, parameter 4 is the slope of controller’s sigmoid (controller slope), parameter 5 is the center of controller’s sigmoid (controller’s bias), parameter 6 is the temperature of the competitive expert, parameter 7 is the temperature of the cooperative expert.

FREE PARAMETER	WEIGHT OF PE1	WEIGHT OF PE2	OPPONENT'S TEMPERATURE	CONTROLLER SLOPE (β)	CONTROLLER BIAIS (δ)	COMPETITIVE TEMPERATURE	COOPERATIVE TEMPERATURE
WEIGHT OF PE1	1						
WEIGHT OF PE2	0.07%	1					
OPPONENT'S TEMPERATURE	7.28%	7.56%	1				
CONTROLLER SLOPE (β)	0.35%	0.59%	4.06%	1			
CONTROLLER BIAIS (δ)	0.62%	3.15%	2.22%	17.18%	1		
COMPETITIVE TEMPERATURE	0.98%	1.93%	0.04%	0.38%	0.07%	NaN1	
COOPERATIVE TEMPERATURE	1.53%	4.13%	0.06%	0.01%	0.00%	0.04%	1
	0	0	1	1	1	1	

Table 4: Mean of common variance (in percentage) between free parameters across all subjects. First line is the mean value of the correlation. Second line is the p-value corrected with Bonferroni for multiple correlations.

We added the above table (Table 4) to the extended table section. Moreover, we added the parameter recovery to the extended figures and the corresponding explanation on **page 4 line 117** of the SI:

*'To check if our estimated parameters are identifiable, we generated 1000 datasets with the mixed-intention influence model playing against the choice sequences that the artificial agent generated. In this approach the mixed-intention influence model adapts to the choices of the AA, but the choices of the AA **do not** depend upon the choices of the mixed-intention influence model. We randomly drew free parameters from a normal distribution centered at the mean of the observed parameters of participants and with the standard deviation observed in the data. We then estimated the free parameters recovered when fitting the mixed-intention influence model on these new datasets. We then made linear regressions on each generative parameter with all recovered parameters as regressors to see if the variance of a specific generative free parameter is best explained by its corresponding recovered free parameter. A score of 1 mean that the variance of the generative parameter is totally explained by the variance of the recovered parameter. We can see in **Extended fig 4** that every generative free parameters is best explained by its corresponding recovered free parameter. Moreover, as can be seen in Extended Table 1, there are covariations between the free parameters of our generative model (mixed-intention influence learning). However, we do not interpret the value of these parameters. In addition, we wanted to keep the mixed-intention influence model exactly the same as that used in the literature to relate and compare our results to it. Obviously, it would have been possible to twist these models to avoid such correlations between parameters, but this would have been at the expense of the difficulty of making comparisons with the existing models.'*

c- How were individual models fit to the data?

Models were fit using the Variational Based method used in the VBA toolbox. All priors were set to their default values. This method allowed us to find free parameters that minimize the free energy of the model (Daunizeau, Adam, and Rigoux 2014).

We added this information in the methods **line 646, page 21**:

“Models were individually fit using the Variational Based method with the VBA toolbox. Every prior was set to its default value. With this method we were able to find free parameters that minimize the free energy of the model (Daunizeau, Adam, and Rigoux 2014).”

d- Why was depth 1 and 2 used for the k-ToM model, and depth 2 and 3 used for BSL?

Depths in these two different types of models do not represent the same thing. For the BSL model, the depth represents the length of the learnt sequences of others’ past choices. We chose depths of two and three because we found an effect of the two and three most recent trials on the strategy of participants in the model free analysis. Concerning the k-ToM model, the depth represents the level of recursion in theory of mind. Increasing from one level to the next level increases the computational cost a great deal, and the biological plausibility decreases (Devaine, Hollard, and Daunizeau 2014a). Thus, level 3 of k-TOM is unlikely to occur here.

e- While the Bayesian Model Selection makes sense, it would be helpful to provide metrics such as the pseudo-R2 and/or predictive of each model to give the reader an idea of the ‘absolute’ fit of the models.

Below we used the mean balanced accuracy as a metric of individual fit. The mean of the balanced accuracy is equal to 62% (range from 40% to 83%). This metric is more robust (conservative) than classical accuracy for behavioral choices (Brodersen et al. 2013).

We have added this information (but not the figure) in the Supp. Inf. on **line 87, page 3**.

Figure X. Violin plot of the balanced accuracy (bacc) across all subjects. The bacc represents the absolute fit of the model corrected for bias in binomial data (Brodersen et al. 2013).

15) Influence model – Equations 9 and 10: was the sign difference between the cooperative and competitive mode attributed by design (i.e. the sign changed on the first trial of a new mode?). This would assume that the participant knows there is a change in the player’s strategy and when this change happens, which is not the case. A model where the sign changes after the participant has learned that that the other player changed strategy would seem more realistic. Similarly in the k-ToM model, is the utility table learned or fixed according to the true payoffs?

We think that there may be a misunderstanding about the mixed-intention influence model. We apologize for our lack of clarity. As explained in the previous point 0) and 3), the two experts are working in parallel following their own walk. These 2 experts were continuously (not just at the time of change between modes) weighted according to their difference in reliability. Therefore, we are not assuming that participants *a priori* know there is a change in the other’s strategy when this change happens. In our models, the utility tables were fixed according to the payoff matrix.

We used the same methods for the mixed-intention k-ToM models. That is, we had 2 experts running in parallel, integrated by their weighted sum, according to their difference in reliability, and the utility tables were fixed.

We made this change in the Supplementary Information to be clearer, **page 12 line 371**:

For the mixed intention setting, we ran the competitive and cooperative models in parallel, avoiding the need for the payoff matrix to be learnt. On the first trial, each expert gives a prior probability that the other would choose the action a ($p_{coop,0}^* = p_{comp,0}^* = 0.5$), then each expert follows its own walk in generating, on each trial, the probability that the other will choose option a for each possible mode of interaction, P_{comp}^a and P_{coop}^a .

16) fMRI analyses:

a- GLM2: if a participant's bias parameter is really high or really low, this will result in a large majority of trials being classified as a single category. I can see in the methods that three participants were excluded because of that; however, in the remaining included participants what were the number of trials in each classified category?

The mean number of trials classified as competitive by participants was 81.4 (± 3.75 standard error). This represents approximately half of the trials (total number of trials = 163). We have now added this information in the text on **page 25 line 718**.

b- Logistic regression predicting P(stay) from timeseries and controller valence (l.270-282). Because the dlPFC and rTPJ/IPS regions used in this analysis were identified as showing stronger RPE signals in competitive versus cooperative, and RPE is calculated by fitting the model to choice data, it seems circular that the activity in those specific regions would in turn predict choice. I would remove this analysis, or find an unbiased way to run it (for example, run it across all voxels, similar to a searchlight procedure, to find in an unbiased way which regions of the brain significantly interact with the controller valence to predict P(stay)).

Following the reviewer's advice, we have now removed this analysis.

c- Functional connectivity analyses: because learning in the competitive condition is more difficult (as measured by a lower winning rate), and generally the brain seems more engaged in trials classified as competitive, could these results be explained by generally higher functional coupling in the brain in competitive trials, independent of the specific regions that were examined?

The reviewer raises a very interesting point. We therefore tested the hypothesis of a difference in either global or local efficiency in a network including all ROIs of the *conn toolbox* between the two classified modes as determined by the controller. We found no difference in global efficiency between the two modes ($T(26) = -0.75$; $p_{uncorrected} = 0.46$ when selecting the 15% most connected nodes, i.e., adjacency matrix threshold at cost = 0.15). This threshold was selected to enable the connectivity graph to be as close as possible to a "small world" when referring to both global and local efficiencies.

In addition, no node had a different local efficiency between the two modes of interactions, and thus no difference was found in general effective coupling.

We have now added these analyses on **pages 11 line 337**:

"This difference in connectivity could not be due to a general higher coupling during trials classified as competitive compared to trials classified as cooperative. Indeed, we tested for a difference in global efficiency between competitive classified and cooperative classified trials using graph theory and found no differences between the two modes of interaction (Global efficiency: $T(26)=0.75$; $p_{uncorrected} = 0.46$). Moreover, we also tested for differences between modes of interaction (competitive and cooperative) in local efficiency, and found that no nodes exhibited different local efficiency between trials classified as competitive and those classified as cooperative."

Moreover, we noted an error in our previous connectivity analysis, which was not 'functional' but 'effective' connectivity (regression vs correlation analyses). We have now replaced the word 'functional' by 'effective' in the manuscript.

17) Table S1: it's unclear how the AA's probability of choosing black is calculated from each of the 16

possible histories presented in the table.

As stated in the SI in the paragraph ‘Specification of the Artificial agent algorithm’, the artificial agent (AA) selected its target according to the probability for the player to choose a specific color after a given history. This probability is calculated based on the stored frequency (in a matrix as Table S1) that the participant chooses each target after each possible history of four elements composed by two choices and two outcomes (see table S1). Precisely, we counted the number of past occurrences of a given history (latest two choices and outcomes) and we counted the number of times the participant chose the black target after this same history. We then computed the ratio of the two to know the probability for the player to choose the black card in this given situation.

We call the probability of the player choosing the black card P_{black} . In Competitive trial blocks, the AA will choose the black card with probability $1 - P_{black}$, while in Cooperative trial blocks, it will choose the black card with probability P_{black} . A cooperative choice of the AA is defined as an AA choice to choose the most likely target chosen by the participant. Thus, even in competitive trial blocks, the AA can make cooperative choices. Since the algorithm needs to be initialized, we arbitrarily defined the first five trials as random (the AA plays the black target with probability 0.5). The possible combinations that are not encountered during these initialization trials were assigned with a probability of choosing the black target of 0.5.

18) There are quite a lot of typos/tense mistakes/missing third person ‘s’, etc, throughout the manuscript. It may be helpful to have the manuscript proofread more carefully.

We have now a native English speaker proofread our manuscript.

19) It would be great if the authors were willing to share their code and data, for open science and reproducibility practices.

We added all the codes and data on a github page (https://github.com/remiphilipp/Mixture_intention).

Reviewer #2 (Remarks to the Author)

Philippe and colleagues present a paper that explores a novel social decision-making task that switches rapidly between interactional contexts. In the competitive context two participants play “Hide and Seek”, whereas in the cooperative context they play a complementary task, in which identical actions are rewarded. They undertake a comprehensive model comparison of between different “modularized” social decision-making models and conclude that the Mixed-intention Influence model provides the best fit to the behavioral choice data. They then explore in neural correlates of RPEs in both interactional contexts in a model-based fMRI analysis and find ventral striatum (which also correlates with what the authors call reliability difference), rACC, lateral OFC and probably IPS as common PE effects in both contexts, whereas in right dlPFC, TPJ and IPS exhibited significantly stronger PE effects in the competitive contexts. Finally, they showed through a PPI analysis that the regions encoding the reliability difference (vStr, and vmPFC) are functionally connected to those regions that showed a stronger PE effect in the competitive context.

This a very comprehensive paper with and interesting task and lots of details in both the modeling of behavior and in terms of model-based fMRI. In general, the paper is well-written (although at times a bit long). It addresses a topic that should be also interesting to researchers outside the immediate field of social decision-making and therefore suited for the broad readership of Nature Communications. I would support a publication, if the authors could respond to the following questions and suggestions.

Computational modeling

I was a bit surprised by how the Mixed Intention Influence model computed the reliability of both controllers. In essence, the difference of the updated value signal in both controller is filtered through a bias-corrected softmax function to compute a linear weight for both controllers. This is quite different from how one would usually estimate the reliability of two signals through the precision of their estimates (as it is done in the HGF model or theoretically in the model-based/model-free RL literature (e.g. Daw et al., NN, 2005). I don’t see how this softmax has anything to do with the reliability of the controllers, it is only calculating which controller provides the higher estimate at the moment. The reliability estimate is a critical element in the model and also for the results, so this part needs to be better explained and justified (and maybe named differently).

We agree with the reviewer that the reliability estimation is a key aspect of the mixed-intention influence model. We have motivated and justified our choice of this reliability measure on point 9 of reviewer #1. Specifically, the softmax indicated in Fig 3.a calculates which controller provides the higher absolute estimate at time t. Intuitively, the closer the absolute value of DV is to 0, the more random the choice of the expert is. Thus, the absolute value of DV is a proxy of its reliability, and the difference of unsigned DV is a measure of difference in reliability.

Again, when we performed a Bayesian model selection among the 4 tested reliability signals, the model with the difference of unsigned DV as a controller is the best model (protected exceedance probability $pEP=0.83$) (see our answer to the point 8 reviewer #1).

We added the following paragraph in supplementary information **line 383 page 13**:

To motivate our definition of the reliability signal, we tested 4 different definitions of reliability signals for the winning model:

- $|DV_{competitive}| - |DV_{cooperative}|$ Difference of unsigned DV
- $|PE_{competitive}| - |PE_{cooperative}|$ Difference of unsigned PE (O’Doherty et al. 2021)
- $PE_{competitive} - PE_{cooperative}$ Difference of PE (Wan Lee, Shimojo, and O’Doherty 2014)
- $Entropy_{competitive} - Entropy_{cooperative}$ Difference of entropy: $-p * \log(p) - (1 - p) * \log(1 - p)$ (Charpentier, ligaya, and O’Doherty 2020)

When we performed a formal Bayesian model selection between these 4 reliability difference measures, the model with the ‘Difference of unsigned DV’ as controller was the best model (protected exceedance probability $pEP=0.83$). Moreover, the mixed-intention influence models with any one of the 4 reliability difference signals performs better than all the 15 other models presented in Fig 2.b. This demonstrates the robustness of the mixed-intention influence model, regardless of the chosen reliability difference measure.

Furthermore, the posterior predictives calculated from the Mixed Intention model (Fig 3b) are quite different from the model-free analysis of the participant data in Figure 2a. Overall, the model appears to be more sensitive to the a longer trial history than the participants as demonstrated by significant effects for trials t-3 for all 3 model-free indices. I think this difference and what it means in terms of the decision-making process deserves some discussion.

We have answered this point on point 10 from reviewer #1. We have made changes in the new Fig 3.b based on the results from our new analyses.

Neural data

Overall I find the model-based fMRI analyses convincing, but I suggest that the authors spend some time explaining that the ventral striatum is involved in the “reliability difference” at the time of choice (which right now seems more like a value difference), and in prediction error representations after the outcome. What does this dual involvement mean for the characterization of the network that supports social decision-making in different interactional contexts?

The more reliable the cooperative expert is compared with the competitive expert at the time of choice, the more likely participants should expect to win future social interactions, and the more engaged is the ventral striatum (Fig 4.c). At the time of choice, we observed engagement of this same brain region with the PE regardless of whether trials were classified as cooperative or competitive by the controller (Fig 5.a). Our interpretation of this finding, for social decisions, is that when one is more likely to be in a cooperative rather than in a competitive context, the ventral striatum, encoding PE at outcome, also anticipates desirable outcomes more strongly at the time of choice. This finding shows that the reliability difference and PE signals are used to adapt to fluctuating intentions of others during different interactional contexts. This is consistent with previous fMRI studies reporting that activation of the ventral striatum positively reinforces cooperation (Rilling et al. 2002). The striatum is reliably active in relation to others' rewards and contains cells that link one's own reward to self or others' actions (Báez-Mendoza and Schultz 2013). In addition, the mPFC, and especially the dorsal mPFC, has been shown to be critical for tracking what each agent is doing during series of either cooperative or competitive games (Klein-Flügge, Bongioanni, and Rushworth 2022; Wittmann et al. 2016), for tracking the beliefs of other individuals (Jamali et al. 2021; Park et al. 2019), as well as for encoding group behavior of others (Báez-Mendoza et al. 2021) and implementing a behavioral strategy in competitive social contexts (Konovalov et al. 2021).

Moreover, note that the reliability difference ($|DV_{competitive}| - |DV_{cooperative}|$) does not represent a classical value difference as is the case in the value-based decision making literature. This computation represents the difference of expectation between the predictions from the 2 experts (cooperative and competitive).

We have added the following sentences on **page 13 line 404** of the discussion:

The ventral striatum was involved in computing both the reliability difference between the competitive and cooperative experts at the time of choice (Fig 4.c) and the prediction error at the outcome (Fig 5.a). This ventral striatum engagement could reflect both that ventral striatum anticipates more social victories at the time of choice, when one is more likely to be in a cooperative rather than a competitive context, and that it encodes unexpected social victories. These findings indicate that the reliability difference and PE signals are both used to adapt to fluctuating intentions of others during different interactional contexts (Fig 3.a). This is consistent with previous studies reporting that ventral striatum is engaged with

mutual cooperation (Rilling et al. 2002) and with the fact that this region is reliably active in relation to others' rewards and contains cells that link own rewards to self or others' actions (Báez-Mendoza et al. 2021; Báez-Mendoza and Schultz 2013).

Minor points and clarifications

Figures

in Figure 5 there are some inconsistencies in the display threshold for different contrasts. in panel A the authors use a continuous color scale (probably) from $p < 0.001$ uncorr to $p < 0.05$ FWE, whereas in panel B we see a discrete color scale with only uncorrected p-values. This should be homogenized. Furthermore, it would be really helpful, if the violin plots in that figures could be put on the same y-axis and a visual indicator at the zero line. Based on this, it seems to me that rACC only shows a selective effect for competitive, but not for cooperative. Is this just a visual impression and can it be statistically verified that rACC also shows a PE effect in competitive blocks?

We have now homogenized Fig 5.a and 5.b by using the same statistical color scale. We have also used the same y-axis for Fig 5.c.

To answer the reviewer's point about the rACC, we already directly compared the PE for trials classified as competitive relative to those classified as cooperative in Fig 5.b. There was no difference in rACC activity in this direct comparison.

In Figure 6 it remains unclear whether all blue regions (vStr and vmPFC) are use as a single combined seed region or whether the PPI was calculated separately for each?

We apologize for the lack of clarity concerning these blue regions. We have now specified on **page 26 line 748** that the PPI was computed from the combined signal of these 2 regions combined as a single seed.

Brain-behavior correlation

The paragraph on page 9 (lines 270-283) are very convoluted and almost incomprehensible. When I read this paragraph I was unable to understand how this analysis was conducted. For instance, what is the "average of the weighted time series"? Why do you take 7 scans (longer than the trials actually lasts) of the weighted signal (what is that anyways?) and use that as a predictor, even though each trial has separable events (cue, decision, outcome) in which distinct decision-related computations are carried out. Averaging across them appears to be too coarse in this case. I think this whole paragraph has to be completely re-written.

As noted on point 16.b) of reviewer #1, we have now removed this analysis for the sake of clarity.

Data Sharing

I looked at the Github page form the project and was surprised to find only the VBA code for the winning model, but not for all the others. Were the other models also estimated using the VBA toolbox, or were different estimation methods used. In any case, please provide the code for other models in the comparison as well.

We added all the codes and data on a github page (https://github.com/remiphilipp/Mixture_intention).

Reviewer #3 (Remarks to the Author)

In this paper, the authors used a dyadic card matching game, where the participant will play against an artificial agent (referred to as AA; though they believe they are playing against a real human). The task involves making a decision between a red and black card by each of the players, and the participant's goal is to always match the cards. The goal of the computer agent switches between cooperation (matching cards) and competition (non-matching cards), in fixed block sizes of 10 and 13 trials, respectively. The participants are not informed about the switches in the strategy of the opponent, and indeed the experimenters did not even mention anything about the possible goals of the other opponent in the instruction phase of the experiment. The computer agent uses the recent history of the participants' game play to make a decision, given the agent's goal at the time.

Using model-free analyses (logistic regression over probability of staying), the authors showed that the participants' behavior is mainly influenced by the history of their outcomes (win/lose), and a so-called "cooperativity signature" which is approximately whether the agent's last action signalled a cooperative intent. The authors used a comprehensive suite of computational models, including heuristic models, as well as different variations of four classes of model-free and model-based algorithms that can be organized along two dimensions of Bayesian/non-Bayesian and mentalizing/non-mentalizing models. For each mentalizing model they considered three versions: competition-only, cooperation-only and mixed-intentions. The mixed-intention models are characterized by two expert modules, which compute the best choice given each competitive and cooperative intentions for the opponent and the final strategy is determined by weighting these strategies with their corresponding reliabilities. Using Bayesian model selection, they identified the best fitting model to be a non-Bayesian mentalizing model (referred to as influence model) which considers both cooperation and competition intentions for the opponent in determining the best strategy to play against such an opponent (mixed-intention). They demonstrated that the choice behavior of the mixed-intention influence model, as well as a hidden variable in the model which tracks the difference between the reliability of the predictions of the competition expert module and the cooperation expert module, are influenced by the same variables that affect the choice behavior of participants, identified in the logistic regression analyses.

Finally, using the mixed-intention influence model to build model-based GLMs, they have identified brain regions that track the difference in reliability between experts at the decision time, as well as the reward prediction error at the time of the outcome. They have also shown that right dlPFC, and the rTPJ are more correlated with reward prediction error in trials classified as competitive versus cooperative. Last but not least, they used two regions associated with the representation of reliability between experts (i.e., ventral striatum and vmPFC) as the seeds in a connectivity analysis. They showed that in trials classified as competitive compared to cooperative, the connectivity of these two seed regions is stronger with brain regions which were more involved in encoding reward prediction error in the same contrast of classified competitive versus cooperative trials.

Big picture comments

The authors propose to study how people "detect cooperative and competitive intentions of others" (according to the title of the paper). Yet the experimental design is not conducive to testing how people detect cooperative and competitive intentions. First, participants are never asked to report the intentions of the other player. So the most important variable — participants' detection of others intentions — must be recovered from a model of a different behaviour — card selection. Thus, from this experimental design, we can't separate the cognitive processes involved in the detection of intentions, from the cognitive processes involved in the formulation of a policy to best respond to those intentions.

We agree that the previous title of the paper was a bit misleading. Our goal was not to study how people "detect cooperative and competitive intentions of others". The goal of our paper is to study the neurocomputational mechanisms engaged in adapting to hidden and fluctuating intentions of others. We are now proposing a new title to avoid any confusion: 'Neurocomputational mechanisms involved in adaptation to fluctuating intentions of others'.

It is correct that participants were never asked to report the intentions of the other player. We deliberately chose not to ask participants what was the intention of the other, in order to avoid influencing their beliefs or consciously formulate the nature of the intentions of the other. The goal of our paper was not to separate the detection of intentions from the process of adapting to successive changes in hidden intentions. Our study focuses on characterizing the computations and brain systems engaged in adapting to hidden and fluctuating intentions of others.

Second, the intentions that the participants are asked to infer are atypical to the point of being completely inexplicable. The ‘other player’ (a simple artificial agent) switches every 10 or 13 trials, between ‘cooperative’ and ‘competitive’ modes. Although the authors argue plausibly that in real life, the same individuals sometimes cooperate and sometimes compete, this deterministic phase switching is not plausible. Also, the agent “learns to cooperate” by probability matching to the player’s history of behaviour, without distinguishing between rounds when the agent was cooperating vs competing. Because of this learning rule, the agent’s cooperative behaviour is much less predictable than a human partner who intends to cooperate, with much lower success rates. Because participants were not asked, we don’t know what participants thought was going on, to explain these confusing patterns of behaviour.

The intentions that the participants were asked to infer are not atypical: they consist of adapting to changing intentions (cooperative or competitive) of others. However, the neurocomputational mechanisms engaged in adapting to fluctuating intentions of others are far from a trivial problem. As noted in our introduction, so far previous studies have only focused on situations in which participants are explicitly informed about the nature of the interactions, either in a collaborative context alone or in a competitive context alone. For this reason, our design may appear somewhat artificial, because the switches between cooperative and competitive intentions were deterministic. However, and crucially, these switches between modes were NOT signaled. Such hidden switches between cooperative and competitive intentions are ubiquitous because many real life situations require coordination (e.g., in negotiations) while hiding one’s intentions to maximize one’s own benefits. In our experimental design, from the point of view of the participant, the other’s intentions were not signaled, furthermore, the deterministic change between intentions was not accessible. Thus, although we agree that in real life, the ‘phase switching’ between strategies is probabilistic rather than deterministic, we think that it is *first* necessary to study in detail the neurocomputational mechanisms engaged in adapting to fluctuating intentions of others (even if these fluctuations are deterministic for an external observer). Moreover, it should be noted that **our winning computational model would remain valid if the switches were probabilistic** because the mixed-intention influence model only uses choices and reward histories to adapt its choice’s policy, and does not use the real mode of interaction.

Moreover, the agent’s cooperative behavior was less predictable than a human partner who might very well signal that he is willing to cooperate. This was due to the fact that the AA learns continuously to predict the participant’s choices without distinguishing Cooperative and Competitive trials. However, it was precisely the purpose of our paradigm to study adaptation to hidden intentions and to avoid stereotypical behavior (e.g., always choose the same target). Our AA learning rule allowed a smooth continuity between the two interactive modes, thereby avoiding salient changes between the two modes of interaction. In addition, as the decision of the AA is probabilistic, it could make competitive choices in a Cooperative block (and conversely, it could make cooperative choices in a Competitive block). A cooperative choice of the AA is defined as an AA choice following the most likely target to be chosen by the participant (as defined in the Supp. Information). Again, such behavior allowed us to avoid a transient change of behavior and to hide the deterministic aspect of the changes between types of intentions.

Finally, if the regularity in the switches (every 10 or 13 trials) between modes was detected, we would expect a learning effect over time, reflected by a higher success rate for late relative to early blocks, both for the Cooperative blocks and for the Competitive blocks. This was not the case as demonstrated by a panel data logistic regression on the success rates in the Cooperative blocks as a function of block rank (no rank effect, effect size on success rate =0.14%, p=0.637). Similarly, there was no learning effect over blocks for the competitive blocks (no rank effect, effect size on success rate =0.06%, p=0.779). In addition,

similar logistic regressions on the probability to switch showed that there was no change in the switch strategy across Competitive blocks (size effect of Competitive block rank: 0.004, $p=0.095$) or across Cooperative blocks (size effect of Cooperative block rank: 0.001, $p=0.765$).

We have added the following sentences in the results section on **page 18 line 552**:

Importantly, if the regularity in the switches (every 10 or 13 trials) between modes was detected, we would expect a learning effect over time, reflected by a higher success rate for late relative to early blocks, both for the Cooperative blocks and for the Competitive blocks. This was not the case as demonstrated by a panel data logistic regression on the success rates in the Cooperative blocks as a function of block rank (no rank effect, effect size on success rate = 0.14%, $p = 0.637$). Similarly, there was no learning effect over blocks for the Competitive blocks (no rank effect, effect size on success rate = 0.06%, $p=0.779$). In addition, similar logistic regressions on the probability to switch showed that there was no change in the switch strategy across Competitive blocks (size effect of competitive block rank: 0.004, $p=0.095$) or across Cooperative blocks (size effect of Cooperative block rank: 0.001, $p=0.765$).

Setting design issues aside, what can we make of the results that the authors do report?

We believe that the clarifications provided above justify the choice of our design. We think that this design does allow us to investigate the brain computations underlying adaptation to hidden and changing (cooperative/competitive) intentions of others. Therefore, we do not see ‘design issues’ in our experiment.

For the behavioural data and model selection. The model selection analyses of the behavioural data are thorough and technically well done. Yet the models that are tested are not drawn from a set of plausible alternative psychological or neural hypotheses, particularly for human social cognition. For example, a whole set of models tested here assume that people do not adjust their play at all in response to the switching strategies of the agents — but these are not plausible alternative models of human cognition. It is not a plausible hypothesis about human cognition that people are completely inflexible or insensitive to changing play by their partners.

We agree with the fact that some of the previously tested models (Bayesian Sequence Learner (BSL); Reinforcement Learning (RL); Random Bias (RB); Win/Stay-Lose/Switch (WSLS)) are less plausible models of human social cognition in general. This is precisely what we argued in our introduction (lines 80-107). The reason why these models were included as alternative models to the ones which are ‘good’ models of social cognition (influence models; fictitious model, Active inference; k-ToM; Hierarchical Gaussian Filter (HGF)) is to actually demonstrate by rigorous model comparisons that the observed behavior is best recovered by models accounting for mentalizing (Fig 2b). Thus, the BSL, RL, RB and WSLS models provided control models. Including these models in our Bayesian model selection showed that they were insufficient to explain participant’s behavior when adapting to the changing hidden intentions of others.

To put the reviewer’s concerns to rest, we demonstrate the robustness of our previous results by a new analysis including 4 models taking into account only the changing intentions of others: mixed-intention influence model, mixed-intention 1-ToM, mixed-intention 2-ToM and the active inference model. This analysis demonstrates that our results are robust and remain identical: the mixed-intentions influence model is the best able to account for observed behavior.

Figure XI. Exceedance probabilities (EP) for model selection across 4 models accounting for changing intentions. EP is the probability that a given model is more frequent than the others in our observed population of participants.

In addition to these 4 models, we have performed another analysis also including mentalizing models considering only a single intention: influence cooperative, influence competitive, 1 and 2 ToM cooperative, 1 and 2 ToM competitive. This new analysis allowed us to test whether participants really adapted to the changing intentions of others, or, if their behavior could be explained by such simpler models. Again, the **results confirm the robustness of our original findings in which the mixed-intention model was the best model accounting for observed behavior.**

Figure XII. Exceedance probabilities (EP) for model selection across 10 models accounting for mentalizing. EP is the probability that a given model is more frequent than the others in our observed population of participants.

We added the following paragraph in the method section **lines 653 page 21**:

To demonstrate the robustness of our model selection we took a subset of models able to adapt to a switch of intention, i.e., mixed-intention influence model, mixed 1-ToM, mixed 2-ToM and active inference model (the active inferences model tries to minimize the distance between its generative model of the world and its perception of the environment via its sensors). The Bayesian model selection is robust and the most

frequent model is still the mixed-intention influence model ($pEP = 0.88$). In addition, if we also add to this set of models all mentalizing models (influence cooperative, influence competitive, 1 and 2 ToM cooperative, 1 and 2 ToM competitive), the mixed-intention influence model is still the most frequent ($pEP=0.86$).

Likewise, while the simpler Influence model is sufficient to explain the current data, the authors themselves suggest that more complex models like POMDP or k-TOM were not selected not because they are poor models of human cognition, but "because in our setting the only information that can be integrated by participants is their own choices, rewards and the history of the choices made by the other."

The mixed-intention influence model needs to be distinguished from the influence model. This latter model is not sufficient to explain observed behavior (see Fig.VI above) but it is still a valid model of strategic thinking (Griessinger and Coricelli, 2015). Our winning mixed-intention influence model is obviously not a simple model of human social cognition because: (a) it relies upon 2 influence models; (b) it requires a controller able to weigh the relative reliability between these 2 influence models.

Importantly, POMDP and k-ToM needs the model of the "game" which determines the dynamics of interactions. For example, when the game is cooperative, a POMDP agent (Khalvati, Park, et al. 2019) and a k-ToM agent (Khalvati, Mirbagheri, et al. 2019) assume a transition function in which a cooperative action increases the cooperation in others. While the amount of this increase is a free parameter in the model, the main concept of "encouraging cooperation" or "conformity" as in Khalvati et al. 2019 is hard-coded in the model. This dynamic would be totally reversed in a competitive setting. In fact, changes in the nature of the response to actions play a major role in the current task. K-ToM also requires the reward of others, whether explicitly during the game, or in the form of a prior obtained from previous life experience/common knowledge. This is also not available in the current task.

The reason why the POMDP or k-TOM were not selected is because our model selection analysis demonstrated that these 2 models were simply not the best models. This does not mean that these models are not valid models of human social cognition. In our previous discussion, we stated that it is because in our 'setting the only information that can be integrated by participants is their own choices, rewards and the history of the choices made by the other'. This explanation intuitively justifies why the POMDP and k-ToM were not as good as the mixed-intention influence model in the context of our task. We acknowledged that these models are more sophisticated mathematically than the mixed-intention influence model, but this does not necessarily mean that the more parsimonious model would perform more poorly to account for the observed behavior. As explained in our discussion, we think that the POMDP and k-ToM models could not profit from their ability to reproduce more flexible behavior because the nature of the social interaction was never explicitly signaled and because the rewards of the other were not observed (Castro-Rodrigues et al. 2022; Kim et al. 2019).

In addition, we have added the following justifications and citations to answer this point in the discussion on **page 14 line 428**:

'In contrast, the structure of the task is not directly observable because the nature of the social interaction (i.e, cooperative or competitive) is never explicitly signaled to participants, and the rewards of the other are not observed. Thus, the resulting complexity and uncertainty might result in decreasing the use of model-based strategies, as suggested recently (Castro-Rodrigues et al. 2022; Kim et al. 2019), and in increasing the use of simpler trial and error learning.'

So what scientific conclusion is supported by the model selection? Possibly, the behavioural and model selection component of the paper might have been included mainly to set up the regressors for the fMRI analyses, in which case the authors should say so.

The reviewer is suggesting that we have included 'non-social learning' models in our analysis to artificially inflate the performance of the winning (mixed-intention influence model) model. This is clearly

not the case. First, as noted with the new analyses reported above, the model selection is robust to removing ‘non-social learning’ models. Second, we have used the model selection for two purposes: (a) to test two alternative hypotheses: existence of a unique process adapting to the two modes of interactions (e.g., competitive influence model alone, 2 ToM competitive model, ...) or of two parallel processes working together (mixture of experts); (b) to use computational variables of the winning model to assess whether those variables are encoded at the brain system level.

We have made the following changes on **page 9 line 284** to explicitly state the methodology used with our computational models:

‘Having characterized the computations of the dynamical adaptation to changing intentions of others, we then tested whether latent variables of these computations are encoded at the brain system level.’

From the discussion of the paper, I infer that one hypothesis is that "social behavior can be explained by a Controller Theory according to which cooperative/competitive social behavior results from the interaction of multiple systems, each proposing possible strategies for action." However, the model fit in this paper does not test this hypothesis. This stated hypothesis predicts that on every trial, two systems in the participants’ brain each represent an independent expected reward (and later a prediction error), and a separate arbitration system then weights the two to choose an action. To test this hypothesis, the authors should have built a GLM with the predicted reward, at the decision phase, and the expected reward (at the outcome phase) from each system, over all trials regardless of how they are classified; and a third expected reward (and RPE) from the weighted sum of the two based on reliability. Then they could test whether there are distinct RPEs computed on every trial, for each system and the arbitrated value.

First, as noted in the introduction (lines 108-112), our hypothesis was to test the predictions of different families of learning models against one another, investigating not only non-Bayesian vs Bayesian models and non-mentalizing vs mentalizing models, but also a mixture of models deploying an arbitration process whereby the influence of attributing intentions to others is dynamically modulated depending on which type of intention (i.e. cooperative vs competitive) is most suitable to guide behavior at a given time. To do so, at the algorithmic level, we have shown, using robust Bayesian models selection and generative analyses (reviewer #1, answer to point 14), that this controller theory (i.e., mixed-intention influence model) is the best framework to explain participants’ behavior.

Second, we did not assume, as presumed by the reviewer, that there are separate brain regions for each expert and for their weighted sum. This is because: (1) the algorithmic and the brain system levels have to be distinguished (Lockwood, Apps, and Chang 2020). Separate expert systems can be observed at the behavioral and computational levels for cooperation and competition, but it does not necessarily follow that separate (distinct) brain systems have to be assumed for these experts; (2) we did not study experts reflecting different cognitive processes, as investigated previously (e.g., model-based vs model-free, or emulation vs imitation) (Charpentier, Iigaya, and O’Doherty 2020; Wan Lee, Shimojo, and O’Doherty 2014). Instead, the **same** cognitive process (i.e., influence learning) was assumed to be implemented by our experts. Thus, we did not assume that there are separate brain regions for the cooperative and the competitive experts. Indeed, contrary to mixture of models theories which weigh experts implementing different cognitive processes, in our case, the two experts, each using different priors, implement an identical process. This is a key conceptual distinction which extends the mixture of models to a more general view in which the experts are differentiated not by their cognitive processes but by their priors.

In fact, as noted by reviewer 1, point 5, when investigating the brain system(s) engaged by the 2 experts, we observed common brain networks encoding the cooperative and the competitive experts. This finding was obtained in an ANOVA including DV and PE of each expert. The experts showed common brain activity in the ventral striatum for the decision value of staying on the same target ($P=0.007$ FWE cluster corrected, threshold at $p<0.001$ at the cluster level, Fig IV). When directly comparing between the decision value of the 2 experts, we did not find any separate brain region ($P>0.95$ FWE cluster corrected, threshold at $p<0.001$ at the cluster level). For the reward prediction error (PE), the two experts showed common activity in the ventral putamen, the anterior medial PFC, posterior cingulate cortex and the lateral OFC (Fig V) (all $P_s<0.02$ FWE cluster corrected, threshold at $p<0.001$ at the cluster level). These results, displayed in

Fig V, were obtained by building, for the mixed-intention influence model, a new GLM that included, at the first level, decision values (DV) from the competitive expert, decision value from the cooperative expert, PE from the competitive expert and PE from the cooperative expert. Specifically, there were 4 onsets including the time of the cue presentation (cards on screen), participant's button press, AA's choice and feedback time. At the time of the cue onset, the parametric regressors were the decision value for staying on the same target generated by the competitive expert, and the DV for staying on the same target generated by the cooperative expert. Similarly, at the time of feedback, the regressors were the reward prediction error (PE) generated by the competitive expert and the PE generated by the cooperative expert.

For the sake of completeness, we also ran the GLM proposed by the reviewer. This GLM is the same as that reported above with DVcoop, DVcomp, PEcoop, PEcomp as regressors, plus 2 additional regressors: $DV_{total} = P_{coop} * DV_{coop} + (1 - P_{coop}) * DV_{comp}$ and $PE_{total} = PE_{coop} + PE_{comp}$. However, this GLM could no longer identify the two experts at the brain system level, probably because the weighted sum competes for variance with the previous regressors, and because the 2 experts' decision values are correlated (even if this is not imposed by the structure of the mixed-intention influence model).

To answer this point, we made the following changes in the introduction, Supplementary information and discussion on **pages 5, 6 (SI), 14**:

Introduction **page 5 line 122**:

'We found that the behavior in our task was best accounted for by a mixture of influence models, referred to as mixed-intention influence Model. At the algorithmic level, two expert systems implemented an identical influence learning process, working together to make strategic decisions. These two experts differ only by their priors, one expert assessed competitive intentions and the other assessed cooperative intentions, while a controller weighted between these experts according to their relative reliabilities. Each expert system uses a classic RL algorithm complemented with a mentalizing term to infer the other's actions. This mixed-intention influence model accounted for observed behavior in which the other's goal is often only partially congruent with one's own, allowing us to explain a continuous range of behavior between pure cooperation and pure competition. At the brain system level, we hypothesized that the two experts would engage similar brain regions because they implement an identical process, each using different priors. This hypothesis extends the mixture of models to a more general view in which the experts are differentiated not by their cognitive processes (model based/model-free; emulation/imitation...)(Charpentier, Iigaya, and O'Doherty 2020; Wan Lee, Shimojo, and O'Doherty 2014) but by their priors. Indeed, in the current study, the two experts implemented an identical process (i.e., influence learning), each using different priors. In addition, we hypothesized that the signal that computes the reliability difference between the two experts is encoded in the brain. A brain network including the ventromedial prefrontal cortex (vmPFC) and the ventral striatum tracked the reliability difference signal from the controller.'

Supplementary information on **page 6 line 178**:

'To investigate the brain system engaged by the 2 experts, we performed an ANOVA including DV and PE of each expert. The experts showed common brain activity in the ventral striatum for the decision value of staying on the same target ($P = 0.007$ FWE cluster corrected, threshold at $p < 0.001$ at the cluster level, Extended Figure 8.a). When directly comparing between the decision value of the 2 experts, we did not find any other brain region ($P > 0.95$ FWE cluster corrected, threshold at $p < 0.001$ at the cluster level). For the reward prediction error (PE), the two experts showed common activity in the ventral putamen, the anterior medial PFC, posterior cingulate cortex and the lateral OFC (Extended Figure 8.b) (all $P_s < 0.02$ FWE cluster corrected, threshold at $p < 0.001$ at the cluster level). These results, were obtained by building, for the mixed-intention influence model, a new GLM including, at the first level, decision values (DV) from the competitive expert, decision values from the cooperative expert, PE from the competitive expert and PE from the cooperative expert. Specifically, there were 4 onsets including the time of the cue presentation (cards on screen), participant's button press, AA's choice and feedback time. At the time of the cue onset, the

parametric regressors were the decision value for staying on the same target, generated by the competitive expert, and the DV for staying on the same target generated by the cooperative expert. Similarly, at the time of feedback, the regressors were the reward prediction error (PE) generated by the competitive expert and the PE generated by the cooperative expert’.

Figure XIII. Common ventral striatal region encoding decision value of the cooperative and competitive experts.

Figure XIV. Common brain regions encoding prediction error of the cooperative and competitive experts included the ventral putamen, the anterior medial PFC, posterior cingulate cortex and the lateral OFC.

Discussion on **page 14 line 450**:

Nevertheless, this finding does not indicate that each expert is encoded in separate brain regions. Indeed, additional analyses demonstrated that the computational signals (decision value and PE) of the two experts commonly engaged similar brain regions (see Supp. Information and Extended Fig 8).

Instead, the authors claim that the VMPFC and ventral striatum represent the ‘control system’ arbitrating between the two systems. But the actual evidence in the paper shows that these regions are correlated, across trials, with a regressor composed of (a) highest weighted, whether the participant won on the previous trial, and (b) less weighted, whether the agent’s action on the previous trial singled a cooperative intention. I honestly don’t see how this pattern of activation supports the argument that the VMPFC and ventral striatum are involved in “arbitration”. In the fMRI GLM, this regressor competes for variance with a regressor for expected reward — it would be helpful to see what are the unique components of each of those regressors when controlling for one another.

We believe that there may be a misunderstanding about the findings displayed in Fig 4.a and Fig 4.c. Indeed, the Beta weights displayed in fig 4.a refer to the logistic regression of the latent variable Δ of our mixed-intention influence model. These beta weights are NOT the beta from the GLM corresponding to the vmPFC or ventral striatum activity. Thus the betas from Fig 4.a do NOT reflect that the vmPFC and ventral striatum activity are correlated, across trials, with a regressor composed of: ‘(a) highest weighted, whether the participant won on the previous trial, and (b) less weighted, whether the agent’s action on the previous trial singled a cooperative intention’. Concerning the Fig. 4.c, this figure directly indicates that the vmPFC and ventral striatum correlate with the controller (Δ) that arbitrates between the two experts. Thus,

the interpretation of Fig 4.c (vmPFC and ventral striatum are involved in an “arbitration” process) is not a consequence of Fig. 4.a.

In addition, following the reasoning of the reviewer, we checked whether the correlation between vmPFC/striatal BOLD signal and the reliability difference (Δ) could be driven mainly by the previous winning interaction and by the Cooperativity signature of AA or whether this correlation is due to higher level features caught by the controller of our model. To do so, we built a new fMRI GLM including 4 onsets: the time of the cue presentation (cards on screen), participant’s button press, AA’s choice and feedback time, with 4 non-orthogonalized parametric regressors at the time of the cue presentation: the reliability difference (Δ) and the regressors at t-1 of the linear regression shown in Fig 4.a (Cooperativity signature of AA(t-1), previous winning interaction (t-1) and participant switch(t-1)). The results of this new analysis are similar to our previous findings (**Fig XV**), i.e., the vmPFC/ventral striatal activity correlates with the reliability difference (Δ). No cluster significantly correlated with the other 3 regressors (Cooperativity signature of AA(t - 1), previous winning interaction (t - 1) and participant switch (t - 1)). This analysis confirms that the **vmPFC and ventral striatum represent the ‘control system’ arbitrating between the two systems**, and not previous winning outcome or cooperativity signature of AA.

Please note that to avoid any confusing interpretation, we have also changed the terminology ‘cooperativity signature’ in the figures and text to ‘Cooperativity signature of AA’.

We have now changed the Methods and Results concerning GLM1 to report this new GLM1 analysis controlling for winning outcome and cooperativity signature of AA. We have also replaced previous Fig 4c with the new results shown in Fig XV and we have changed the description of these results on **page 10 and 24**:

Methods on **page 24 line 701**:

‘Specifically, in GLM1, there were 4 onsets, including the time of the cue presentation (cards on screen), participant’s button press, AA’s choice and the feedback time. Parametric regressors were the difference in reliability Δ , the Cooperativity signature of AA (t - 1), the previous winning interaction (t - 1) and the participant switch (t - 1) at the time of the cue onset. At the outcome phase, parametric regressors were the reward prediction error (PE) as well as Δ to control for the effect of the believed intention of the other on the encoding of PE by the brain’.

Results on **page 10 line 289**:

First, we constructed a GLM (GLM1) to identify brain regions tracking the arbitration process (i.e., Δ : signed reliability difference, reliability for cooperation minus that of competition) between the two experts (one for cooperation, the other for competition). We added the reliability difference Δ as parametric regressor at the decision stage, as well as the previous winning interaction, the previous switch and Cooperativity signature of AA of the previous trial, as non-orthogonalized parametric regressors, to allow them to compete for the variance. At the outcome phase, we added the reward prediction error as a parametric regressor and we controlled for the effect of the other’s intention by adding Δ as a non-orthogonalized regressor. The bilateral ventral striatum (x,y,z = 14,12,-2 and x,y,z = -13,7,-6), and vmPFC (x,y,z = 6,46,-6) tracked the difference in reliability between experts (Δ) at the decision time (all results at p<0.05 whole-brain family-wise error (FWE), Fig. 4c). Thus, activity observed in the vmPFC /ventral striatum (Fig 4c) represents the reliability difference itself, and cannot be explained by the previous outcome alone or the Cooperativity signature of AA. Bilateral dorsal striatum (DS; x,y,z = 17,6,-12 and -14,3,-11), bilateral orbitofrontal cortex (OFC; x,y,z = 44,36,-14 and -44 52 8), posterior cingulate cortex (PCC; x,y,z = 2,-34,38),

and bilateral angular gyrus ($x,y,z = 45,-30,46$ and $-54,-62,39$) encoded the reward prediction error at the outcome time ($p < 0.05$ FWE, Fig. 5a)

Figure XV. Reliability difference (Δ) at the time of decision, when making Δ compete for the variance with previous winning interaction, previous switch and Cooperativity signature of AA of the previous trial.

FWIW, I do think it would be interesting to test whether the VMPFC and ventral striatum (or other brain regions) are independently predicted by whether the participant won on the previous trial, and by whether the agent signalled an intent to cooperate on the previous trial. The authors specific algorithmic model suggests that these two parameters are composed into a single weighted sum, but there are many alternative representations that could implement the same computation, including representing each of these parameters separately. One possible test of the authors' specific model, for example, would be whether the relative weight of those two features, in one subset of one participant's data, better predicts the activity in the brain in the other half of the same participant's data, compared to when shuffling across participants.

Again, we think that this comment is based on a misunderstanding of the results presented in Fig 4a and 4c. Indeed, Fig 4a represents the linear regression of the reliability difference Δ . This reliability difference is first extracted from the winning model (i.e., mixed-intention influence model, based on Bayesian selection), and then only, to explain part of Δ variance, we regressed it against the same regressors as those used in the behavioral analysis of the probability to stay (Fig 2a). Thus, the logic of our analysis, shown in Fig 4a, was to better understand how Δ , obtained from the winning model, is linked to observable behavioral features (Cooperativity signature of AA, previous winning interaction, switch).

Furthermore, and importantly, we did not claim that our algorithmic model (i.e., mixed-intention influence model), is based on a weighted sum of these features. We only claimed that part of the Δ variance (60%) is explained by a weighted sum of these 3 features. Again, Fig 4.c reflects the correlation between brain activity and Δ (used via a sigmoid function to weight the two experts), obtained from the computational mixed-intention influence model.

The modifications we made to GLM1, reported in the previous point, now allow us to exclude the interpretation that vmPFC and ventral striatum activity could be explained by the previous outcome alone or by the Cooperativity signature of AA.

Honestly, one thing that's extremely confusing throughout the paper is that sometimes they compute functions of the probability of choosing black, sometimes of the probability of 'staying' on the same choice as last time, and sometimes the probability that the agent is cooperating, but these are mixed up in a confusing way. The authors need to be much clearer about when the probabilities are in terms of choosing black. For example, Figure 3c baffles me. Prediction error is $\text{Received_reward} - \text{Predicted_reward}$, so should be high when things turn out better than expected; the Figure shows that — just in cooperative trials — when the outcome was WORSE than expected participants are highly likely

to STAY on their current card choice. That makes no sense, and it is a symptom of my trouble with this paper that I cannot even guess which kind of problem this is.

We apologize for the lack of clarity using different terminologies for the probabilities. In the revised version of the paper, we now always use the probability to switch target when referring to the participants' behavior. We use the probability of choosing the black card only when referring to the behavior of the artificial agent and the behavior of models. We have decided to keep this terminology for the Artificial Agent to accurately report the strategy of the AA's algorithm (see Supplementary Information, description of computational models). Concerning the probability that the other is cooperating, this probability is generated by the winning mixed-intention influence model. We have clarified this point (**lines 233-235** of SI, legend of fig.4b).

To answer the point concerning Fig. 3.c concerning the link between the predictive effect of PE on the probability to stay on the same target, we have previously discussed it in the Discussion on lines 452-458. When the PE is negative, we argued that the higher probability to stay on the same card in trials classified as cooperative reflects a signal sent to others to promote the willingness to cooperate despite having obtained no reward (Fig 3c). The fact that participants stay more on the same target during trials classified as cooperative, after a high negative prediction error, (i.e., unexpected social defeats) is surprising, therefore, we previously added the extended figure 2 (Supp. Information). This figure shows that if we separate trials not into "classified cooperative" or "classified competitive" but rather into actual Cooperative or Competitive trials, we no longer observe this effect on the probability to stay on the same target. Thus, there is a crucial distinction between trials classified as cooperative by the controller, and real trials in which the AA cooperates. This distinction may be at the source of the misunderstanding of the reviewer: a negative PE (outcome worse than expected) is associated with a lower probability to stay on the same target for real cooperative trials only (Extended Fig. 7), and not for trials classified as cooperative by the controller (Fig. 3c). Thus, Fig. 3c simply reflects that the controller at time t is able to determine how PE at time t-1 is used for current decisions at time t. Our interpretation of this effect is that enduring a cost (here staying on the same target for trials classified as cooperative by the controller) is a key feature of successful coordination since an individual who wants to trigger reciprocity is willing to incur a cost to promote cooperation from another.

To clarify this misunderstanding, we have changed the description of our results on **page 8 line 247**: In addition, we also performed another logistic regression analysis using the same variables and the actual mode of interaction (i.e., Competitive block trials *versus* Cooperative block trials), rather than the classified mode of interaction as made by the controller. We did not find the same interaction effect when we compared real Competitive and Cooperative block trials ($Block\ type_t * rPE_{t-1}: estimate = 0.003, p = 0.56, Block\ type_t: estimate = 0.03, p = 0.229, and PE_{t-1}: estimate = 0.14, p < 0.0005, \chi^2$ test; **Extended Fig.7, see SI**), which shows that the classified intentions, but not Competitive or Cooperative blocks, affected the use of prediction error.

Note that the PE used in this analysis is the $Reward(t) - P(t)$, with $P(t) =$ Integrated value computing the weighted sum of the probability to win according to the two experts (figure 3a).

However it helped me understand another puzzle in the paper, which is that RTPJ and DLPFC are recruited for high prediction errors in competitive trials. Assuming this is the same RPE shown in Fig 3, and 'high positive prediction errors' mean that participants are more likely to switch cards on the next trial — I am going to assume high RPE is unintuitively defined as when the outcome was WORSE than expected, not better. That is, I am assuming that the more participants were surprised by a loss, on a trial classified as competitive, the more activity in these brain regions. I'm not sure what the authors think this means (they conclude it shows "a differentiation in the implementation and use of the outcome of the social interaction as a function of the classified interaction", which is a restatement of the results.) Thinking about this task, I imagine these may be the first one or two trials after the model switches from classifying a trial as cooperative (generating higher expected rewards) to competitive (because of lower achieved rewards). Thus one possible interpretation of the results is that RTPJ and DLPFC activity are involved

when participants infer that the agent is now playing more competitively — or see the agent as successfully catching or tricking them. I am not sure whether the authors would consider this an alternative interpretation, or a restatement of their own interpretation?

It is **incorrect** that ‘high RPE is unintuitively defined as when the outcome was worse than expected’. In fact, this is just the opposite: high positive PE reflects unexpected reward delivery (i.e., outcome better than expected). This positive PE corresponds to the right side of the x-axis from Fig 3.c.

Also, it is not correct that the PE displayed in Fig 3c corresponds to the PE shown in Fig 5. Indeed, in Fig 3c the PE at $t - 1$ predicts the probability to stay on the same target, based on the classification at time t , while Fig 5b represents PE at time t for trials classified as competitive relative to those classified as cooperative at time t . Thus, the difference of switch probability according to the previous PE (especially for negative PE) and the current classification of the trial (competitive or cooperative) (left portion of Fig 3.c. corresponding to a negative PE) **does not** correspond directly to the fMRI results shown in Fig 5.b. We apologize for the lack of clarity which may have led to this misunderstanding.

We have now specified:

“It should be noted that this differential PE coding reflects the classification of the current trial into cooperative vs. competitive (fMRI results Fig.5b), whereas the behavioral results shown in Fig. 3c shows that the effect of rPE on behavior depends upon how the next trial might be classified.” on page 10 line 317

Similarly, it is incorrect to link Fig 3.c with Fig 5.b when stating ‘the more participants were surprised by a loss, on a trial classified as competitive, the more activity in these brain regions’. The statement we made about Fig 5.b, is that for trials classified as competitive, the PE is more strongly encoded in the rTPJ and DLPFC relative to trials classified as cooperative. Again, the purpose of Fig. 3.c, as for Fig. 3.b, was to link the prediction of the model with the observed behavior. We should not try to link Fig. 3.c with Fig 5.b because they do not relate to the same time step. We believe that part of the misunderstanding might be due to the lack of clarity in our description of Fig 3.c which concerns the correlation between Probability to stay at time t depending on the classification on trial t and RPE at $t-1$. We have now added this important information in the new Figure 3 and its legend.

To clarify the meaning of our sentence “a differentiation in the implementation and use of the outcome of the social interaction as a function of the classified interaction”, we briefly summarize the functioning of our model. First, each expert computes a decision value. Second, the controller estimates the reliability of each expert and computes a probability to be in a cooperative interaction. Third, an integrated value is computed and a choice is made. Finally, a prediction error is computed at the outcome. Fig 5b shows that the brain system engaged in computing PE depends on the previous classification of the trial.

We are not clear about the reasoning from the reviewer to come up with the ‘interpretation of the results that RTPJ and DLPFC activity are involved when participants infer that the agent is now playing more competitively — or see the agent as successfully catching or tricking them’. However, our results actually show that this statement is not correct, because they indicate that the RTPJ and DLPFC activations depend on the already inferred intention of the other. The PE computed at the outcome does not define the classification of the current trial but is used to compute the classified intention at the next trial. Thus, one possible interpretation of the rTPJ/DLPFC results is that when the other’s intention seems to be competitive, participants will encode the prediction error more strongly and use this to make the next decision. However, when the previous interaction leads to a higher reliability of the cooperative expert, the participant has a higher probability to stay on the same target than on the previous trial. Then at the time of outcome he does not encode the integrated prediction error in these regions, which are engaged in theory of mind.

We have now rephrased our interpretation in the discussion **page 14 line 441** to be more precise.

In the context of our mixed-intentions task, when participants make a choice driven by the competitive expert, which is judged to be more reliable than the cooperative expert, the following PE is specifically encoded in the DLPFC and rTPJ/IPS. Thus, priors on the context of interaction (competitive or cooperative) during the decision time, modulate the way the reward prediction error will be implemented at the time of outcome.

In sum, this review is getting unpardonably long. I have tried very hard to evaluate the scientific claims about the human mind and brain that these authors are making with the results.

Minor points:

- In figure 1b, they should make it clear that these are the payoff matrices of the AA, not the participant. The payoff matrices are the payoff of both the participants (bottom left of each square), and the Artificial Agent (top right of each square). We have now made it clearer in the legend by adding : “Participants payoff (bottom left of each square) and the Artificial Agent payoff (top right of each square).”

- Isn't the analysis of predicting P(stay) as a function of time series of activity in dlPFC and rTPJ a circular analysis? Because dlPFC and TPJ correlate with PE, PE affects P(stay) and so the activity of dlPFC and TPJ should affect P(stay) by definition. Are they trying to make causal statements using this analysis? We have now removed this analysis.

- What is the claim for extended figure 3b?

The claim was, as in Fig 2.B of the classical Hampton et al. (2008) paper, to show that the mixed-intention influence model not only best explains the observed behavior, but also best explains the observed BOLD signal observed as compared to a simpler RL model. Thus, we can assert that regions best explained by the mixed-intention influence model encode a higher level of information that the simpler RL model does not capture.

- In general, the text can be improved language-wise; it requires lots of edits. Also there are errors in the equations of supplementary material.

We have edited the English language carefully and have also checked the equations from the Supp. Information.

Bibliography

- Báez-Mendoza, Raymundo, Emma P Mastrobattista, Amy J Wang, and Ziv M Williams. 2021. "Social Agent Identity Cells in the Prefrontal Cortex of Interacting Groups of Primates." *Science* 374(6566): eabb4149.
- Báez-Mendoza, Raymundo, and Wolfram Schultz. 2013. "The Role of the Striatum in Social Behavior." *Frontiers in Neuroscience* 7(7 DEC): 1–14.
- Brodersen, Kay H. et al. 2013. "Variational Bayesian Mixed-Effects Inference for Classification Studies." *NeuroImage* 76(May 2014): 345–61. <http://dx.doi.org/10.1016/j.neuroimage.2013.03.008>.
- Castro-Rodrigues, Pedro et al. 2022. "Explicit Knowledge of Task Structure Is a Primary Determinant of Human Model-Based Action." *Nature Human Behaviour* 6(8): 1126–41.
- Charpentier, Caroline J., Kiyohito Iigaya, and John P. O'Doherty. 2020. "A Neuro-Computational Account of Arbitration between Choice Imitation and Goal Emulation during Human Observational Learning." *Neuron* 106(4): 687–699.e7. <https://doi.org/10.1016/j.neuron.2020.02.028>.
- Daunizeau, Jean, Vincent Adam, and Lionel Rigoux. 2014. "VBA: A Probabilistic Treatment of Nonlinear Models for Neurobiological and Behavioural Data." *PLoS Computational Biology* 10(1).
- Devaine, Marie, Guillaume Hollard, and Jean Daunizeau. 2014a. "The Social Bayesian Brain: Does Mentalizing Make a Difference When We Learn?" *PLoS Computational Biology* 10(12).
- . 2014b. "Theory of Mind: Did Evolution Fool Us?" *PLoS ONE* 9(2).
- Hampton, A. N., P. Bossaerts, and J. P. O'Doherty. 2008. "Neural Correlates of Mentalizing-Related Computations during Strategic Interactions in Humans." *Proceedings of the National Academy of Sciences* 105(18): 6741–46. <http://www.pnas.org/cgi/doi/10.1073/pnas.0711099105>.
- Hill, Christopher A et al. 2017. "A Causal Account of the Brain Network Computations Underlying Strategic Social Behavior." *Nature Neuroscience* 20(8). <http://www.nature.com/doi/10.1038/nn.4602>.
- Jamali, Mohsen et al. 2021. "Single-Neuronal Predictions of Others' Beliefs in Humans." *Nature* 591(7851): 610–14.
- Khalvati, Koosha, Saghar Mirbagheri, et al. 2019. "A Bayesian Theory of Conformity in Collective Decision Making." *Advances in Neural Information Processing Systems* 32(NeurIPS).
- Khalvati, Koosha, Seongmin A Park, et al. 2019. "Modeling Other Minds: Bayesian Inference Explains Human Choices in Group Decision-Making." *Science Advances* 5(11): eaax8783. <http://advances.sciencemag.org/content/5/11/eaax8783.abstract>.
- Kim, Dongjae, Geon Yeong Park, John P. O'Doherty, and Sang Wan Lee. 2019. "Task Complexity Interacts with State-Space Uncertainty in the Arbitration between Model-Based and Model-Free Learning." *Nature Communications* 10(1). <http://dx.doi.org/10.1038/s41467-019-13632-1>.
- Klein-Flügge, Miriam C., Alessandro Bongioanni, and Matthew F.S. Rushworth. 2022. "Medial and Orbital Frontal Cortex in Decision-Making and Flexible Behavior." *Neuron*.
- Konovalov, Arkady, Christopher Hill, Jean Daunizeau, and Christian C. Ruff. 2021. "Dissecting Functional Contributions of the Social Brain to Strategic Behavior." *Neuron*: 1–15. <https://doi.org/10.1016/j.neuron.2021.07.025>.
- Lockwood, Patricia L., Matthew A.J. Apps, and Steve W.C. Chang. 2020. "Is There a 'Social' Brain? Implementations and Algorithms." *Trends in Cognitive Sciences* 24(10): 802–13. <https://doi.org/10.1016/j.tics.2020.06.011>.

- O'Doherty, John et al. 2021. "Why and How the Brain Weights Contributions from a Mixture of Experts." : 1–18.
- Park, Seongmin A., Mariateresa Sestito, Erie D. Boorman, and Jean Claude Dreher. 2019. "Neural Computations Underlying Strategic Social Decision-Making in Groups." *Nature Communications* 10(1): 1–12. <http://dx.doi.org/10.1038/s41467-019-12937-5>.
- Rilling, James K, David A Gutman, Thorsten R Zeh, and Giuseppe Pagnoni. 2002. "A Neural Basis for Social Cooperation." *Neuron* 35: 395–405.
- Wan Lee, Sang, Shinsuke Shimojo, and John P. O'Doherty. 2014. "Neural Computations Underlying Arbitration between Model-Based and Model-Free Learning." *Neuron* 81(3): 687–99. <http://dx.doi.org/10.1016/j.neuron.2013.11.028>.
- Wittmann, Marco K. et al. 2016. "Self-Other Mergence in the Frontal Cortex during Cooperation and Competition." *Neuron* 91(2): 482–93. <http://dx.doi.org/10.1016/j.neuron.2016.06.022>.
- Zoh, Yoonseo, Steve W.C. Chang, and Molly J. Crockett. 2022. "The Prefrontal Cortex and (Uniquely) Human Cooperation: A Comparative Perspective." *Neuropsychopharmacology* 47(1): 119–33.

REVIEWER COMMENTS

Reviewer #1 (Remarks to the Author):

Generally, the authors did a great job at addressing my concerns, however there are still a few points of confusion and concern that need to be addressed:

Points 1-3) Thank you for clarifying how the predictions of the two experts are not constrained to be anti-correlated by the model, and providing data supporting that, with the predictions of the two experts taking different trajectories in different participants depending on best-fitting parameters (point 0). However (point 1), it still seems to me that the behavioral signatures of the two strategies are not independent from each other. On every choice the participant can either stay (=signature of cooperation) or switch (=signature of competition), meaning that the two strategies cannot be independently characterized. Is that correct? As for the logistic regression analyses (point 2), to me the results do indeed suggest that the cooperative expert can recover the cooperative strategy, but the competitive expert cannot. However, those analyses do not appear to suggest the opposite, i.e. the competitive expert being able to recover the competitive strategy. Right now, the data suggests that the competitive expert may simply be making random choices (i.e. not different than 50% stay/50% switch). This may be due to the point above, i.e. the signatures (stay or switch) not being independent from one another. I think a more specific signature needs to be found here, at least for the competitive expert.

Point 4) On p.4-5 of the rebuttal letter, the authors mention and show that the two reliability signals are uncorrelated. Then, on p.6, they write: “when one expert is reliable, the other one is often unreliable. Globally, when one expert has a precise prediction regarding the best future choice for the participant, the other expert often proposes a less reliable choice”. Those two statements seem to contradict each other, so it would be great if the authors could further clarify. Are the two experts’ reliability uncorrelated or anti-correlated?

Point 5) Finally, not finding separate neural signatures of the two strategies is problematic as it is inconsistent with the claims of the behavioral model, which assumes two ‘experts’ learning in parallel – to do that, the experts should rely on at least partially separable neural mechanisms. If the reliability signals of the two experts are uncorrelated, maybe the two reliability signals could be dissociated in the brain (whereas it may make sense that the DV and PE are found in the same reward-related areas)? If reliability is related to precision more than it is related to reward, it may be a better neural signal to distinguish the two experts. If that’s not the case, I believe this should be addressed as a limitation in the discussion, rather than trying to convey this as an expected finding in the discussion (page 14). While I can agree with the interpretation that the two experts are implementing the same cognitive process with different priors depending on the context (contrary to other existing papers on arbitration), I find this interpretation inconsistent with the behavioral model, which assumes two separate experts learning

in parallel independently from each other. If the authors assume a single cognitive process, why even consider a behavioral model with two independent experts in the first place, rather than one learning algorithm which makes different inferences depending on the prior/context?

Point 12) I appreciate that the authors have added a new analysis to address this point (page 19), however, I do not understand what analysis was actually run and what the coefficients of variations reported correspond to.

Point 14) The confusion matrix presented is problematic. While it is reassuring to see that the winning model is recovered, any model that cannot be appropriately recovered probably should not be included in the set to start with. Moreover, I find it quite worrying that a lot of the ToM models are better recovered by a random bias model. This suggests that those models are wrongly defined and non-identifiable to start with, and this should be addressed to ensure ToM models can be appropriately recovered. Otherwise, claims such as those made in the discussion about the strength of the computational approach (p.13-14) cannot be made.

Two quick comments about the abstract:

- The authors changed the title in response to reviewer #3, however, the abstract is still overall consistent with the old title. The aim of this paper does not appear to address “how the brain decides whether others’ intentions are to cooperate or compete” – I do not believe that the computational model or the fMRI analyses answer this question, as this is not a choice that participants are asked to make in the task.
- In the abstract, the sentence “The ventral striatum and ventromedial prefrontal cortex tracked the reliability of this arbitration process” seems incorrect as those regions track the reliability difference between the two strategies and there is no reliability associated with the arbitration process itself.

Reviewer #2 (Remarks to the Author):

The authors have addressed all my questions and comments. I support the publication of this paper in Nature Communications.

Reviewer #3 (Remarks to the Author):

I have read the response to reviewers. What a lot of work goes into these responses.

Reviewer #1 (Remarks to the Author):

Generally, the authors did a great job at addressing my concerns, however there are still a few points of confusion and concern that need to be addressed:

Points 1-3) Thank you for clarifying how the predictions of the two experts are not constrained to be anti-correlated by the model, and providing data supporting that, with the predictions of the two experts taking different trajectories in different participants depending on best-fitting parameters (point 0). However (point 1), it still seems to me that the behavioral signatures of the two strategies are not independent from each other. On every choice the participant can either stay (=signature of cooperation) or switch (=signature of competition), meaning that the two strategies cannot be independently characterized. Is that correct?

The signature of cooperation of the participants is indeed the probability of staying on the same target, and the signature of competition is the probability of switching to the other target. Thus, it is correct that these two probabilities are not independent because if one is probability p the other one is $1-p$.

As for the logistic regression analyses (point 2), to me the results do indeed suggest that the cooperative expert can recover the cooperative strategy, but the competitive expert cannot. However, those analyses do not appear to suggest the opposite, i.e. the competitive expert being able to recover the competitive strategy. Right now, the data suggests that the competitive expert may simply be making random choices (i.e. not different than 50% stay/50% switch). This may be due to the point above, i.e. the signatures (stay or switch) not being independent from one another. I think a more specific signature needs to be found here, at least for the competitive expert.

As stated in our previous point 2, our two logistic panel data regressions (one for the cooperative model alone and one for the competitive model alone) do indicate that each expert is able to reproduce its corresponding behavioral signature. Indeed, the probability to stay of the competitive expert was 0.4966 (CI[0.4807; 0.5105]), i.e. the probability to switch was $1-0.4966=0.5034$ ($P=0.53$). This indicates that the competitive expert is able to recover the competitive strategy (playing $P(\text{Switch})=0.5$). Even if this may appear as random choices when considering all trials on average (50% stay/50% switch), this is not the case because the underlying process which leads to this average of 50% switch is the one describe by Hampton et al., (*PNAS*, 2008) (thus reproducing participants' behavior during the competitive block).

Thus, we do not think that it is necessary to define a new behavioral competitive signature for the participants. As noted before, we still think that the definitions of the behavioral signatures are justified by the fact that the Nash equilibrium can be used to identify two independent behavioral strategies. When the subject plays the matching pennies game, the Nash equilibrium is to switch the target with the probability of 0.5 in order to be unpredictable. By contrast, when they play the coordination game, there is a pure-strategy Nash equilibrium which is to always choose the same target. Although the coordination game also has a mixed strategy equilibrium, this is not evolutionarily stable and produces less desirable outcomes for both players (i.e., Pareto dominated

by the pure Nash equilibrium). Therefore, the probability of staying or switching is much more likely to deviate from 0.5 when the subject plays the coordination game than the competitive matching-pennies task.

Point 4) On p.4-5 of the rebuttal letter, the authors mention and show that the two reliability signals are uncorrelated. Then, on p.6, they write: “when one expert is reliable, the other one is often unreliable. Globally, when one expert has a precise prediction regarding the best future choice for the participant, the other expert often proposes a less reliable choice”. Those two statements seem to contradict each other, so it would be great if the authors could further clarify. Are the two experts’ reliability uncorrelated or anti-correlated?

The two reliability signals of the two experts are uncorrelated: $R^2 = 0.005$ ($p = 0.148$) expert (as stated in previous point 4 and figure II). They are not anti-correlated. What we meant in our previous response was simply that the absence of correlation between the two reliability signals shows that the two strategies are not necessarily reliable at the same time.

We recognize that the statement we previously made on p.6 of our rebuttal letter regarding the Decision Value is misleading and we have removed the sentences: ‘when one expert is reliable, the other one is often unreliable. Globally, when one expert has a precise prediction regarding the best future choice for the participant, the other expert often proposes a less reliable choice’.

Point 5) Finally, not finding separate neural signatures of the two strategies is problematic as it is inconsistent with the claims of the behavioral model, which assumes two ‘experts’ learning in parallel – to do that, the experts should rely on at least partially separable neural mechanisms. If the reliability signals of the two experts are uncorrelated, maybe the two reliability signals could be dissociated in the brain (whereas it may make sense that the DV and PE are found in the same reward-related areas)? If reliability is related to precision more than it is related to reward, it may be a better neural signal to distinguish the two experts. If that’s not the case, I believe this should be addressed as a limitation in the discussion, rather than trying to convey this as an expected finding in the discussion (page 14). While I can agree with the interpretation that the two experts are implementing the same cognitive process with different priors depending on the context (contrary to other existing papers on arbitration), I find this interpretation inconsistent with the behavioral model, which assumes two separate experts learning in parallel independently from each other. If the authors assume a single cognitive process, why even consider a behavioral model with two independent experts in the first place, rather than one learning algorithm which makes different inferences depending on the prior/context?

We did not find separate brain regions when comparing the 2 experts directly ($P > 0.96$). We acknowledge that an absence of finding of separate neural signatures of the two strategies may be seen as inconsistent with the behavioral model, which assumes two ‘experts’ learning in parallel. However, this view is implicitly assuming that the algorithmic and the brain system levels should be similar, while they we observe that they can be distinguished (Lockwood, Apps, and Chang 2020). Separate expert systems can be observed at the behavioral and computational levels for

cooperation and competition, but it does not necessarily follow that separate (distinct) brain systems have to be assumed for these experts

Moreover, reliability alone is not a better neural signal to distinguish the two experts than the reliability difference because only the Mixed-intentions Influence model (and not the influence cooperative or the competitive influence models alone) successfully reproduced the effect of the Cooperativity signature of AA on the probability to stay, as in the participants (Fig. 3b and Extended Fig.1, see SI), on the probability to stay, as was the case for the participants (Fig. 3b and Extended Fig.1, see SI), and it is precisely this behavior which is important to reproduce.

The reviewer found that not finding separate neural signatures of the two strategies is problematic as it is inconsistent with the behavioral model which assumes two ‘experts’ learning in parallel. For reasons of parsimony, we assumed a single process (influence learning) with different priors for each expert. However, even if we assume this single cognitive process, we still need two experts to have distinct priors depending on the cooperative or competitive context. We recognize that it was not *a priori* hypothesized that the mixed intentions influence model would be the winning model (in fact based on our previous work, we initially expected that a POMDP model would be the winning model, as in Khalvati et al., *Science Advances*, 2019). This is why, as stated in the introduction, we tested not only non-Bayesian vs Bayesian models and non-mentalizing vs mentalizing models, but also mixture of models. However, the results strongly demonstrate the ‘mixed intentions influence’ model was most frequently the best fit across the population (Fig. 2b), indicating that arbitration between a cooperative and a competitive expert best explains observed participants’ behavior, rather than either expert taken individually (called ‘Inf comp’ for Influence competition model and ‘Inf coop’ for Influence cooperative model alone in Fig 2b). Moreover, only the Mixed-intentions Influence model (and not the cooperative or the competitive one) successfully reproduced the effect of the Cooperativity signature of the AA on the probability to stay, as in the participants (Fig. 3b and Extended Fig.1, see SI).

To tease the reviewer’s concerns to rest, we have added the following paragraph in the discussion to acknowledge the difficulty of the interpretation :

The fact that we did not observe that each expert is encoded in separate brain regions may be seen as a limitation, since the behavioral model assumes two ‘experts’ learning in parallel. However, the algorithmic and the brain system levels have to be distinguished⁵⁹. Separate expert systems can be observed at the behavioral and computational levels for cooperation and competition, but it does not necessarily follow that separate brain systems must be assumed for these experts. Indeed, the two experts, each using different priors, implement the same cognitive process (i.e., influence learning). This is a different situation than previous mixture of models theories which weigh experts implementing different cognitive processes²⁹. This is a key conceptual distinction which extends the mixture of models to a more general view in which the experts are differentiated not by their cognitive processes but by their priors.

Point 12) I appreciate that the authors have added a new analysis to address this point (page 19), however, I do not understand what analysis was actually run and what the coefficients of variations reported correspond to.

As noted in our original point 12, we performed a marginal effect of panel data logistic regression clustered by subject of switch strategy on the mode of interaction (competition or cooperation), the

cooperativity signature of the AA and their interactions. Thus, the reported coefficients of variations are the marginal effects in this analysis.

Point 14) The confusion matrix presented is problematic. While it is reassuring to see that the winning model is recovered, any model that cannot be appropriately recovered probably should not be included in the set to start with. Moreover, I find it quite worrying that a lot of the ToM models are better recovered by a random bias model. This suggests that those models are wrongly defined and non-identifiable to start with, and this should be addressed to ensure ToM models can be appropriately recovered. Otherwise, claims such as those made in the discussion about the strength of the computational approach (p.13-14) cannot be made.

We thank the reviewer for pointing out that some of the tested models cannot be recovered. The following candidate models fail to pass model recovery: HGF, 1-TOM coop, Mixed intention 1-ToM, 2-ToM coop, 2-ToM comp, Mixed intention 2-ToM, active inference and fictitious model. Note that we kept all the 1-ToM in the new confusion matrix for completeness, even though the 1-ToM coop and the Mixed intention 1-ToM were not recovered. We have now amended the model space and the confusion matrix is now displayed below and in **Extended Fig. 5**. We have also updated Fig 2b to report only the new set of models, removed from Fig. 2b the models which did not pass the model recovery, and updated Table 1 with the new set of models. The specifications of all models (including HGF, Mixed intention 1-ToM, 2-ToM coop, 2-ToM comp, Mixed intention 2-ToM, active inference and fictitious model) are also left for completeness in the Supp. Mat., but we no longer discuss them the paper.

Therefore we have changed the following sentence in the discussion:

One strength of our computational approach was to assess and compare a large variety of competing models, such recursive learning model 1-TOM, influence models for only cooperative strategies or competitive strategies, a mixture of experts using influence models, Bayesian Sequence learner, Reinforcement Learning and Heuristic models. Many have never previously been directly tested against each other. Other models (active inference, fictitious learning and Hierarchical Gaussian Filter) were also tested but did not pass the model recovery analyses, and are only reported in the Supp. Mat. for completeness.

Confusion matrix. We generated 5 datasets for each of our 31 participants (total of 155 datasets). To do so we used randomly generated free parameters with mean value, the fitted free parameters of each participant independently. The rows are the generative models. The columns are the predicted model. The colors represent the probability that one model is more frequent (in our population of generated dataset) than another given a Bayesian model selection (Exceedance probability)

Two quick comments about the abstract:
 - The authors changed the title in response to reviewer #3, however, the abstract is still overall consistent with the old title. The aim of this paper does not appear to address “how the brain decides whether others’ intentions are to cooperate or compete” – I do not believe that the computational model or the fMRI analyses answer this question, as this is not a choice that participants are asked to make in the task.

We have changed the sentence ‘How does the brain decide whether the others’ intentions are to cooperate or compete when the nature of the interactions is not explicitly and truthfully signaled?’ to: *How does the brain adapt to fluctuating intentions of others’ when the nature of the interactions (to cooperate or compete) is not explicitly and truthfully signaled?*

- In the abstract, the sentence “The ventral striatum and ventromedial prefrontal cortex tracked the reliability of this arbitration process” seems incorrect as those regions track the

reliability difference between the two strategies and there is no reliability associated with the arbitration process itself.

We thank the reviewer for pointing out the missing word ‘difference’. The sentence now reads:

The ventral striatum and ventromedial prefrontal cortex tracked the reliability difference between these experts.

The new abstract now reads:

Humans frequently interact with agents whose intentions can fluctuate between competition and cooperation over time. How does the brain adapt to fluctuating intentions of others when the nature of the interactions (to cooperate or compete) is not explicitly and truthfully signaled? We used model-based fMRI and a task in which participants thought they were playing with another player. In fact, they played with an algorithm that alternated without signaling between Cooperative and Competitive strategies. A neurocomputational mechanism with arbitration between competitive and cooperative experts outperformed other learning models in predicting choice behavior. The ventral striatum and ventromedial prefrontal cortex tracked the difference of reliability between these experts. When attributing competitive intentions, these regions increased their coupling with a network that distinguished prediction error related to competition versus cooperation. These findings provide a neurocomputational account of how the brain arbitrates dynamically between cooperative and competitive intentions when making adaptive social decisions.

REVIEWERS' COMMENTS

Reviewer #1 (Remarks to the Author):

I would like to thank the authors for taking the time to address my remaining comments and I am happy to support publication of the manuscript at that stage.

One final recommendation I would like to make is that more explanation about how to run the code shared on github to reproduce the results and figures would be appreciated. Maybe the authors are still in the process of updating that repository, but if not, it seems like what has been shared would only reproduce the model fitting, but none of the other analyses?

Reviewer #1 (Remarks to the Author):

I would like to thank the authors for taking the time to address my remaining comments and I am happy to support publication of the manuscript at that stage. One final recommendation I would like to make is that more explanation about how to run the code shared on github to reproduce the results and figures would be appreciated. Maybe the authors are still in the process of updating that repository, but if not, it seems like what has been shared would only reproduce the model fitting, but none of the other analyses?

We are now providing more explanations about how to run the code shared on github to reproduce the results and figures. This repository now includes not only how to reproduce the model fitting, but also the other analyses.

We have updated the following links to share the data in a public repository. All display items (figures and tables) in the main manuscript and supplementary information can be reproduced using these data.

DOI for github with data and scripts : DOI: 10.5281/zenodo.10299140

<https://zenodo.org/records/10299140>

Link to github : https://github.com/remiphilipp/Mixture_intention.git

Link to fMRI data : <https://neurovault.org/collections/EOCXPHRJ/>